# 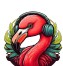 Audio Flamingo 2: An Audio-Language Model with Long-Audio Understanding and Expert Reasoning Abilities

**Sreyan Ghosh** [* 1 2]  **Zhifeng Kong** [1]  **Sonal Kumar** [2]  **S Sakshi** [2]  **Jaehyeon Kim** [1]  **Wei Ping** [1]  **Rafael Valle** [1]
**Dinesh Manocha** [2]  **Bryan Catanzaro** [1]

Project: https://research.nvidia.com/labs/adlr/AF2/

## Abstract

Understanding and reasoning over non-speech sounds and music are crucial for both humans and AI agents to interact effectively with their environments. In this paper, we introduce **Audio Flamingo 2** (AF2), an Audio-Language Model (ALM) with advanced audio understanding and reasoning capabilities. AF2 leverages (i) a custom CLAP model, (ii) synthetic Audio QA data for fine-grained audio reasoning, and (iii) a multi-stage curriculum learning strategy. AF2 achieves state-of-the-art performance with only a 3B parameter small language model, surpassing large open-source and proprietary models across over 20 benchmarks. Next, for the first time, we extend audio understanding to long audio segments (30 secs to 5 mins) and propose **LongAudio**, a large and novel dataset for training ALMs on long audio captioning and question-answering tasks. Fine-tuning AF2 on LongAudio leads to exceptional performance on our proposed **LongAudioBench**, an expert-annotated benchmark for evaluating ALMs on long audio understanding capabilities. We conduct extensive ablation studies to confirm the efficacy of our approach.

## 1. Introduction

Understanding non-speech sounds, non-verbal speech, and music (collectively referred to as "audio" in this paper) is essential for real-world applications such as detecting anomalies in industrial environments, recognizing emotional cues, and improving assistive technologies for the

*Work done during an internship at NVIDIA. [1]NVIDIA, Santa Clara, CA, USA [2]University of Maryland, College Park, MD, USA. Correspondence to: Sreyan Ghosh <sreyang@umd.edu>, Zhifeng Kong <zkong@nvidia.com>.

*Proceedings of the $42^{nd}$ International Conference on Machine Learning*, Vancouver, Canada. PMLR 267, 2025. Copyright 2025 by the author(s).

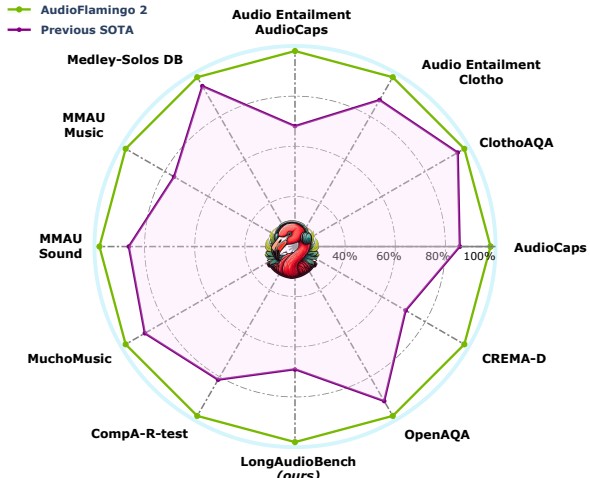

Figure 1: Audio Flamingo 2 versus previous SOTA ALMs on audio understanding and reasoning benchmarks (values normalized). AF2 outperforms all baselines and has smaller model footprints.

impaired. While Large Language Models (LLMs) have demonstrated remarkable reasoning capabilities through language, extending these systems to comprehend audio is key to building intelligent systems capable of reasoning with contextual auditory cues (Kong et al., 2024). Verbal speech, inherently tied to language, benefits significantly from (L)LM advancements (Watanabe et al., 2018; Chen et al., 2024a); however, the potential to enhance perception and reasoning over non-verbal audio remains largely underexplored (Ghosh et al., 2024c). Audio-Language Models (ALMs) extend language models with audio understanding capabilities. Contrastive Language-Audio Pre-training (CLAP) (Elizalde et al., 2023a) was among the first encoder-only ALMs to bridge audio and language with contrastive learning. Building on this, subsequent efforts introduced Large ALMs (LALMs), which integrate audio encoders with pre-trained decoder-based LLMs, enabling open-ended Audio Question Answering (AQA) and free-form response generation (Gong et al., 2021; Tang et al., 2024; Ghosh et al., 2024c). However, despite these developments, even the most advanced LALMs continue to underperform on expert-level reasoning tasks compared to foundational tasks like

event classification. For example, Gemini-1.5-Pro (Team et al., 2024), one of the most advanced models, achieves only 54.4% and 48.5% on the MMAU sound and music subsets (Sakshi et al., 2024), a benchmark for evaluating expert-level audio reasoning. This underscores the challenges of improving an LLM's ability to understand and reason over audio, which we attribute to the lack of high-quality training data and robust audio representations.

**Main Contributions:** In this paper, we propose **Audio Flamingo 2** (AF2), a parameter-efficient ALM that combines a 3B-parameter small decoder LM with a 203M-parameter audio encoder, achieving state-of-the-art audio understanding and reasoning capabilities. There are three major innovations in our method: (1) **Data:** Recent studies highlight that improving data quality can rival or even surpass the performance gains achieved by scaling compute and model size (Abdin et al., 2024). To this end, we propose **AudioSkills** (Section 4.1), a large-scale, skill-specific AQA training dataset featuring complex, reasoning-intensive questions paired with each audio. We design questions that target *seven* distinct skills, with the primary aim of improving fine-grained reasoning capabilities in ALMs. (2) **Audio Encoding:** We propose AF-CLAP (Section 3.1), where we scale CLAP training to over 8M audio-caption pairs, incorporating synthetic data, and propose an improved contrastive loss for better representational quality and robustness. (3) **Training Strategy:** We propose a novel 3-stage curriculum training strategy (Section 5) for improved performance.

Additionally, for the first time, we extend audio understanding to *long audios*, moving beyond 30-second clips to audios lasting up to 5 minutes. To enable the model to comprehend and reason over long audio, we introduce **LongAudio**, a novel dataset comprising over 260k carefully curated AQA instances with audios ranging from 30 seconds to 5 minutes. LongAudio spans 10+ audio categories and supports 6 tasks, including captioning and 5 reasoning-based QA tasks. Next, we introduce **LongAudioBench**, an expert-annotated benchmark for evaluating ALMs on long audio understanding. AF2, trained on LongAudio, significantly outperforms our baseline. In summary, our main contributions are:

1. We present Audio Flamingo 2, a SOTA ALM with advanced audio understanding and reasoning capabilities.

2. We propose innovations in data generation, architecture design, representation learning, and training strategies.

3. We introduce the long audio understanding task and create dedicated training and evaluation datasets to drive progress in this area.

4. Audio Flamingo 2 outperforms larger and proprietary LALMs across over 20 benchmarks, despite being smaller and trained exclusively on public datasets.

5. We conduct systematic ablation studies to demonstrate the impact of each design choice.

## 2. Related Work

**Audio-Language Models:** ALMs can be classified into two broad categories: *1) Encoder-only ALMs:* Encoder-only ALMs are a class of Multi-Modal Language Models (MLLM) that learn a shared space between the audio and language modalities with an encoder-only LM and an audio encoder. CLAP, a pioneering encoder-based ALM inspired by CLIP (Radford et al., 2021), showed state-of-the-art performance on audio-language tasks like retrieval, zero-shot classification, etc. Following this, several attempts have been made to improve CLAP by scaling data (Wu et al., 2022b), incorporating additional training objectives (Ghosh et al., 2024d), or with synthetic data (Ghosh et al., 2025). Other notable works include Wav2CLIP (Wu et al., 2022a), AudioClip (Guzhov et al., 2022) and CoL-LAT (Silva et al., 2023). *2) Decoder-based ALMs:* With the advent of LLMs, Pengi (Deshmukh et al., 2023), a pioneering decoder-based ALM, achieved SOTA results on a variety of audio classification tasks. Following Pengi, a large number of Large ALMs were introduced, including fully open-source models like LTU (Gong et al., 2024), LTU-AS (Gong et al., 2023a), SALMONN (Tang et al., 2024), AudioGPT (Huang et al., 2024), GAMA (Ghosh et al., 2024c), Audio Flamingo (Kong et al., 2024), and open-access models like Qwen-Audio (Chu et al., 2023) and Qwen-2-Audio (Chu et al., 2024). A majority of the advances have focused on scaling model size and datasets, with very little advancements in data quality or audio encoder representations. This has eventually translated to performance advancing on foundational tasks like classification and captioning but under-performing on expert-level reasoning, the skill required for advancing towards Artificial General Intelligence (AGI) (Morris et al., 2024).

**Long Audio Understanding:** Current ALMs are limited to perceiving at most 30 seconds of audio (Chu et al., 2024), with a majority confined to at most 10 seconds (Ghosh et al., 2024c; Gong et al., 2024). This can be attributed to the fact that these models employ audio encoders that support only short audio encoding and datasets where a majority of the data is only at most 10 seconds. While long speech has received some attention (Chiu et al., 2019; Arumugam et al., 2023), and the field on long video understanding has seen advancements recently (Weng et al., 2025; Xue et al., 2024), to the best of our knowledge, no current work or datasets exist for long audio understanding and reasoning.

## 3. Audio Flamingo 2 Architecture

Fig. 2 summarizes the AF2 architecture, consisting of four primary components: *i)* AF-CLAP: a CLAP-based audio encoder with sliding window feature extraction, *ii)* audio representation transformation layers for additional capacity, *iii)* a decoder-only language model, and *iv)* gated cross-attention (XATTN-Dense) layers for audio conditioning.

Figure 2: Overview of **Audio Flamingo 2**'s cross-attention architecture and three-stage curriculum training.

### 3.1. AF-CLAP Audio Encoder

CLAP, pre-trained with contrastive loss on audio-caption pairs, shows strong audio understanding and natural language alignment (Wu et al., 2022b; Elizalde et al., 2023a). These make CLAP a suitable choice for building ALMs. However, CLAP is less favored in prior works (Tang et al., 2023; Chu et al., 2023) due to its under-performance compared to SSL pre-trained audio encoders (Gong et al., 2023b; LI et al., 2024; Ghosh et al., 2024c). We hypothesize this is due to the limited availability of high-quality audio-caption pairs, which causes CLAP representations to struggle with compositional reasoning (Ghosh et al., 2024d) and linguistic variations in captions (Selvakumar et al., 2024).

In this section, we address these issues and introduce an improved version of CLAP called AF-CLAP. Specifically, we focus on: *i)* constructing a large-scale, high-quality training dataset, and *ii)* improving the training objective to for better representational robustness.

#### 3.1.1. AF-CLAP TRAINING DATASET

We scale the training dataset for AF-CLAP to over 8M (10-second) audio-caption pairs. We collect these data from open-source audio and video datasets, with an emphasis on audio diversity and caption accuracy.

Existing models like Laion-CLAP Wu et al. (2022b) and MS-CLAP (Elizalde et al., 2023b) rely heavily on single-label audio classification datasets for captions. This limits their ability to generalize to complex, real-world audio with diverse compositions. Automated captioning efforts (e.g., Yuan et al. (2024)) yield only about 1.4M pairs, lack diversity, and under-represent critical domains like home sounds.

Inspired by the recent success of training vision LMs on images from long videos (Venkataramanan et al., 2024), we collect audio from open long-video datasets. Specifically, we select 100k diverse videos from MiraData (Ju et al., 2024) and Video Recap (Islam et al., 2024) (see Appendix H.1 for selection details). We segment these videos into 10-second clips and generate video captions using Qwen2-VL-2B-Instruct and audio captions using Qwen2-Audio. To ensure diversity and reduce redundancy, we filter out segments with audio-visual similarity above a threshold $p$. We then prompt GPT-4o(2024-05-13) (Prompts 23 and 22) to generate audio-centric captions that exclude visual attributes and emphasize sound events. Using this approach, we collect approximately **5.5M** new audio-caption pairs, including 4M from unlabeled short audios and 1.5M from long videos. Detailed dataset statistics are provided in Table 9.

#### 3.1.2. AF-CLAP TRAINING OBJECTIVE

Compared to the standard contrastive loss in audio-language pre-training (Wu et al., 2022b; Elizalde et al., 2023a), we improve the training objective for better robustness to linguistic variations and compositional reasoning abilities.

**Improving Linguistic Invariance:** CLAP-like models struggle to generalize to linguistic variations in captions that humans easily understand (Selvakumar et al., 2024) (e.g., failing to equate *helicopter* and *chopper*). To address this, for every caption in the dataset, we generate $M-1$ linguistically varied captions with identical semantics and composition. These variations, along with the ground-truth caption, are treated as *positives*.

**Improving Compositional Reasoning:** Captions with different word orders or structures often convey distinct relationships between acoustic events, such as temporal sequencing or attribute binding. However, CLAP-like models struggle to capture these nuances (e.g., differentiating whether one sound follows or precedes another) (Ghosh et al., 2024d). To address this, we introduce composition-aware negatives. For every $M$ positive captions, we generate $N$ variations with modified temporal or attribute compositions and use them as *negatives*. An example is below:

> **Original Caption:** A dog barking followed by the sound of a train approaching.
> **Positive:** A dog barking followed by the sound of a railcar approaching.
> **Negative:** A dog barking preceded by the sound of a railcar approaching.

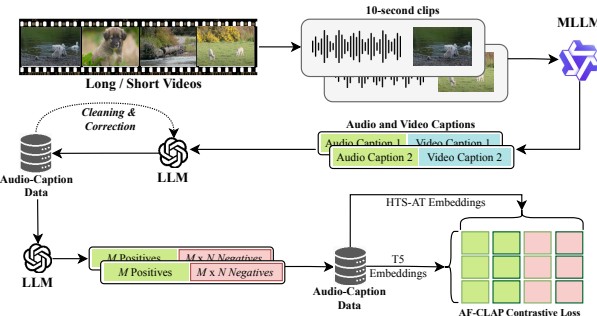

Figure 3: Illustration of AF-CLAP training process. We collect long and short videos, segment it into 10-second clips, caption it, and prompt an LLM to generate audio captions. The data is then used to train CLAP with a modified contrastive loss.

**Contrastive Loss.** Each sample $x$ in the training data $\mathcal{X}$ now has $M$ positives $\{\mathcal{P}(x)_m\}_{m=1}^M$ and $M \times N$ negatives $\{\mathcal{N}(x)_{m,n}\}_{m=1,n=1}^{M,N}$. Let $\mathcal{A}(x)$ be audio embedding from the HTSAT$_\text{large}$ audio encoder (Chen et al., 2022) with MLP projection, and $\mathcal{T}(x)$ be the text embedding from Flan-T5 text encoder (Raffel et al., 2020; Chung et al., 2024) with MLP projection. Let $\texttt{s}(u, v) = \exp(u^\top v / \tau)$ be the similarity function with temperature $\tau$. Our training objective is:

$$\mathcal{L} = -\frac{1}{B} \sum_{i=1}^{B} \log \frac{S(i,i)}{S_\text{neg}(i) + \sum_{j=1}^{B} S(j,i)},$$

where

$$S(j,i) = \sum_m \texttt{s}(\mathcal{T}(\mathcal{P}(x_j)_m), \mathcal{A}(x_i)),$$

$$S_\text{neg}(i) = \sum_{m,n} \texttt{s}(\mathcal{T}(\mathcal{N}(x_i)_{m,n}), \mathcal{A}(x_i)),$$

where $B$ is the batch size. Our approach encourages CLAP to learn both linguistic invariance and compositional reasoning, aligning its capabilities more closely with human-like understanding and thus providing more human-aligned representations. AF-CLAP achieves SOTA performance on various retrieval and audio classification benchmarks. Since comparing CLAP performance is beyond our scope, we compare results in Tables 7 and 10.

### 3.2. Audio Conditioning Architecture & LLM

**Audio Feature Extraction.** For each 10-second segment, we extract dense audio features $h \in \mathbb{R}^{64 \times 2048}$ from the penultimate layer of AF-CLAP. This approach yields higher-quality features compared to mean-pooled representations from the final layer (see Appendix B.2). For longer audio, we use non-overlapping sliding windows to compute and concatenate audio features. The maximum number of sliding windows varies across training stages, with up to 30 windows (5 minutes) when training on LongAudio. Once the sliding window features are obtained, we apply

RoPE (Su et al., 2023) with a base of 4096 to encode temporal information into the features.

**Representation Transformation Layers.** To expand model capacity, we apply three self-attention layers to the audio feature representations, each with 8 heads and an inner dimension of 2048 (Kong et al., 2024; Vaswani et al., 2017).

**Gated Cross-Attention.** Following Audio Flamingo, we use gated cross-attention dense (XATTN-Dense) layers from Flamingo (Alayrac et al., 2022) to condition audio representations on the LLM. Each layer consists of two blocks: (1) a residual block with cross-attention and tanh gating and (2) a residual block with a dense layer and tanh gating. These layers are inserted before each LLM block.

The XATTN-Dense layers reduces the quadratic attention complexity in prefix tuning to linear complexity. For instance, let $l_1 = 80$ be the text token length and $l_2 = 30 \times 64$ be the number of audio embeddings. Prefix tuning requires a self-attention complexity of $(l_1 + l_2)^2 = 4 \times 10^6$, whereas our cross-attention complexity is around $l_1 \times l_2 \approx 1.5 \times 10^5$.

**Frozen Language Model.** Our architecture uses Qwen2.5-3B (Yang et al., 2024a), a decoder-only causal LLM with 3B parameters, 36 hidden layers, and 16 attention heads. We find this model to have the best cost-performance ratio, as it has enough capacity for audio understanding and is light enough (see Section 6.6 for a comparison of LLM sizes).

## 4. Audio Flamingo 2 Training Data

Table 23 summarizes the datasets used to train AF2. We first convert common benchmark datasets used in Audio Flamingo to AQA format (see Appendix F.1). In addition, we propose two datasets: AudioSkills for audio expert reasoning and LongAudio for long audio understanding.

### 4.1. AudioSkills: An Expert Audio Reasoning Dataset

Expert-level reasoning requires mastery of diverse and specialized skills (Ericsson, 2003; Huang & Chang, 2022). However, most existing audio datasets focus on surface-level properties, such as acoustic events or category classification. This limitation extends to QA pairs derived from these datasets, which often fail to require expert-level reasoning.

To address this gap, we introduce **AudioSkills**, a high-quality, skill-specific synthetic dataset designed to prioritize the development of reasoning and problem-solving abilities. This dataset is carefully curated to ensure diversity and relevance, grounded in the hypothesis that expert reasoning emerges from the mastery of various relevant skills and world knowledge. Specifically, we use a combination of open-source sound and music datasets and synthetic audio and prompt GPT-4o, along with any available metadata, to create QA pairs. AudioSkills focuses on audios $\leq 30$ seconds long to enable effective scaling on open-source

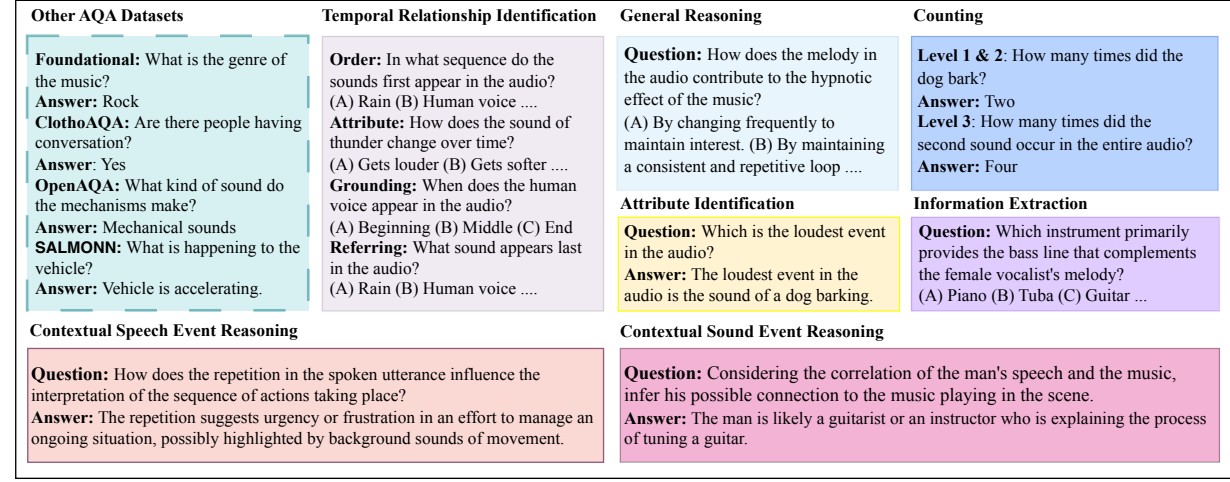

Figure 4: Examples from **AudioSkills**. Compared to other AQA datasets, questions in AudioSkills require deliberate reasoning.

datasets. It spans *seven* distinct skills selected for their plausibility, relevance, and ablation insights as follows:

1. **Temporal Reasoning:** Understanding temporal relationships between acoustic events in the audio. We generate 4 types of MCQ-type QA pairs: *i) Temporal Order:* Understanding the sequential order of events or sounds; *ii) Temporal Attribute:* Understanding how sound attributes change over time; *iii) Temporal Referring:* Answering questions referring to sounds at specific temporal locations in the audio (e.g., start, middle, end); *iv) Temporal Grounding:* Identifying the temporal location of specific acoustic events.

2. **Attribute Identification:** Recognizing attributes of events in multi-event audio, such as sound characteristics (e.g., *loud* bang) or gender (*e.g., male speech*). We generate QA pairs using attributes extracted via the method proposed by Kumar et al. (2024).

3. **Counting:** Counting occurrences of specific sounds in an audio. We use the Synthio TTA model (Ghosh et al., 2024b) to create QA pairs at 3 difficulty levels: *i) Level 1:* Single sounds concatenated; count the event occurrences. *ii) Level 2:* Multiple interleaved sounds; count the main sound occurrences. *iii) Level 3:* Same as Level 2 but referenced by their temporal position.

4. **Contextual Sound Event Reasoning:** Identifying the purpose of a sound or acoustic event in the context of other sounds and events in the audio. This skill requires audio understanding, temporal reasoning, world knowledge, and various other skills. Inspired by CompA-R in GAMA (Ghosh et al., 2024c), we expand the dataset from ≈200k to ≈350k QA pairs.

5. **Contextual Speech Event Reasoning:** Similar to Contextual Sound Event Reasoning but focused on identifying the purpose of spoken utterances in relation to other sounds or events.

6. **Information Extraction:** Focuses on understanding characteristics of the audio beyond just surface-level properties, detailed content analysis, and the application of external world knowledge when necessary.

7. **General Reasoning:** This category encompasses questions that do not fall into the specific skill types above but require a combination of multiple skills or unique abilities, such as identifying relationships between multiple events or interpreting complex scenarios.

In total, we generate ≈4.2M QA pairs. Figure 4 compares AudioSkills to other open-source AQA datasets, highlighting that these datasets lack the complexity required for deliberate reasoning. While training on such QA pairs is useful for alignment (Zhou et al., 2024; Wolf et al., 2023), it does not equip models with the specialized skills needed to handle complex questions (Sakshi et al., 2024; Ghosh et al., 2024a). *Additional details, including statistics, metadata, and prompts, are provided in Appendix D.1 and Table 14.*

### 4.2. LongAudio: A Long Audio Understanding Dataset

We construct **LongAudio**, the first large-scale long audio understanding dataset, which is comprised of over 80K unique audios and approximately 263K AQA pairs. Audios are sourced from open long-video datasets, including a subset of MiraData(Ju et al., 2024) (featuring diverse content from natural scenes to gaming) and the entire Video Re-Cap(Islam et al., 2024) (egocentric videos of daily activities). To ensure diversity, we filter MiraData by clustering videos and carefully selecting samples from each cluster (see Appendix H.1). Fig. 5 (left) categorizes these videos and topics across several domains. We generate captions for video and audio segments using Qwen2-VL-2B-Instruct and Qwen2-Audio, respectively. GPT-4o is then used to

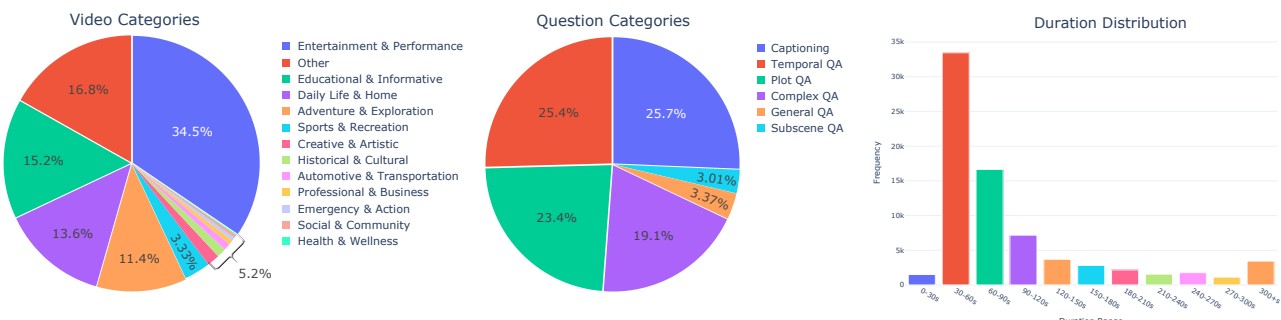

Figure 5: The proportion of video categories (for sourcing audios) (**left**), question categories (**middle**), and distribution of durations (**right**) for the LongAudio dataset with 262,928 unique AQA and 80k unique audios. We target captioning and reasoning tasks.

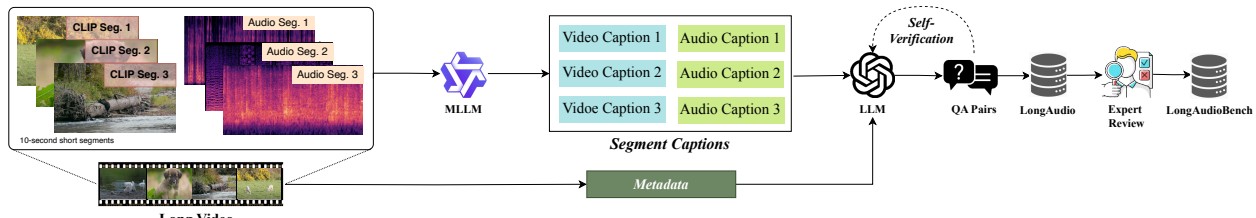

Figure 6: The pipeline for generating **LongAudio**. The process begins by segmenting the long video into short video and audio clips, each ≈10 seconds. These clips are individually annotated with captions. Subsequently, an LLM is employed to generate question-and-answer pairs based on the captions of these clips. A subset of the data goes through expert review to construct **LongAudioBench**.

generate questions that ask for captions, challenge models to reason, or extract information from long audios. Fig. 6 outlines the LongAudio creation process. Fig.5 (middle) shows the distribution of these categories as outlined below:

1. **Captioning:** The task is to generate detailed and accurate descriptions of long audio, focusing on capturing the essence of the entire audio.

2. **Plot QA:** The task is to answer questions related to the overarching narrative or storyline of the audio, requiring an understanding of temporal and causal relationships between individual acoustic events.

3. **Temporal QA:** The task is to identify the temporal positions of acoustic events and their relationships, such as order or overlap within the audio, including how certain attributes change with time.

4. **Needle QA:** The task is to locate and reason about a specific "needle" segment embedded within a longer audio "haystack," ensuring the response is tied explicitly to the needle.

5. **Subscene QA:** The task is to answer questions about a specific subscene within the audio, requiring identification and understanding of localized events.

6. **General QA:** The task is to address broad, open-ended questions about the audio that may span multiple events, themes, or contexts, demonstrating overall comprehension and reasoning.

Fig.5 (right) shows the distribution of audio lengths, and Table 15 shows examples of each category.

**LongAudioBench:** We sample 10% (≈3k) from LongAudio, proportionally across QA types, for expert human verification. Annotators manually verify and correct these instances, discarding any incorrect ones. Subsequently, three authors conduct a quality check on the corrected samples. The full annotation process is detailed in Appendix E.7. The final version of LongAudioBench has 2,429 instances. For evaluation, following a wealth of prior work (Ghosh et al., 2024c; Yang et al., 2024b), we use an LLM-as-a-judge framework (Zheng et al., 2023): we prompt GPT-4o with the model response and ground truth answer using prompt 25 and score the response on a scale of 1 to 10.

## 5. Audio Flamingo 2 Training Strategy

We train AF2 using a 3-stage curriculum learning strategy, progressively increasing the audio context length and improving data quality: ***Stage 1: Pre-training.*** This stage focuses on multi-modal alignment to align audio representations with the LLM. For training, we leverage large-scale and plausibly noisy classification and captioning datasets. Inspired by (Cheng et al., 2024), we include a small amount of high-quality QA data to improve pre-training. During this stage, CLAP and LLM layers are frozen, and only the audio representation transformation and gated cross-attention layers are trainable. The audio context is restricted to $w = 3$ windows (30 seconds). ***Stage 2: Fine-tuning.*** This stage

Table 1: Comparison of Audio Flamingo 2 with previous SOTA LALMs on foundational audio understanding benchmarks. CAP = Captioning, AQA = Audio Question Answering, CLS = Classification, ZS=Zero-Shot. Also, see note in Section C.

| Dataset | Task | Previous SOTA | Ours | Dataset | Task | Previous SOTA | Ours |
|---|---|---|---|---|---|---|---|
| ClothoAQA$_{unanimous}$ | AQA - ACC | 74.9% - Qwen2 Audio | **86.9%**$_{+12.0\%}$ | Clotho-v2 | CAP - CIDEr | 0.45 - Qwen2 Audio | **0.46**$_{+0.01}$ |
| ClothoAQA$_{non-bin}$ | AQA - ACC | 49.5% - Audio Flamingo | **52.6%**$_{+3.1\%}$ | AudioCaps$^{ZS}$ | CAP - CIDEr | 0.46 - Audio Flamingo | **0.58**$_{+0.12}$ |
| MusicAVQA$_{audio}$ | AQA - ACC | 72.1% - Qwen Audio | **72.3%**$_{+0.2\%}$ | CREMA-D$^{ZS}$ | CLS - ACC | 26.5% - Audio Flamingo | **36.6%**$_{+10.1\%}$ |
| NonSpeech7k | CLS - ACC | 83.9% - Audio Flamingo | **84.3%**$_{+0.4\%}$ | Ravdess$^{ZS}$ | CLS - ACC | 20.9% - Audio Flamingo | **26.3%**$_{+5.4\%}$ |
| CochlScene | CLS - ACC | **91.6%** - Pengi | 82.1%$_{-9.5\%}$ | GTZAN$^{ZS}$ | CLS - ACC | 65.2% - Audio Flamingo | **69.1%**$_{+3.9\%}$ |
| NS$_{source}$ | CLS - ACC | 60.1% - Pengi | **62.0%**$_{+1.9\%}$ | Medley-solos-DB$^{ZS}$ | CLS - ACC | 85.6% - GAMA | **85.8%**$_{+0.2\%}$ |
| NS$_{instrument}$ | CLS - ACC | **78.8%** - Qwen Audio | 71.1%$_{-7.7\%}$ | US8K$^{ZS}$ | CLS - ACC | **71.2%** - Audio Flamingo | 68.0%$_{-3.2\%}$ |
| FSD50k | CLS - mAP | 47.9% - GAMA | **49.2%**$_{+1.3\%}$ | ESC50$^{ZS}$ | CLS - ACC | 80.6% - GAMA | **83.9%**$_{+3.3\%}$ |

focuses on improving audio understanding and reasoning by learning skills. For training, we use high-quality short-audio classification, captioning, and QA datasets. This stage is often referred to as *instruction-tuning* (Chu et al., 2024; Ghosh et al., 2024c). During this stage, only the LLM layers are frozen, and the CLAP model, audio representation transformation layers, and gated cross-attention layers are trainable. The audio context length is increased to $w = 9$ windows (1.5 minutes). **Stage 3: Long Fine-tuning.** This stage focuses on context-length extension and teaching skills specific to enabling reasoning on long audios. For training, we employ our proposed LongAudio dataset, keep the audio representation transformation and gated cross-attention layers trainable, and increase $w = 30$ windows (5 minutes).

# 6. Experiments

## 6.1. Experimental Setup

We train our model using 128 NVIDIA A100 80GB GPUs. During pre-training, we use an effective batch size of 1024, the AdamW optimizer (learning rate = $10^{-4}$, weight decay = 0.1), and bf16 with automatic mixed precision for efficiency. For fine-tuning and long fine-tuning, we adopt dynamic batching based on audio length, ensuring batch sizes are multiples of 2, with effective batch sizes ranging from 128 to 1024 (see Appendix H.2). We benchmark our model against recent SOTA LALMs, including GAMA, Audio Flamingo, Qwen-Audio, Qwen2-Audio, LTU, LTU-AS, SALMONN, AudioGPT, and Gemini (Flash v2 and Pro v1.5), GPT-4o-audio and report results for the best model. For zero-shot evaluations, we exclude corresponding datasets from all training stages, consistent with prior work (Gong et al., 2024; Ghosh et al., 2024c). For LongAudioBench, as current LALMs (except Gemini) do not support audio inputs of length ≥30 seconds, we adopt a two-step cascaded approach. First, we generate captions for short segments of the input audio, and then we prompt the same LALM to answer the question using these captions. Our experiments with passing the original long audio consistently outperformed our cascaded approach. For Gemini, we prompt it with the entire original long audio.

## 6.2. Smaller but Better

We employ standard benchmark datasets for evaluation (Table 24). Foundational benchmarks include ClothoAQA, MusicAVQA (audio-only), NonSpeech7k, NSynth, CREMA-D, Ravdess, GTZAN, Medley-solos-DB and USD8K and Clotho-v2 and AudioCaps. Reasoning benchmarks include MMAU (sound and music subsets), Audio Entailment, OpenAQA-test, MuchoMusic, CompA-R-*test*, MusicInstruct (Long subset), MusicQA, CMM (audio-language subset) and LongAudioBench.

**Foundational Audio Understanding:** Table 1 presents the performance of AF2 on foundational audio understanding benchmarks. For evaluation, we follow the similarity-based retrieval approach using our CLAP model, proposed by Deshmukh et al. (2023) and widely adopted. While foundational audio understanding is not our primary focus, AF2 shows competitive results against SOTA LALMs while having half of their size (e.g., Qwen(2)-Audio, GAMA, and LTU are equipped with 7B LLMs and large audio encoders).

**Expert Reasoning:** Table 2 compares the performance of AF2 on audio reasoning benchmarks. We employ the original evaluation strategy and metrics. With a much smaller LLM, AF2 outperforms all LALMs by large margins. We provide fine-grained results and failure cases of AF2 on LongAudioBench in Table 13 and Table 22.

Table 2: Comparison of Audio Flamingo 2 with previous SOTA LALMs on audio reasoning benchmarks, all AQA-based.

| Dataset | Previous SOTA | Ours |
|---|---|---|
| MMAU Sound | 61.7% - Gemini Flash v2 | **65.1%**$_{+3.4\%}$ |
| MMAU Music | 56.5% - Gemini Flash v2 | **72.9%**$_{+16.4\%}$ |
| AE Clotho | 83.3% - Qwen Audio | **92.5%**$_{+9.2\%}$ |
| AE AudioCaps | 64.2% - Qwen Audio | **93.3%**$_{+29.1\%}$ |
| CompA-R-*test* | 80.0% - GAMA-IT | **96.4%**$_{+16.4\%}$ |
| MuchoMusic | 51.4% - Qwen Audio | **56.5%**$_{+5.1\%}$ |
| OpenAQA | 80.0% - GAMA-IT | **86.0%**$_{+6.0\%}$ |
| MusicInstruct (Long) | 86.1% - MusiLingo | **90.2%**$_{+4.1\%}$ |
| MusicQA | 90.0% - MusiLingo | **93.0%**$_{+3.0\%}$ |
| CMM Hallucination | 76.0% - SALMONN | **82.0%**$_{+6.0\%}$ |
| LongAudioBench (*ours*) | 45.3% - Gemini Flash v2 | **64.2%**$_{+18.9\%}$ |

## 6.3. Enhanced Audio Features Boost Performance

Table 3 compares the performance of AF2 using AF-CLAP against various SOTA CLAP models on benchmark datasets. The MMAU score is averaged across the sound and music subsets. "AF-CLAP 630k" refers to AF-CLAP trained using the same strategy but with Laion-CLAP's 630k audio-text pairs. "AF-CLAP w/ con." represents our model trained with the same data but using the standard contrastive loss formulation. The results show that replacing AF-CLAP with other CLAP models results in a performance drop across benchmarks, highlighting the importance of robust audio representations for improving performance.

Table 3: Benchmark results on various CLAP as audio encoders.

| Model | AudioCaps | GTZAN | MuchoMusic | MMAU (avg) |
|---|---|---|---|---|
| Laion-CLAP | 0.51 | 65.2 | 52.3 | 63.8 |
| MS-CLAP | 0.55 | 65.8 | 53.1 | 64.3 |
| AF-CLAP 630k | 0.51 | 66.0 | 54.6 | 63.9 |
| AF-CLAP w/ con. | 0.53 | 66.2 | 54.4 | 66.8 |
| AF-CLAP | **0.58** | **69.1** | **56.5** | **69.0** |

## 6.4. High Quality Data Boosts Reasoning Abilities

Table 4 compares the impact of training data composition on performance. "w/ 1/2 data" and "w/ 1/3 data" refer to experiments using random subsets comprising half and one-third of the total instances from each dataset. Key findings include: (1) AudioSkills significantly enhances AF2's reasoning performance and shows notable improvements when combined with other datasets. (2) For challenging tasks like reasoning and zero-shot classification, data diversity is crucial, as using 1/2 of the data outperforms using only OpenAQA. (3) More data consistently improves performance.

Table 4: Comparison of training on different data compositions.

| Model | AudioCaps | GTZAN | MuchoMusic | MMAU (avg) |
|---|---|---|---|---|
| w/ 1/2 data | 0.48 | 51.3 | 46.7 | 49.3 |
| w/ 1/3 data | 0.41 | 47.7 | 43.9 | 45.1 |
| w/o AudioSkills | **0.58** | 68.8 | 42.6 | 48.6 |
| w/ OpenAQA | 0.55 | 19.3 | 39.8 | 38.2 |
| +AudioSkills | 0.55 | 22.6 | 51.3 | 57.5 |
| Original | **0.58** | **69.1** | **56.5** | **69.0** |

## 6.5. Cross-Attention Outperforms Prefix-Tuning

Table 5 compares GAMA, a SOTA 7B LALM using prefix tuning, trained on the same data and strategy (excluding stage 3), with AF2. AF2 significantly outperforms GAMA, attributed to superior audio representations and the shift cross-attention-based conditioning.

Table 5: Comparison of AF2 with GAMA, a SOTA LALM, trained on our same data and same training recipe.

| Model | AudioCaps | GTZAN | MuchoMusic | MMAU (avg) |
|---|---|---|---|---|
| GAMA (orig.) | 0.67 | 13.8 | 33.7 | 38.1 |
| GAMA (ours) | 0.41 | 54.7 | 40.3 | 52.8 |
| AF2 | **0.58** | **69.1** | **56.5** | **69.0** |

## 6.6. Effect of Scaling LLM

Fig. 7 illustrates the performance of AF2 across various LLM sizes, ranging from 0.5B to 7B, trained with and without AudioSkills. The results demonstrate that data quality often surpasses the performance gains achieved by simply scaling compute. Training with AudioSkills not only delivers superior performance overall but also significantly boosts reasoning capabilities, even at smaller LLM sizes. In contrast, when AudioSkills is excluded, reasoning performance heavily depends on model size, with performance scaling more gradually as parameters increase. This highlights the critical role of high-quality, skill-specific data like AudioSkills in driving robust reasoning capabilities, regardless of model size.

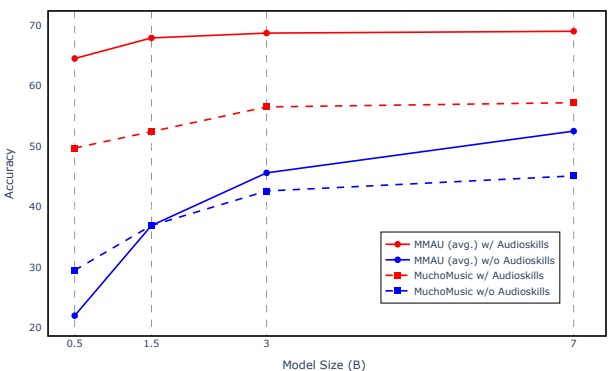

Figure 7: Performance comparison of AF2 on different LLM sizes, w/ and w/o AudioSkills. More results in Table 19.

Table 6: Overview of 10 training schedules detailing whether CLAP, XATTN, and LLM components (in order) are frozen or unfrozen during each stage and their impact on performance.

| | Training Stages | | | | Benchmarks | | | |
|---|---|---|---|---|---|---|---|---|
| | | | | | AudioCaps | GTZAN | MMAU | LongAudioB |
| 1 stage | ❄️🔥❄️ | - | - | - | 0.50 | 56.7 | 58.8 | 5.1 |
| | 🔥🔥🔥 | - | - | - | 0.50 | 52.5 | 56.3 | 4.9 |
| 2 stage | ❄️🔥❄️ | 🔥🔥❄️ | - | - | 0.56 | 68.9 | 68.0 | 5.1 |
| | ❄️🔥❄️ | ❄️🔥❄️ | - | - | 0.59 | 70.9 | 63.2 | 5.7 |
| | ❄️🔥❄️ | ❄️🔥❄️ | - | - | 0.59 | 72.3 | 60.7 | 5.7 |
| | 🔥🔥❄️ | 🔥🔥❄️ | - | - | 0.52 | 65.7 | 63.4 | 5.3 |
| 3 stage | ❄️🔥❄️ | 🔥🔥❄️ | ❄️🔥❄️ | - | 0.58 | 69.1 | **69.0** | **6.4** |
| | ❄️🔥❄️ | 🔥🔥❄️ | 🔥🔥🔥 | - | 0.45 | 62.6 | 59.1 | 6.2 |
| 4 stage | ❄️🔥❄️ | 🔥🔥❄️ | ❄️🔥🔥 | ❄️🔥❄️ | **0.62** | **74.2** | 64.5 | **6.4** |
| | ❄️🔥❄️ | 🔥🔥❄️ | ❄️🔥🔥 | ❄️🔥❄️ | 0.59 | 74.0 | 62.0 | 6.3 |

## 6.7. Effect of Training Schedules

Table 6 compares 10 training schedules and their impact on AF2 performance. For 1- and 2-stage training, data from later stages is combined with earlier stages. Additionally, we evaluate a 4-stage curriculum where stage 2 is repeated with LLM fine-tuning, and long fine-tuning is shifted to stage 4. Key findings include: (1) Fine-tuning the LLM improves classification and captioning tasks due to the style memorization effect (Ghosh et al., 2024a), which benefits

retrieval-based evaluation (see Appendix C) but reduces performance on reasoning tasks. (2) Pre-training alignment is essential before fine-tuning. (3) Gradual context-length extension is critical for effective long-audio understanding.

## 7. Conclusion, Limitations and Future Work

In this paper, we introduce Audio Flamingo 2, an ALM designed for long audio understanding and expert reasoning. Our model leverages a custom CLAP, trained on a large-scale dataset with a novel objective function, and is trained on synthetic reasoning AQA data to develop unique skills essential for real-world reasoning. Audio Flamingo 2 achieves SOTA performance in audio understanding and reasoning despite being small in model footprints. Additionally, we propose LongAudio and LongAudioBench to advance the field of ALM reasoning over long audio contexts.

For future work, we aim to address current limitations, including: (1) enhancing speech content understanding capabilities, (2) scaling AudioSkills to include more diverse datasets, and (3) developing audio encoders inherently capable of processing long audio.

## Impact Statement

Audio Flamingo 2 represents a transformative step in advancing ALMs by addressing the dual challenges of expert reasoning and long audio understanding. By integrating a small yet efficient architecture with novel synthetic datasets like AudioSkills and LongAudio, AF2 bridges critical gaps in reasoning and auditory comprehension, setting new state-of-the-art benchmarks. Beyond technical achievements, AF2 holds significant potential for real-world applications, including assistive technologies, anomaly detection, and multimedia analysis. By open-sourcing both the model and datasets, this work aims to democratize advancements in audio-language modeling, fostering innovation and collaboration across the research community.

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

# A. Table of Contents

# B. AF-CLAP

### B.1. Training Hyper-parameters

We train AF-CLAP on 8 A100 80GB GPUs. We train it with a learning rate of 5e-4, $M=N=3$, and an effective batch size of 256 for 12 epochs. This batch size is smaller than that in the literature, but we do so due to computational constraints.

### B.2. Feature Extraction Layer

As described in Section 3.2, unlike Audio Flamingo which uses CLAP audio embeddings directly, AF2 discards the CLAP head and rather leverages dense features from the final layer of the audio encoder (HTS-AT in our case). Table 8 compares the performance of AF2 trained with CLAP head features versus dense audio features from the encoder's last layer. The results clearly demonstrate that using dense features significantly improves AF2's performance.

### B.3. Training Datasets

Table 9 provides detailed statistics of datasets used for training AF-CLAP.

### B.4. Comparison with prior-art

Table 7 presents the performance of our CLAP on audio-to-text and text-to-audio retrieval tasks using the Clotho and AudioCaps datasets. "AF-CLAP 630k" refers to AF-CLAP trained using our proposed strategy but with Laion CLAP's 630k audio-text pairs. "AF-CLAP w/ van. con." represents AF-CLAP trained with the same data but using the standard contrastive loss formulation. "AF-CLAP w/o noise red." represents our AF-CLAP trained using the same data and method but without our noise reduction step. Our final proposed AF-CLAP achieves state-of-the-art performance across all metrics. Overall, AF-CLAP 630k does not lead to any improvements over Laion-CLAP, as our data augmentation and cleaning are only applicable to captions obtained

from complex real-world audios, which represents only a significantly small portion of Laion-CLAP.

Table 10 highlights the performance of AF-CLAP on zero-shot audio classification benchmarks, where it consistently achieves SOTA results. We make the same conclusion for Laion-CLAP as previously stated.

We emphasize that benchmark datasets do not holistically evaluate a CLAP model's capabilities, as highlighted in several works (Ghosh et al., 2024d; Wu et al., 2023a; Selvakumar et al., 2024; Ghosh et al., 2025). While AF-CLAP outperforms baselines on benchmark datasets, its audio features are significantly more robust and enhance the audio perception capabilities of (L)ALMs (see also Table 3).

# C. Limitations of CLAP Retrieval Based Evaluation

We attribute the suboptimal performance of AF2 on some datasets to the limitations of the CLAP-based evaluation method, which often fails to retrieve the correct label corresponding to the model's open-ended generation response. we follow the evaluation scheme introduced by Deshmukh et al. (2023) and widely adopted in prior works (Gong et al., 2024; Ghosh et al., 2024c). This approach uses a CLAP model to retrieve a label from the label set by comparing the model's open-ended generation and assigning the label with the highest similarity as the predicted output. We show some examples below where even a correct prediction by AF2 leads to incorrect retrieval and therefore lower accuracy:

> **Dataset:** GTZAN, **Correct Label:** Rock, **Predicted Label:** Punk Metal, **Retrieved Label:** Metal
>
> **Dataset:** ESC50, **Correct Label:** Pouring water, **Predicted Label:** liquid, , **Retrieved Label:** Water drops

However, fine-tuning the LLM leads to style memorization, which favors this evaluation method. We show some examples of prediction shift and how this leads to increase in accuracy:

> **Dataset:** GTZAN, **Correct Label:** rock, **Predicted Label:** rock, **Retrieved Label:** rock
>
> **Dataset:** ESC50, **Correct Label:** clock alarm, **Predicted Label:** clock alarm, **Retrieved Label:** clock alarm

# D. AudioSkills

### D.1. Dataset Statistics

Table 14 presents detailed category-wise statistics on our proposed AudioSkills dataset. We also list the meta-data used for data generation. For meta-data, transcripts were

Table 7: Performance comparison of AF-CLAP with baselines on Text-to-Audio and Audio-to-Text retrieval on AudioCaps and Clotho.

| | AudioCaps | | | | | | Clotho | | | | | |
|---|---|---|---|---|---|---|---|---|---|---|---|---|
| Model | Text-to-Audio | | | Audio-to-Text | | | Text-to-Audio | | | Audio-to-Text | | |
| | R@1 | R@5 | R@10 | R@1 | R@5 | R@10 | R@1 | R@5 | R@10 | R@1 | R@5 | R@10 |
| MMT | 36.1 | 72.0 | **84.5** | 39.6 | 76.8 | 86.7 | 6.7 | 21.6 | 33.2 | 7.0 | 22.7 | 34.6 |
| ML-ACT | 33.9 | 69.7 | 82.6 | 39.4 | 72.0 | 83.9 | 14.4 | 36.6 | 49.9 | 16.2 | 37.6 | 50.2 |
| CLAP | 34.6 | 70.2 | 82.0 | 41.9 | 41.9 | 84.6 | 16.7 | 41.1 | 54.1 | 20.0 | 44.9 | 58.7 |
| CompA-CLAP | 36.1 | 72.6 | 81.6 | 45.2 | 80.1 | 86.7 | 16.8 | 43.5 | 56.1 | 19.7 | 45.2 | 55.6 |
| Laion-CLAP | 36.1 | 71.8 | 83.9 | 46.8 | 82.9 | 90.7 | 16.1 | 38.3 | 51.1 | 22.7 | 48.5 | 60.8 |
| AF-CLAP-630k | 36.3 | 71.8 | 83.9 | 46.3 | 83.0 | 90.9 | 16.1 | 38.3 | 51.4 | 22.7 | 48.7 | 60.8 |
| AF-CLAP w/ van. con. | 36.8 | 72.0 | 84.0 | 45.2 | 83.5 | 90.0 | 16.5 | 41.9 | 55.3 | 22.9 | 51.0 | 63.2 |
| AF-CLAP w/o noise red. | 35.4 | 70.7 | 81.6 | 45.0 | 82.6 | 91.0 | 16.3 | 40.5 | 52.7 | 22.8 | **51.2** | 59.4 |
| AF-CLAP (ours) | **37.3** | **72.9** | 84.0 | **46.9** | **84.1** | **91.9** | **17.3** | **43.9** | **56.8** | **23.2** | **51.2** | **63.5** |

Table 8: Performance comparison of AF2 with different audio feature extraction methods from CLAP. Head refers to audio features extracted from the CLAP head, and Dense refers to dense features extracted from the last layer of the audio encoder.

| Model | AudioCaps | GTZAN | MuchoMusic | MMAU |
|---|---|---|---|---|
| Head $\mathbb{R}^{1\times256}$ | 0.50 | 54.7 | 42.6 | 47.0 |
| Dense $\mathbb{R}^{64\times2048}$ | **0.58** | **69.1** | **56.5** | **69.0** |

Table 9: Statistics of audio-caption datasets used for CLAP training. ∗ indicates the dataset was collected by us. Furthermore, all datasets with captions were made to go through our cleaning stage to reduce noise, as mentioned in Section 3.1.1.

| Dataset | #Audio-Text Pairs |
|---|---|
| YouTube-8M∗ (Abu-El-Haija et al., 2016) | 3,947,057 |
| Sound-VECaps (Yuan et al., 2024) | 1,657,029 |
| MiraData∗ (Ju et al., 2024) | 748,320 |
| Action2sound∗ (Chen et al., 2024b) | 306,602 |
| NSynth (Engel et al., 2017b) | 289,205 |
| Freesound (Font et al., 2013) | 256,695 |
| AudioSet Strong∗ (Hershey et al., 2021) | 216,622 |
| VGGSound (Chen et al., 2020) | 185,161 |
| FMA (Defferrard et al., 2016) | 106,412 |
| Video Recap∗ (Islam et al., 2024) | 64,627 |
| CochlScene (Jeong & Park, 2022) | 60,855 |
| FSD50k (Fonseca et al., 2022) | 40,966 |
| MACS (Morato & Mesaros, 2021) | 31,675 |
| Laion630k$_{BBCSoundEffects}$ (Wu et al., 2022b) | 31,201 |
| MagnaTagATune (Law et al., 2009) | 25,863 |
| SoundDescs (Koepke et al., 2022) | 23,085 |
| Clotho (Drossos et al., 2020) | 19,195 |
| TAU-Urban (Heittola et al.) | 14,400 |
| MusicCaps (Agostinelli et al., 2023) | 5,479 |
| WavText5K (Deshmukh et al., 2022) | 4,347 |
| SONICS (Rahman et al., 2024) | 1,602 |
| WavCaps-SoundBible(Mei et al., 2024) | 935 |
| MUSDB18 (Rafii et al., 2019) | 276 |
| MedleyDB-Pitch (Bittner et al., 2014) | 103 |
| **Total** | **8,037,712** |

obtained from Whisper $_{Large-v3}$ (Radford et al., 2023). We generate all audio captions from Qwen2-Audio and visual captions from Qwen2-VL.

# E. LongAudio

## E.1. Detailed Statistics

Table 11 presents detailed statistics of LongAudio and LongAudioBench, categorized into the various types of QAs.

## E.2. Examples

Table 15 shows category-wise examples from LongAudio.

## E.3. Comparison with other datasets

Table 12 compares the duration statistics of LongAudio with other AQA datasets. LongAudio stands out with the longest average audio durations.

## E.4. Fine-grained results for AF2

Table 13 presents category-wise fine-grained results of AF2 on LongAudioBench. While AF2 demonstrates strong performance, fine-tuning on LongAudio further improves scores across all categories, particularly in tasks unique to LongAudioBench, such as NeedleQA and SubsceneQA, which were not encountered during AF2's two-stage training. These results emphasize the significance of our proposed LongAudio dataset in effectively extending (L)ALM context length and improving long-audio reasoning.

## E.5. Success and Failure Cases on LongAudioBench

Table 22 presents success and failure cases of AF2 on LongAudioBench.

## E.6. Metadata

For MiraData, metadata includes visual captions (e.g., main object, background, and style) as detailed in the original paper and shown in prompt 9. For Video ReCap, metadata focuses on action captions describing activities, as shown in prompt 15.

Table 10: Performance comparison of our CLAP with baselines on Zero-shot Audio classification benchmarks.

| Model | ESC-50 | US8K | VGGSound | FSD50K | TUT | AudioSet | NSynth |
|---|---|---|---|---|---|---|---|
| Wav2CLIP | 41.4 | 40.4 | 10.0 | 3.0 | 28.2 | 5.0 | 5.9 |
| AudioClip | 69.4 | 65.3 | 9.9 | 6.6 | 29.5 | 3.7 | 6.8 |
| CLAP | 82.6 | 73.2 | 16.4 | 14.0 | 29.6 | 5.1 | 9.9 |
| Laion-CLAP | 88.2 | 74.1 | 21.2 | 22.4 | 58.4 | 20.8 | 11.8 |
| CoLLAT | 84.0 | 77.0 | - | 19.0 | 29.0 | 9.0 | - |
| CompA-CLAP | 86.5 | 88.1 | 21.9 | 19.6 | 56.7 | 21.6 | 11.8 |
| AF-CLAP-630k | 88.1 | 74.3 | 21.2 | 22.0 | 57.5 | 19.3 | 12.0 |
| AF-CLAP w/ van. con. | 91.0 | _91.8_ | 23.5 | **28.1** | _63.0_ | 21.8 | **17.4** |
| AF-CLAP w/o noise red. | _91.1_ | 91.3 | _23.9_ | 26.9 | **63.2** | _22.8_ | 15.5 |
| AF-CLAP *(ours)* | **91.3** | **92.3** | **24.1** | _27.2_ | **63.2** | **23.7** | _15.9_ |

Table 11: Dataset statistics for LongAudio and LongAudioBench.

| Category | LongAudio | LongAudioBench |
|---|---|---|
| Captioning | 67,498 | 917 |
| Plot QA | 61,511 | 237 |
| Complex QA | 50,315 | 361 |
| Subscene QA | 7,908 | 176 |
| Temporal QA | 66,836 | 253 |
| General QA | 8,860 | 485 |
| **Total** | 262,928 | 2,429 |

Table 12: Comparison of duration statistics of LongAudio with other AQA datasets. LongAudio stands out with the longest average audio durations.

| Method | Min | Max | Avg |
|---|---|---|---|
| Clotho-AQA (Lipping et al., 2022) | 15.00 | 30.00 | 22.53 |
| AudioEntailment (Deshmukh et al., 2024) | 15.00 | 30.00 | 22.49 |
| CompA-R (Ghosh et al., 2024c) | 0.47 | 10.01 | 9.87 |
| OpenAQA (Gong et al., 2024) | 0.06 | 180.00 | 12.01 |
| MU-LLAMA (Liu et al., 2024) | **29.12** | 29.12 | 29.12 |
| Salmonn (Tang et al., 2024) | 0.47 | 1069.04 | 10.62 |
| LongAudio *(ours)* | 5.00 | **1797.71** | **117.08** |

Table 13: Fine-grained scores of AF2 on LongAudioBench w/ and w/o fine-tuning on our dataset LongAudio.

| Category | AF2 | AF2 w/o LongAudio |
|---|---|---|
| Captioning | 63.75% | 46.02% |
| Plot QA | 68.02% | 44.15% |
| Temporal QA | 62.61% | 51.00% |
| Needle QA | 63.13% | 35.92% |
| Subscene QA | 64.03% | 33.61% |
| General QA | 63.61% | 42.39% |
| Avg | 64.19% | 50.22% |

### E.7. Human Verification for LongAudioBench

The human verification process has been approved by our institution's Institutional Review Board (IRB).

**Annotation Method.** The annotation process for LongAudioBench was carried out in two stages by a group of experts. In the first stage, experts corrected and verified each QA pair, including discarding completely erroneous pairs. In the second stage, experts reviewed QA pairs corrected by

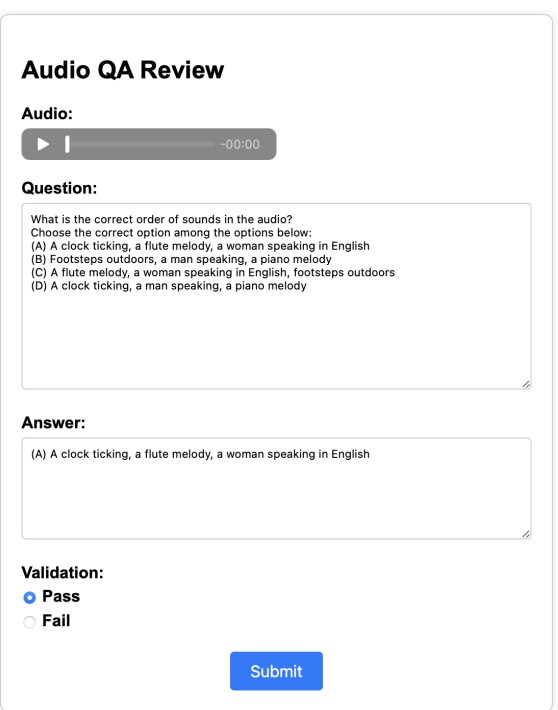

Figure 8: Snapshot of the annotation tool used for LongAudio annotation.

each other. Thus, each QA pair was at least annotated by 2 individuals. Their primary role in both stages was to annotate the data while ensuring accuracy, consistency, and adherence to the predefined guidelines (mentioned next). We build an annotation tool for this purpose, and Fig. 8 provides a snapshot of the tool. This multi-stage process, involving annotation followed by a detailed review, was designed to enhance the reliability and depth of the annotated dataset, leveraging the combined expertise and experience of the annotators.

**Annotation Guidelines.** The authors of the paper put forward the following annotation guidelines for annotating and verifying LongAudioBench:

• **Accuracy and Consistency:** Ensure all annotations

Table 14: Detailed statistics of AudioSkills, categorized into individual reasoning types, together with details on open-source datasets, additional meta-data, and prompt used for QA generation.

| Reasoning Type | Size | Datasets Used | Meta-Data Used | Prompt Reference |
|---|---|---|---|---|
| Temporal | 188,799 | AudioSet-SL, MusicBench | Time Stamped Events (GT), Caption | Fig. 9, 10 |
| Attribute Identification | 201,455 | AudioSet-SL | Attribute values from Kumar et al. (2024), Caption | Fig. 26 |
| Counting | 50,493 | Synthetic | Transcript, Caption | pythonic |
| Contextual Sound Event Reasoning | 982,847 | AudioSet-SL, MusicBench, FMA | Time Stamped Events, Caption | Fig. 13,27, 28, 29 |
| Contextual Speech Event Reasoning | 1,272,782 | AudioSet-SL | Transcript, Caption | Fig. 30 |
| Information Extraction | 858,639 | AudioSet-SL, MusicBench, MTG-Jamendo, MusicNet | Time Stamped Events, Caption | Fig. 31, 32, 33, 34, 37, 36 |
| General Reasoning | 704,040 | AudioSet-SL, MusicBench, MusicNet, FMA | Time Stamped Events, Caption | Fig. 27, 33, 34, 35, 36 |

are accurate and consistent, strictly adhering to the predefined standards and guidelines.

- **Two-Stage Process:** Ensure all QAs go through the 2-stage verification process mentioned above.

- **Listening Requirements:** Listen to the complete audio before annotating to ensure the QA pair is contextually accurate.

- **QA Pair Validation:** *i)* Ensure the audio in the QA pair is valid, not corrupt, and corresponds to the question. *ii)* Discard or flag QA pairs containing irrelevant or ambiguous content.

- **Question Format:** *i)* All questions must be in English. *ii)* Check if the tagged QA type (e.g., captioning, PlotQA) is correct and relevant. *iii)* Avoid mentioning identifiable information about the audio, such as names or metadata.

- **Answer Format:** *i)* All answers must follow the MCQ or open-ended format, pre-defined for each question category. *ii)* Additionally, annotators should ensure answers are comprehensive and free of ambiguities.

- **Tool Usage:** *i)* Use the annotation tool (Fig. 8) for all corrections, reviews, and annotations. *ii)* Document and report any technical issues encountered during the process.

- **Quality Assurance:** *i)* Each QA pair must be reviewed by at least two individuals to ensure reliability and adherence to the guidelines. *ii)* Maintain detailed logs of discarded QA pairs and reasons for exclusion. *iii)* Escalate ambiguous or complex QA pairs to the team for collective review.

- **Collaboration:** *i)* Maintain open communication with other annotators to clarify task-specific doubts or resolve disagreements during review. *ii)* Share feedback on ambiguous QA pairs to iteratively improve the annotation process.

## F. Prompts

### F.1. Prompts for foundational datasets

Tables 26 and 27 list the prompts used to convert foundational audio classification and captioning datasets into the AQA format for training AF2. For the answers, only the label was retained, excluding any additional text. Initially, we experimented with adding a prefix to the labels to convert them into sentence format, but this approach did not yield favorable results.

### F.2. Prompts for other tasks

Table. 26 and 26 list prompts we used to convert foundational audio classification and captioning datasets into AQA format for training AF2.

1. Prompt 9: Prompt used for generating **Temporal QA** for LongAudio from MiraData.

2. Prompt 11: Prompt used for generating **Plot QA** for LongAudio from MiraData.

3. Prompt 12: Prompt used for generating **Caption QA** for LongAudio from MiraData.

4. Prompt 13: Prompt used for generating **Contextual Sound Event Reasoning QA** for AudioSkills.

5. Prompt 14: Prompt used for generating **Caption QA** for Video-ReCap.

6. Prompt 15: Prompt used for generating **Subscene QA** for LongAudio from Video-ReCap.

7. Prompt 16: Prompt used for generating **General QA** for LongAudio from Video-ReCap.

8. Prompt 17: Prompt used for generating **Subscene QA** for LongAudio from MiraData.

9. Prompt 18: Prompt used for generating **Temporal Reasoning QA** for AudioSkills.

10. Prompt 19: Prompt used for generating **linguistically variant positives** for short-audio captions for CLAP training.

11. Prompt 20: Prompt used for cleaning and removing noise from synthetic short-audio captions.

12. Prompt 21: Prompt used for generating **compositionally different negatives** for short-audio captions for CLAP training.

13. Prompt 22: Prompt used for generating **synthetic short-audio captions** from Video ReCAP.

14. Prompt 23: Prompt used for generating **synthetic short-audio captions** from MiraData.

Table 15: Category-wise examples from **LongBench**.

| Category | Question | Answer |
|---|---|---|
| Captioning | How would you describe the scene from the audio? | Two women converse in a car with ambient bird chirping and occasional mechanical noises in the background. |
| | What is taking place in the audio? | The audio features ambient wind, distant motor vehicle hums, and occasional background noises creating an eerie atmosphere. |
| Plot QA | What is the predominant mood conveyed by the background music and sound effects in the audio segments? | Sad and suspenseful. |
| | What is the primary theme conveyed by the audio in the initial segments? | The audio conveys a solemn and spiritual theme with classical music and religious references. |
| Temporal QA | What is the correct order of sounds in the audio? Choose the correct option among the options below:
(A) A clock ticking, a flute melody, a woman speaking in English
(B) Footsteps outdoors, a man speaking, a piano melody
(C) A flute melody, a woman speaking in English, footsteps outdoors
(D) A clock ticking, a man speaking, a piano melody | (A) A clock ticking, a flute melody, a woman speaking in English |
| | When can the sound of a horse galloping be heard? Choose the correct option among the options below:
(A) At the beginning
(B) In the middle
(C) Towards the end | (B) In the middle |
| Needle QA | What audio event indicates an interruption during the conversation between the man and woman? | The background sound of a phone ringing from 0.53 seconds to 7.40 seconds. |
| | What audio element creates a sense of suspense in the dark room scene? | The cinematic strings playing a scary tune along with a church bell and distant horror scream. |
| Subscene QA | What happened between the sound of ocean waves and the sound of mechanisms functioning? | The sound of a phone being dropped on a table followed by mechanical fan noises. |
| | What sounds occur between brushing teeth and using a phone? | Music playing continuously with occasional running water and a shower sound. |
| General QA | What continuous background noise is heard while the person is using the laptop? | The sound of an engine idling consistently. |
| | What was the continuous background noise throughout the audio? | Engine idling. |

15. Prompt 24: Prompt used for rewriting audio captions and removing implausible acoustic elements.

16. Prompt 25: Prompt used for evaluating responses for questions in LongAudioBench.

17. Prompt 26: Prompt used for generating **Attribute QA** for AudioSkills.

18. Prompt 27: Prompt used for generating **Complex QA** for AudioSkills.

19. Prompt 35: Prompt used for generating **QA** for AudioSkills using FMA.

20. Prompt 36: Prompt used for generating **QA** for AudioSkills using MusicNet.

21. Prompt 37: Prompt used for generating **QA** for AudioSkills using MTG-Jamendo.

22. Prompt 38: Prompt used for **self-verification** of generated QA pairs for MiraData.

23. Prompt 39: Prompt used for **self-verification** of generated QA pairs for ReCap dataset.

# G. More Results and Design Analysis

## G.1. Effect of RoPE

Table 16 compares the performance of AF2 with and without applying RoPE in the audio transformation layers. We see a drop in performance in LongAudioBench and MMAU without the application of RoPE, which highlights its importance in long-context and reasoning capabilities.

Table 16: Performance comparison of AF2 with and without RoPE in the audio transformation layer.

| Frequency | AudioCaps | GTZAN | LongAudioB | MMAU (avg) |
|---|---|---|---|---|
| w/o RoPE | 0.56 | 67.2 | 4.3 | 59.1 |
| w/ RoPE | **0.58** | **69.1** | **6.4** | **69.0** |

## G.2. Effect of Audio Transformation Layers

Table 17 compares the performance of AF2 with and without the audio transformation layers. We see a drop in performance without the audio transformation layers, which highlights its importance in expanding audio representation learning and adaptation capacity.

Table 17: Performance comparison of AF2 with and without the audio transformation layer.

| Frequency | AudioCaps | GTZAN | MuchoMusic | MMAU (avg) |
|---|---|---|---|---|
| w/o transform | 0.55 | 64.3 | 51.8 | 59.0 |
| w/ transform | **0.58** | **69.1** | **56.5** | **69.0** |

## G.3. Effect of Cross-Attention Frequency

Table 18 compares AF2's performance across different cross-attention conditioning frequencies. As a reminder, conditioning the audio representations using cross-attention after every LM layer (frequency of 1) yields the best performance. However, conditioning every 3rd layer performs competitively, with a noticeable performance drop when conditioning is reduced to every 6th layer.

Table 18: Comparison of performance with various cross-attention conditioning frequencies.

| Frequency | AudioCaps | GTZAN | MuchoMusic | MMAU (avg) |
|---|---|---|---|---|
| 6 | 0.50 | 56.7 | 50.1 | 61.7 |
| 3 | 0.58 | 70.0 | 54.4 | 65.7 |
| 1 | **0.58** | **69.1** | **56.5** | **69.0** |

## G.4. Results on different LLM Sizes

Table 19 compares the performance of AF2 on various LLM sizes, ranging from 0.5B to 7B.

Table 19: Results for AF2 on different LLM sizes, ranging from 0.5B - 7B.

| Dataset | AF2-0.5B | AF2-1.5B | AF2-3B | AF2-7B |
|---|---|---|---|---|
| AudioCaps | 0.46 | 0.50 | 0.58 | 0.58 |
| GTZAN | 65.6 | 66.1 | 69.1 | 68.8 |
| ClothoAQA | 80.4 | 82.7 | 86.9 | 85.1 |
| MMAU Sound | 61.0 | 65.0 | 64.4 | 65.0 |
| MMAU Music | 68.0 | 70.9 | 72.9 | 73.1 |
| MuchoMusic | 49.7 | 52.4 | 56.5 | 57.2 |
| CompA-R-*test* | 82.4 | 92.4 | 96.4 | 95.1 |

# H. More Details

## H.1. Topic-wise Video Segmentation

To segment videos into topic-specific clusters, we adopted a multi-step clustering approach utilizing both textual of video captions. Below, we detail the methodology:

**1. Feature Extraction:** We extracted semantic feature embeddings from video captions to represent each video. To achieve this, we employed NV-Embed (Lee et al., 2024).

**2. Clustering:** Next, we applied K-Means Clustering to group videos based on their feature representations.

**3. Cluster Selection and Diversity:** After clustering, we analyzed the clusters to ensure relevance and diversity. For each cluster, a representative subset of videos was selected using a combination of random sampling and Determinantal Point Processes (DPPs) (Kulesza et al., 2012), a diversity-based scoring method. This ensured that selected videos captured the range of topics present in each cluster while avoiding redundancy.

**4. Manual Review and Refinement:** To further ensure the quality and relevance of the selected videos, we conducted a manual review of a random sample from each cluster. Videos with ambiguous or low-quality content were discarded or reassigned as needed.

## H.2. Data Loader

AF2 is trained on audio datasets with durations ranging from 0.5 seconds to 10 minutes. Using a generic data loader would result in excessive padding within batches, leading to unstable losses. To address this, we implement a dynamic batching scheme that groups audio samples of similar lengths, minimizing padding. This approach is further constrained by a predefined maximum duration for each training job. The algorithm is detailed in Algorithm 1 and 2, with its two main components explained below:

**Weighted Bucketed Blending.** We begin with several datasets, each assigned a specific weight that determines the relative number of items sampled from the dataset per epoch. For each dataset, audio clips are divided into buckets based on their duration—e.g., short clips are grouped in one bucket, medium-length clips in another, and so on. This bucketing process minimizes padding overhead by ensuring that only audio clips of similar lengths are grouped together.

During training epoch $e$, the number of items to sample from each bucket is determined by the following indices:

$$\text{start\_idx} = \lfloor e \times \text{bucket\_size} \times \text{weight} \rfloor \% \text{bucket\_size},$$

$$\text{end\_idx} = \lfloor (e+1) \times \text{bucket\_size} \times \text{weight} \rfloor \% \text{bucket\_size}.$$

These indices define a "slice" of the bucket for the current epoch, ensuring that each dataset and bucket contributes a controlled number of items. If the index $i$ completes an entire bucket (i.e., when $i \mod \text{bucket\_total} = 0$), the bucket is shuffled deterministically using a seed derived from the dataset name and the current epoch. This approach maintains randomness across epochs while ensuring reproducibility.

**Dynamic Batching.** After blending all datasets and gathering their examples into a single list, we apply *Dynamic Batching* to form more efficient mini-batches. The primary goal is to group audio clips of similar lengths to minimize the total number of padded frames per mini-batch. Each audio example's token count (e.g., frame count or another relevant measure of length) is used to incrementally add examples to a batch, subject to the following constraints:

- **max_sentences**: The maximum number of examples allowed in a batch.

- **max_tokens**: The maximum total frames/tokens permitted in a batch, calculated as batch_size × max_audio_length.

- **bsz_mult**: An optional alignment constraint requiring that the batch size either be smaller than this multiplier or a multiple of it, often for multi-GPU efficiency.

At each step, we compute the total frames/tokens in the tentative new batch, considering the maximum frame length so far. If adding the next audio clip violates either **max_tokens** or **max_sentences**, the current batch is finalized, and a new batch is started. Additionally, if the batch size is a multiple of **bsz_mult**, we consider finalizing the batch early to meet this alignment requirement. Any remaining examples at the end of the data list form the final batch.

This two-stage approach—first weighted bucketed blending, followed by dynamic batching—ensures:

1. Multiple datasets contribute examples proportionally to their desired weights.

2. Effective shuffling to prevent overfitting.

3. Efficient grouping of audio clips by length, reducing wasted padding and speeding up training.

**Effectiveness.** Our dynamic batching scheme reduces the percentage of paddings in each batch from 58% to 16%.

## H.3. Computational Analysis

**Training Cost.** We trained Audio Flamingo 2 on 128 NVIDIA A100 80GB GPUs for each stage and variant. For the original version with the 3B model, stage 1 takes about 4 days to train, stage 2 about 5 days, and stage 3 about 12 hours. We provide detailed cost analysis in Table 20.

Table 20: Final model training cost comparison of Audio Flamingo 2. Costs are estimated using AWS (e.g., https://aws.amazon.com/ec2/instance-types/p4/), and excludes LM training costs and other failures. The cost of experimentation is about 5× the final model training cost.

| LLM Size | Total Params | GPU Hours | Estimated Cost |
|---|---|---|---|
| 0.5B | 1.0B | 104 | ≈USD 55K |
| 1.5B | 2.5B | 160 | ≈USD 84K |
| 3B | 4.7B | 228 | ≈USD 120K |
| 7B | 11.0B | 510 | ≈USD 268K |

Table 21: List of fine pre-training and fine-tuning datasets together with their training composition.

| Dataset | Audio Length | #Audio-Text Pairs | Pretraining Epochs | SFT Epochs |
|---|---|---|---|---|
| AudioSkills | - | 4200K | - | 2.0 |
| CompA-R | 159 hrs | 350k | - | 2.0 |
| MusicBench | 115.5 hrs | 686k | - | 2.0 |
| Mu-LLAMA | 62.9 hrs | 70k | - | 2.0 |
| Salmonn AQA | 800 hrs | 270k | - | 1.0 |
| ClothoAQA | 7.4 hrs | 9.7K | - | 1.0 |
| OpenAQA | 693.2 hrs | 1959.8K | - | 1.0 |
| Clotho-v2 | 24.0 hrs | 19.2K | - | 1.0 |
| MACS | 10.9 hrs | 17.3K | - | 1.0 |
| FSD50k | 80.8 hrs | 41.0K | - | 1.0 |
| CochlScene | 169.0 hrs | 60.9K | - | 1.0 |
| NonSpeech 7k | 6.2 hrs | 6.3K | - | 1.0 |
| Chime-home | 5.0 hrs | 4.5K | - | 1.0 |
| Sonyc-UST | 34.9 hrs | 27.9K | - | 1.0 |
| Emov-DB | 7.8 hrs | 6.8K | - | 1.0 |
| JL-Corpus | 1.4 hrs | 2.4K | - | 6.0 |
| Tess | 1.6 hrs | 2.8K | - | 2.5 |
| OMGEmotion | 3.0 hrs | 1.7K | - | 3.0 |
| MusicAVQA$_{\text{audio-only}}$ | 77.1 hrs | 5.7K | - | 3.0 |
| MusicQA | 62.9 hrs | 70K | - | 3.0 |
| LP-MusicCaps$_{\text{MSD}}$ | 5805.7 hrs | 1331.8K | - | 0.026 |
| LP-MusicCaps$_{\text{MTT}}$ | 126.4 hrs | 46.9K | - | 1.0 |
| LP-MusicCaps$_{\text{MC}}$ | 7.4 hrs | 7.9K | - | 2.0 |
| MusicCaps | 7.4 hrs | 2.6K | - | 6.0 |
| NSynth | 321.3 hrs | 289.2K | - | 1.5 |
| MTG-Jamendo | 3768.9 hrs | 55.6K | - | 1.0 |
| MusDB-HQ | 29.1 hrs | 10.2K | - | 2.0 |
| FMA | 860.7 hrs | 104.2K | - | 1.0 |
| OpenAQA | 693.2 hrs | 1959.8K | 1.0 | 1.0 |
| Laion630k$_{\text{BBCSoundEffects}}$ | 456.9 hrs | 15.1K | 5.0 | - |
| Laion630k$_{\text{Freesound}}$ | 2494.8 hrs | 306.5K | 1.0 | - |
| SoundDescs | 749.7 hrs | 23.1K | 1.0 | - |
| WavCaps | 3793.3 hrs | 402.6 K | 1.0 | - |
| AudioSet | 2617.8 hrs | 950.8K | 1.0 | - |
| WavText5K | 23.8 hrs | 4.3K | 3.0 | - |
| MSP-Podcast | 73.9 hrs | 45.1K | 3.0 | - |
| MELD | 8.7 hrs | 32.9K | 3.0 | - |
| MusicAVQA$_{\text{audio-visual}}$ | 142.4 hrs | 17.9K | 3.0 | - |
| MusicQA | 62.9 hrs | 70K | 1.2 | - |
| LP-MusicCaps$_{\text{MSD}}$ | 5805.7 hrs | 1331.8K | - | 1.0 |
| MTG-Jamendo | 3768.9 hrs | 55.6K | 1.0 | - |

Table 22: Examples of Highest and Lowest Scores for Different Reasoning Types

| Reasoning Type | Prompt | Output | Model Response | Score |
|---|---|---|---|---|
| Captioning | Give a description of the atmosphere in the audio. | Continuous background noise with occasional sounds of objects being moved and water running in an apartment. | Continuous background noise with occasional sounds of water running and objects being moved. | 10 |
| | Give a general description of the audio content. | Background noise and occasional speech are heard as someone moves around a house, decorates, and watches television. | A man speaks in English with a neutral tone, accompanied by background noise and occasional mechanical sounds. | 1 |
| Temporal QA | What can be heard at the beginning of the audio? Choose the correct option among the options below: (A) A man speaking in English, (B) Classical music with drums, (C) Instrumental electronic piece, (D) A pop song. | (A) A man speaking in English | (A) A man speaking in English | 10 |
| | What indicates a transition to an intense and momentous scene during the group of men riding horses in the desert? | The wide brass melody over a powerful string section creates a cinematic and epic atmosphere, suggesting an important moment. | The sound of a train horn honking followed by a train passing by. | 1 |
| Subscene QA | What do you hear between the piano playing and the guitar strumming? | I hear a male voice speaking in English with a sad mood. | A male voice speaking in English with a sad mood. | 10 |
| | What is happening between the sound of game music and the idling engine? | The scene consists of a person playing with the remote control with background whirring and clicking sounds. | A man is speaking in English. | 1 |
| Needle QA | What sudden auditory change occurs that might indicate a shift in the person's physical state while navigating the snow? | A significant increase in breathing and panting sounds, suggesting physical exertion or difficulty. | The sound of breathing becomes prominent, suggesting the person may be exerting effort or experiencing discomfort. | 9 |
| | What does the woman mention about the view from the bedroom? | She mentions seeing the red and yellow cliffs from her bedroom in the morning. | She mentions that the view from the bedroom is amazing. | 2 |
| Plot QA | What is the mood conveyed by the male voice singing in the audio? | Sad. | The male voice singing in the audio conveys a deep and resonant mood. | 10 |
| | What innovative feature does the man highlight about the Pulse smart watch? | It can DJ without using a phone. | The man highlights the smartwatch's ability to monitor heart rate and sleep patterns. | 1 |
| General QA | What natural sound is consistently present throughout the entire audio? | Birds chirping lightly. | Birds chirping lightly. | 10 |
| | What sound is heard consistently throughout the audio clip? | Continuous background noise. | The sound of a rooster crowing. | 1 |

---

**Algorithm 1** Weighted Bucketed Blending & Dynamic Batching

---

**Require:** all_data: A dictionary of datasets, each with contents (bucketed examples) and weight $> 0$
**Require:** epoch: Current epoch index
**Require:** IsBrokenFile($\cdot$): Checks if an audio file is unreadable
**Require:** ShuffleDictFixedRand($\cdot$, seed): Shuffles the bucketed data deterministically
**Require:** batch_by_size_fn($\cdot$): Function implementing dynamic batching by size (Algorithm 2)
  **function** BLENDDATA(all_data, epoch)
      self_data $\leftarrow$ {data : {}, total_num : 0}
      **for** dataset_name $\in$ all_data **do**
          contents $\leftarrow$ all_data[dataset_name].contents
          weight $\leftarrow$ all_data[dataset_name].weight
          **for** (bucket_idx, bucket_data) $\in$ contents **do**
              bucket_total $\leftarrow$ |bucket_data|
              **if** bucket_total $= 0$ **then**
                 **continue**                                   ▷ Skip empty buckets
              **end if**
              start_idx $\leftarrow \lfloor$epoch $\times$ bucket_total $\times$ weight$\rfloor \bmod$ bucket_total
              end_idx $\leftarrow \lfloor$(epoch $+ 1$) $\times$ bucket_total $\times$ weight$\rfloor \bmod$ bucket_total
              **for** idx $\leftarrow$ start_idx to end_idx $- 1$ **do**
                 **if** (idx $> 0$) $\wedge$ (idx $\bmod$ bucket_total $= 0$) **then**
                    seed $\leftarrow$ SumOfCharCodes(dataset_name $\|$ "epoch" $\| \lfloor$idx/bucket_total$\rfloor$)
                    bucket_data $\leftarrow$ ShuffleDictFixedRand(bucket_data, seed)
                 **end if**
                 key $\leftarrow$ idx $\bmod$ bucket_total
                 item $\leftarrow$ bucket_data[key]
                 **if** ISBROKENFILE(item["name"]) **then**
                    **continue**                        ▷ Skip broken files
                 **end if**
                 self_data["data"][self_data["total_num"]] $\leftarrow$ item
                 self_data["total_num"] $\leftarrow$ self_data["total_num"] $+ 1$
              **end for**
          **end for**
      **end for**
      **return** self_data
  **end function**
  **function** DYNAMICBATCHING(indices, num_tokens_fn, max_tokens, max_sentences, bsz_mult) ▷ High-level wrapper for dynamic batching
      batches $\leftarrow$ batch_by_size_fn(indices, num_tokens_fn, max_tokens, max_sentences, bsz_mult)
      **return** batches
  **end function**

---

---

**Algorithm 2** Dynamic Batching by Size (Pseudocode for batch_by_size_fn)

---

**Require:** indices: Array of example indices
**Require:** numTokensVec[$i$]: Number of tokens (or frames) for indices[$i$]
**Require:** max_tokens, max_sentences: Batch constraints
**Require:** bsz_mult: Required multiple of batch size (e.g., for multi-GPU alignment)
  **function** BATCH_BY_SIZE_VEC(indices, numTokensVec, max_tokens, max_sentences, bsz_mult)
      batches ← [ ]                              ▷ List of subarrays of indices
      batchStart ← 0
      tailMaxTokens ← 0
      batchMaxTokens ← 0
      **for** pos ← 0 to |indices| − 1 **do**
          tailMaxTokens ← max(tailMaxTokens, numTokensVec[pos])
          newBatchEnd ← pos + 1
          newBatchMaxTokens ← max(batchMaxTokens, tailMaxTokens)
          newBatchSentences ← newBatchEnd − batchStart
          newBatchNumTokens ← newBatchSentences × newBatchMaxTokens
          overflow ← (newBatchSentences > max_sentences > 0) ∨ (newBatchNumTokens > max_tokens > 0)
          sizeOk ← (newBatchSentences < bsz_mult) ∨ (newBatchSentences mod bsz_mult = 0)
          **if** overflow **then**                     ▷ Finalize the current batch and start a new one
              batches.APPEND(indices[batchStart : pos])
              batchStart ← pos
              batchMaxTokens ← numTokensVec[pos]
              tailMaxTokens ← numTokensVec[pos]
          **else if** sizeOk **then**              ▷ Optionally finalize if batch size is a multiple of bsz_mult
              batches.APPEND(indices[batchStart : newBatchEnd])
              batchStart ← newBatchEnd
              batchMaxTokens ← 0
              tailMaxTokens ← 0
          **else**                         ▷ Continue accumulating examples in the current batch
              batchMaxTokens ← newBatchMaxTokens
          **end if**
      **end for**
                                     ▷ Add any leftover examples in the final batch
      **if** batchStart < |indices| **then**
          batches.APPEND(indices[batchStart :])
      **end if**
      **return** batches
  **end function**

---

Table 23: All datasets used to train our model. We mark the datasets used in Stage 1[1], Stage 2[2], Stage 3[3] or multiple stages of training[1,2,3]. Datasets marked with * were added to AF2 over the ones already present in Audio Flamingo.

| Audio Type | Task | Datasets | #Audio-Text Pairs |
|---|---|---|---|
| General Sound | CAP | WavCaps[1] (Mei et al., 2024), Macs[2] (Martin Morato & Mesaros, 2021), SoundDescs[1] (Oncescu et al., 2021), Clotho-v2[2] (Drossos et al., 2020), WavText5K[1] (Deshmukh et al., 2022), Laion-630k[1] (Wu et al., 2023b) | ∼829 K |
| | AQA | Clotho-AQA[2] (Lipping et al., 2022), Open-AQA[2] (Gong et al., 2023a), Salmonn AQA[2*] (Tang et al., 2023), AudioEntailment[2*] (Deshmukh et al., 2024) CompA-R[2*] (Ghosh et al., 2024c), AudioSkills[1,2*] (*ours*), LongAudio[3*] (*ours*) | ∼1970 K |
| | CLS | AudioSet[2] (Gemmeke et al., 2017), FSD50k[2] (Fonseca et al., 2021), CochlScene[2] (Jeong & Park, 2022), NonSpeech7K[2] (Rashid et al., 2023), Chime-Home[2] (Foster et al., 2015), Sonyc-UST[2] (Cartwright et al., 2019) | ∼1091 K |
| Music | CAP | LP-MusicCaps[2] (Doh et al., 2023), MusicCaps[2] (Agostinelli et al., 2023) | ∼1389 K |
| | AQA | MusicQA[2] (Liu et al., 2024), MusicAVQA[2] (Li et al., 2022) MusicBench[2*] (Melechovsky et al., 2023), Mu-LLAMA[2*] (Liu et al., 2024) AudioSkills[1,2*] (*ours*), LongAudio[3*] (*ours*) | ∼94 K |
| | CLS | NSynth[2] (Engel et al., 2017b), MTG-Jamendo[2] (Bogdanov et al., 2019), FMA[2] (Defferrard et al., 2016), MusDB-HQ[2] (Rafii et al., 2019), | ∼459 K |
| Speech | CLS | MSP-Podcast[1] (Lotfian & Busso, 2017), Emov-DB[2] (Adigwe et al., 2018) JL-Corpus[2] (James et al., 2018), Tess[2] (Pichora-Fuller & Dupuis, 2020), MELD[2] (Poria et al., 2018), OMGEmotion[2] (Barros et al., 2018) | ∼92 K |

Table 24: All datasets used for evaluation in AF2, with additional usage/contextual notes for foundational datasets.

| Dataset Type | Audio Type | Task | Datasets |
|---|---|---|---|
| Foundational | Sound | CAP | Clotho-v2 (Drossos et al., 2020), AudioCaps (Kim et al., 2019) |
| | | CLS | UrbanSound8K (USD8K) (Salamon et al., 2014) (*environmental sound classification*), ESC50 (Piczak, 2015) (*environmental sound classification*), CochlScene (Jeong & Park, 2022) (*acoustic scene classification*), FSD50k (Fonseca et al., 2021) (*sound event classification*) CREMA-D (Cao et al., 2014) (*emotion classification*), Ravdess (Livingstone & Russo, 2018) (*emotion classification*), NonSpeech7k (Rashid et al., 2023) (*non-speech audio classification*) |
| | | AQA | Clotho-AQA (Lipping et al., 2022) (*audio question answering*) |
| | Music | CLS | NSynth (NS)$_{source}$ (Engel et al., 2017a) (*instrument classification*), NSynth (NS)$_{instrument}$ (Engel et al., 2017a) (*instrument classification*), GTZAN (Sturm, 2013) (*genre classification*), Medley-solos-DB (Lostanlen et al., 2019) (*instrument classification*) |
| | | AQA | MusicAVQA$_{audio}$ (Li et al., 2022) (*music audio question answering*) |
| Reasoning | Sound | AQA | OpenAQA (Gong et al., 2024), MMAU Sound (Sakshi et al., 2024), AudioEntailment (AE) Clotho (Deshmukh et al., 2024), AudioEntailment (AE) AudioCaps (Deshmukh et al., 2024), CompA-R-*test* (Ghosh et al., 2024c) |
| | Music | AQA | MMAU Music (Sakshi et al., 2024), MuchoMusic (Weck et al., 2024), MusicInstruct (MI) (Long) (Deng et al., 2023), MusciQA (Liu et al., 2024) |

Table 25: More examples from AudioSkills.

| Category | Example |
|---|---|
| Temporal Relationship Identification | **Order:** In what sequence do the sounds appear in the audio? (A) Car engine, honk, music (B) Music, honk, car engine
**Attribute:** When does the breathing sound change over time? (A) Disappear (B) Get louder (C) Get soft
**Grounding:** When does the sound of a bird chirp occur in the audio? (A) Beginning (B) Middle (C) End
**Referring:** What sound appears first in the audio? (A) Music (B) Car honk (C) Dog bark |
| | **Order:** In what sequence do the footsteps appear in the audio? (A) Slow footsteps, running, door slam (B) Running, door slam, slow footsteps (C) Door slam, slow footsteps, running
**Attribute:** How does the volume of the rain sound change over time? (A) Increases (B) Decreases (C) Stays the same
**Grounding:** When does the sound of a baby crying occur in the audio? (A) Beginning (B) Middle (C) End
**Referring:** What sound appears immediately after the thunder? (A) Wind blowing (B) Rainfall (C) Car alarm |
| Contextual Speech Event Reasoning | How might the playful interaction between the boy and the goat be affected by the tone of the speech and the background sounds? |
| | What can be deduced about the environment and the relationship between the boy and the goat from the audio events? |
| Contextual Sound Event Reasoning | What might the presence of the music and its continuous play suggest about the overall atmosphere of the scene?
Based on the timing and sequence of impact sounds and male speech, identify the possible work environment depicted in the audio. |
| Counting | **Level 1:** How many times was the hammer sound heard? |
| | **Level 2:** How many times did the first sound occur in the entire audio? |
| Information Extraction | What key is the main melody of the audio played in? (A) C major (B) A major (C) A# major (D) G# major |
| | How does the melody in the audio contribute to the hypnotic effect of the music? (A) By changing frequently (B) By maintaining a consistent and repetitive loop (C) By using multiple key changes |
| General Reasoning | Considering the chord sequence Bm7, D, Bm7, D in the audio, what is the role of the melody in relation to these chords? (A) It follows the progression (B) It decorates the progression |
| | How does the melody in the audio contribute to the hypnotic effect of the music? (A) By changing frequently (B) By maintaining a consistent loop |
| Attribute Identification | Which event has the highest pitch in the audio? |
| | Which is the least loud event in the audio? |

```
# Prompt

You are a helpful AI assistant. You need to act as a question-answer generator for long audios that generate Question-Answer pairs that require an
understanding of the order of sound events. These questions refer to types of questions that would ask the model about the order of particular sound events
in the audio. The question should be of MCQ type. You can be flexible with the number of options. The questions should require temporal reasoning to be
accurately answered.  My final objective is to train an audio agent with these question-answer pairs to endow it with long-audio understanding and QA
abilities. The audio is sourced from a video and you will be provided with multiple types of information about both the audio and video to generate the QA
pairs:

Video Short Caption: A short caption summarizing what the entire video is about, from a visual perspective.

Video Dense Caption: A detailed caption detailing all events in the entire video, from a visual perspective.

Video Main Object Caption: A caption describing the main focus of the entire video, from a visual perspective.

Video Style Caption: A short caption summarizing the style, in an abstract manner, in which the entire video is shot, from a visual perspective.

Video Camera Caption: A caption describing the camera angles in which the entire video is shot, from a visual perspective.

Video and Audio Segment Captions: This is the most important information which you should focus on. This is a list of lists. Every inner list has 2 captions
that describe a 10 second segment of the long video. The first caption describes the video while the second caption describes the audio of the 10 second
segment. The list or segments are in chronological order, which means that the first list is of the first 10 second segment, the second list of the second 10
second segment and so on.

Generate 4 question-answer pair with the following requirements:

1. The questions should be of the following types: Order, Attribute, Temporal Referring and Temporal Grounding.

2. Neither the prompt nor the response should be more than 15-20 words.

3. Do NOT use phrases like "according to the details" in both the prompts or responses; you should ask and answer as if you were hearing the audio by
yourself. Also, if the captions have timestamps please ignore them as they are noisy. Do not include them in the questions or answers.

4. **The audio captions may be noisy and not correct. You need to verify using the visual captions and the other visual cues before framing your QA pair.
Additionally, use the other visual details to understand the main importance of the broader scene and make QA pairs accordingly.** \n5. Most importantly,
only generate the questions and answers from an audio perspective, i.e., what you are hearing. An agent should be able to answer it only by hearing the
sounds. Do not ask questions that require visuals.

Here is an example with just the captions (you will be provided with more information as stated earlier):

"captions": [["A man in a black suit is standing in a room. He is holding a spoon in his right hand and a fork in his left hand. He is looking at the spoon
and fork. The man then puts the spoon and fork in his mouth.", "The audio contains a background of ominous instrumental music with a slow tempo featuring a
spooky electronic keyboard harmony and menacing bass on the cello. Additionally, there is a male speaking in Mandarin, who seems to be between 16-25 years
old and conveys a fearful mood."], ["A man is sitting in a room with a spoon in his hand.", "In this audio, there is music playing from the beginning to the
end. A female voice is speaking in English with a sad mood, saying 'Try and bend the spoon'. There's also the sound of breathing in between the spoken
words."], ["A man in a black suit is sitting in a room with a white chair and a wooden cabinet. He is holding a spoon in his hands and looking at it.", "The
background contains quiet music playing throughout the duration of the audio. A male voice speaks in English saying 'What truth? There is no spoon.' The
speaker's tone is neutral."]]). Question (Order type): What is the correct order of sounds as they appear in the audio?

(A) A woman speaking for the first time

(B) A child and a man speaking about a spoon

(C) Gunshots

(D) A man shouting 'Hey' Answer (Order type): (B)(A)(D)(C) – This question can also be phrased as "Arrange the sounds according to their sequence" and then
you provide the options and the answers.

Question (Temporal Referring type): What can be heard towards the beginning of the audio?

(A) A man shouting 'Hey'

(B) Gunshots

(C) A women speak

(D) A child and a man talk about a spoon Answer (Temporal Referring type): (D) A child and a man talk about a spoon – This question can also be phrased as
"What happens first?"

Question (Temporal Grounding type): Where in the audio are suddenly gunshots heard?

(A) the begin

(B) the middle

(C) the end

Answer (Temporal Grounding type): (C) the end

Question (Attribute type): How does the intensity of sounds change throughout the audio?

(A) It becomes more louder and intense

(B) It remains unchanged

(C) It becomes more softer and calm

Answer (Attribute type): (A) It becomes more louder and intense

Generate one question per type. Return a JSON in the following format: {"Temporal Referring Question": Question, Temporal Referring Answer": Answer,
"Temporal Grounding Question": Question, "Temporal Grounding Answer": Answer, "Order Question": Question, "Order Answer": Answer, "Attribute Question":
Question, "Attribute Answer": Answer}.

Only return the JSON and nothing else. Remember that the sounds should be distinguishable from a temporal perspective and pay attention that questions do not
involve overlapping sounds for which temporal order or comparison is hard to make. If a question type is not possible, don't output and only output "None".
Here is the input information:
```

Figure 9: Prompt 1 used for generating **Temporal QA** for LongAudio from MiraData.

```
#Prompt

I want you to act as a complex question/answer generator. I will provide you with an audio caption that describes the order in which multiple sounds occur
(using words like 'with', 'after', 'followed by', etc.). Additionally, I will provide a list of sound events with duration information (e.g., Rain - 0.000-
10.000; the start time is 0.000 seconds, and the end time is 10.000 seconds, so the duration is end time - start time = 10.000 seconds). Your task is to
generate complex multiple-choice questions (MSQs) with four options. I have provided the question type and example question, answers below

Temporal Event Reasoning
(Understanding the sequence and timing of events in the audio. e.g. For the given audio, identify which of the following sounds can be heard for the longest
duration.)
audio caption: "A fire truck's siren can be heard over traffic and human voices."
List of audio events: "['(Fire engine, fire truck (siren)-0.000-10.000)', '(Traffic noise, roadway noise-0.000-10.000)', '(Human voice-8.546-9.227)']"

Type 1. Identify sound with longest duration:
Example Question: For the given audio, identify which of the following sounds can be heard for the longest duration.
Reasoning for Question Generation: This question focuses on audio understanding of the model.
Options:
1. Fire engine, fire truck
2. Human voice
3. Wind
4. Cat Meowing
Correct Option:
Fire engine, fire truck
Rules for generating option:
1. Skip the questions for which the longest noise is background sound or noise
2. You should keep some incorrect options from the sample event list like Human voice and others sound event which are not present in the list of audio
event: Wind, Cat Meowing

Reasoning:
1. Duration calculation:
    Fire engine, fire truck (siren)-0.000-10.000 ->  10.000 - 0.000 = 10.000 (end time - start time)
    Traffic noise, roadway noise -> 10.000 - 0.000 = 10.000
    Human voice -> 9.227 - 8.546 = 0.66
2. Match with option: Only Fire engine and Human voice is present in the option. So the correct answer is Fire engine, fire truck

Type 2. Identify sound with shotest duration:
Example Question: For the given audio, identify which of the following sounds can be heard for the shortest duration.
Reasoning for Question Generation: This question focuses on audio understanding of the model.
Options:
1. Fire engine, fire truck
2. Human voice
3. Wind
4. Cat Meowing
Correct Option:
Human voice

While designing the options, ensure they contrast clearly with the correct choice, making it easier for a model to use reasoning skills to identify the true
cause of the child's emotional response.
Guidelines:
1. Always ask about events with a significant duration. Use the start and end times provided to determine the duration after each event. Ignore insignificant
events, such as background or static noise, which could confuse listeners due to their ambiguous sources.
2. Focus on events that can be generated by identifiable sources, like human activities (crying, clapping) or animal sounds (barking, meowing).
3. If the audio description and list of audio events doesn't include such identifiable activities, do not generate questions or options for it.
4. Each questions must be prepended by phrases like 'Based on the given audio', 'For the given audio sample', 'Given the audio sample', etc.
Output Format:
The output format must be json. I want you to generate 1 question for each categories. Here is an example:
{'Question type': <question type>, 'Question': <generate questions>, 'Options': [<choice1>, <choice2>, <choice3>, <choice4>], 'Answers': <correct option>,
Reason: <reason for selecting a option>, Category: 'Temporal'}. Make sure question, options and answers must not exceed 20, 10, 10 words.
Here is the audio caption: <ap>
time aligned audio event: <ae>
```

Figure 10: Prompt 2 used for generating **Temporal QA**.

# Prompt

You are a helpful AI assistant. You need to act as a question-answer generator for long audios that generate Question-Answer pairs. I want you to generate question-answer pairs that question the broader plot of the audio. The questions should require a sufficient understanding of the contents of the audio and reasoning based on the understanding to be accurately answered. My final objective is to train an audio agent with these question-answer pairs to endow it with long-audio understanding and QA abilities. The audio is sourced from a video and you will be provided with multiple types of information about both the audio and video to generate the QA pairs:

Video Short Caption: A short caption summarizing what the entire video is about, from a visual perspective.

Video Dense Caption: A detailed caption detailing all events in the entire video, from a visual perspective.

Video Main Object Caption: A caption describing the main focus of the entire video, from a visual perspective.

Video Style Caption: A short caption summarizing the style, in an abstract manner, in which the entire video is shot, from a visual perspective.

Video Camera Caption: A caption describing the camera angles in which the entire video is shot, from a visual perspective.

Video and Audio Segment Captions: This is the most important information which you should focus on. This is a list of lists. Every inner list has 2 captions that describe a 10 second segment of the long video. The first caption describes the video while the second caption describes the audio of the 10 second segment. The list or segments are in chronological order, which means that the first list is of the first 10 second segment, the second list of the second 10 second segment and so on.

Generate 1 question-answer pair with the following requirements:

1. The answers should be abstract and crisp and not describe each detail.

2. Neither the prompt nor the response should be more than 15-20 words.

3. Do NOT use phrases like "according to the details" in both the prompts or responses; you should ask and answer as if you were hearing the audio by yourself. Also, if the captions have timestamps please ignore them as they are noisy. Do not include them in the questions or answers.

4. **The audio captions may be noisy and not correct. You need to verify using the visual captions and the other visual cues before framing your QA pair. Additionally, use the other visual details to understand the main importance of the broader scene and make QA pairs accordingly.**

5. Most importantly, only generate the questions and answers from an audio perspective, i.e., what you are hearing. An agent should be able to answer it only by hearing the sounds. Do not ask questions that require visuals.

Here is an example with just the captions (you will be provided with more information as stated earlier):  "captions": [["A man in a black suit is standing in a room. He is holding a spoon in his right hand and a fork in his left hand. He is looking at the spoon and fork. The man then puts the spoon and fork in his mouth.", "The audio contains a background of ominous instrumental music with a slow tempo featuring a spooky electronic keyboard harmony and menacing bass on the cello. Additionally, there is a male speaking in Mandarin, who seems to be between 16-25 years old and conveys a fearful mood."], ["A man is sitting in a room with a spoon in his hand.", "In this audio, there is music playing from the beginning to the end. A female voice is speaking in English with a sad mood, saying 'Try and bend the spoon'. There's also the sound of breathing in between the spoken words."], ["A man in a black suit is sitting in a room with a white chair and a wooden cabinet. He is holding a spoon in his hands and looking at it.", "The background contains quiet music playing throughout the duration of the audio. A male voice speaks in English saying 'What truth? There is no spoon.' The speaker's tone is neutral."], ["A man is sitting at a table with a spoon in his hand.", "The audio contains a wide synth pad playing at a medium-to-high pitch with reverb effect from start to finish of the loop. A child's voice can be heard saying 'Ben, it is only yourself.' The speech is delivered in English with a sad mood by a female singer."], ["A bald boy is looking at a mirror, while a woman with braids is standing behind him.", "The audio contains an ambient track with a synth pad playing at a medium-to-high pitch along with shimmering bells fading out towards the end. A female voice speaking in Mandarin can also be heard saying '\u5929\u5bab\u4e00\u53f7\u5c06\u4e0e\u4f60\u540c\u884c'. The speech has a Harbin accent in Mandarin, delivered by a 34-year-old female speaker."], ["A man is looking at himself in a mirror.", "In this audio, there is continuous background noise along with intermittent source-ambiguous sounds resembling surface contact at intervals (0.58 to 1.37, 2.49 to 2.64, 2.88 to 3.07, 3.47 to 3.64, 3.87 to 4.05, 4.46 to 4.63, 4.86 to 5.06, 5.43 to 5.62, 5.86 to 6.04, 6.43 to 6.62, 6.85 to 7.03, 7.44 to 7.62, 7.86 to 8.04, 8.42 to 8.61, 8.85 to 9.03, 9.42 to"], ["A woman in a black hoodie is holding a gun and pointing it at a man in a black suit. The man is standing in front of a window. The woman is wearing sunglasses and has her hair pulled back. The man is wearing a black suit and has a black belt. The woman is holding a black gun and has her hair pulled back. The man is wearing a black suit and has a black belt. The woman is holding a black gun and has her hair pulled back. The man is wearing a black suit and has a black belt. The woman is holding a black gun and has her hair pulled back. The man is wearing", "The audio contains gunshots, video game background noise, source-ambiguous generic impact sounds, explosions, and gunfire. Additionally, there is a male voice shouting 'Hey' in English with a fearful mood."], ["A man in a black suit is standing in a room with a gun in his hand. He is pointing the gun at two men who are standing in front of him. The man in the suit is wearing a black hat and sunglasses. The room is dark and there is a lot of smoke in the air. The man in the suit is holding a gun and is pointing it at the two men in front of him. The two men are standing in front of him and are also holding guns. The man in the suit is wearing a black hat and sunglasses. The room is dark and there is a lot of smoke in the", "The audio contains sounds of gunshots, artillery fire at intervals (0.00-0.70, 3.26-4.08, and 6.72-10.00), video game background noise throughout (0.00-10.00), automatic gunfire (0.79-6.73), and generic impact sounds (4.56-4.83). Additionally, there is speech in Mandarin by a male aged 16-25 with a neutral mood saying '\u65e5\u672c\u653f\u5e9c\u5df2\u7ecf\u8c03\u4e8e\u4e03\u6708\u56db\u65e5'."]]. Question: Describe the plot of the audio. Answer: A man's introspective journey with a spoon shifts as a child's voice hints at self-discovery, ending in intense gunfire and confrontation.

Generate one question and answer pair. Return a JSON in the following format: {"Question": Question, "Answer": Answer}.

Only return the JSON and nothing else. Here is the input information:

Figure 11: Prompt used for generating **Plot QA** for LongAudio from MiraData.

---

**# Prompt**

You are a helpful AI assistant. You need to now act as an audio caption writer for long audios that describes the scene from an auditory perspective or the sounds that can be heard. My final objective is to train an audio agent with these audio-caption pairs to endow it with long-audio understanding and captioning abilities. The audio is sourced from a video and you will be provided with multiple types of information about both the audio and video to generate the QA pairs:

Video Short Caption: A short caption summarizing what the entire video is about, from a visual perspective.

Video Dense Caption: A detailed caption detailing all events in the entire video, from a visual perspective.

Video Main Object Caption: A caption describing the main focus of the entire video, from a visual perspective.

Video Style Caption: A short caption summarizing the style, in an abstract manner, in which the entire video is shot, from a visual perspective.

Video Camera Caption: A caption describing the camera angles in which the entire video is shot, from a visual perspective.

Video and Audio Segment Captions: This is the most important information which you should focus on. This is a list of lists. Every inner list has 2 captions that describe a 10 second segment of the long video. The first caption describes the video while the second caption describes the audio of the 10 second segment. The list or segments are in chronological order, which means that the first list is of the first 10 second segment, the second list of the second 10 second segment and so on.

Generate 1 audio caption with the following requirements:

1. The caption should be abstract and crisp and describe the broader scene.

2. The caption should not be more than 20-25 words.

3. Do NOT use phrases like "according to the details" or "sound of a" in the generated caption. Also, if the segment captions have timestamps please ignore them as they are noisy. Do not include them in the questions or answers.

4. **The segment-wise audio captions may be noisy and not correct. You need to verify using the visual captions and the other visual cues before framing your QA pair. Additionally, use the other visual details to understand the main importance of the broader scene and make QA pairs accordingly.**

5. Most importantly, only generate the questions and answers from an audio perspective, i.e., what you are hearing. An agent should be able to answer it only by hearing the sounds. Do not ask questions that require visuals.

Do not just return a caption with comma-separated audio events and rather write a linguistically coherent caption in AudioCaps style. Generate one caption. Return a JSON in the following format: {"Caption": Caption}.

Only return the JSON and nothing else. Here is the input information:

Here is an example with just the captions (you will be provided with more information as stated earlier): "captions": [["A man in a black suit is standing in a room. He is holding a spoon in his right hand and a fork in his left hand. He is looking at the spoon and fork. The man then puts the spoon and fork in his mouth.", "The audio contains a background of ominous instrumental music with a slow tempo featuring a spooky electronic keyboard harmony and menacing bass on the cello. Additionally, there is a male speaking in Mandarin, who seems to be between 16-25 years old and conveys a fearful mood."], ["A man is sitting in a room with a spoon in his hand.", "In this audio, there is music playing from the beginning to the end. A female voice is speaking in English with a sad mood, saying 'Try and bend the spoon'. There's also the sound of breathing in between the spoken words."], ["A man in a black suit is sitting in a room with a white chair and a wooden cabinet. He is holding a spoon in his hands and looking at it.", "The background contains quiet music playing throughout the duration of the audio. A male voice speaks in English saying 'What truth? There is no spoon.' The speaker's tone is neutral."], ["A man is sitting at a table with a spoon in his hand.", "The audio contains a wide synth pad playing at a medium-to-high pitch with reverb effect from start to finish of the loop. A child's voice can be heard saying 'Ben, it is only yourself.' The speech is delivered in English with a sad mood by a female singer."], ["A bald boy is looking at a mirror, while a woman with braids is standing behind him.", "The audio contains an ambient track with a synth pad playing at a medium-to-high pitch along with shimmering bells fading out towards the end. A female voice speaking in Mandarin can also be heard saying '\u5929\u5bab\u4e00\u53f7\u5c06\u4e0e\u4f60\u540c\u884c'. The speech has a Harbin accent in Mandarin, delivered by a 34-year-old female speaker."], ["A man is looking at himself in a mirror.", "In this audio, there is continuous background noise along with intermittent source-ambiguous sounds resembling surface contact at intervals (0.58 to 1.37, 2.49 to 2.64, 2.88 to 3.07, 3.47 to 3.64, 3.87 to 4.05, 4.46 to 4.63, 4.86 to 5.06, 5.43 to 5.62, 5.86 to 6.04, 6.43 to 6.62, 6.85 to 7.03, 7.44 to 7.62, 7.86 to 8.04, 8.42 to 8.61, 8.85 to 9.03, 9.42 to"], ["A woman in a black hoodie is holding a gun and pointing it at a man in a black suit. The man is standing in front of a window. The woman is wearing sunglasses and has her hair pulled back. The man is wearing a black suit and has a black belt. The woman is holding a black gun and has her hair pulled back. The man is wearing a black suit and has a black belt. The woman is holding a black gun and has her hair pulled back. The man is wearing a black suit and has a black belt. The woman is holding a black gun and has her hair pulled back. The man is wearing", "The audio contains gunshots, video game background noise, source-ambiguous generic impact sounds, explosions, and gunfire. Additionally, there is a male voice shouting 'Hey' in English with a fearful mood."], ["A man in a black suit is standing in a room with a gun in his hand. He is pointing the gun at two men who are standing in front of him. The man in the suit is wearing a black hat and sunglasses. The room is dark and there is a lot of smoke in the air. The man in the suit is holding a gun in his hand and is pointing it at the two men in front of him. The two men are standing in front of him and are holding guns. The man in the suit is wearing a black hat and sunglasses. The room is dark and there is a lot of smoke in the", "The audio contains sounds of gunshots, artillery fire at intervals (0.00-0.70, 3.26-4.08, and 6.72-10.00), video game background noise throughout (0.00-10.00), automatic gunfire (0.79-6.73), and generic impact sounds (4.56-4.83). Additionally, there is speech in Mandarin by a male aged 16-25 with a neutral mood saying '\u65e5\u672c\u653f\u5e9c\u5df2\u7ecf\u8c03\u6574\u4e8e\u4e03\u6708\u56db\u65e5'."]]}. Question: Describe the plot of the audio. Answer: A man's introspective journey with a spoon shifts as a child's voice hints at self-discovery, ending in intense gunfire and confrontation.

Do not just return a caption with comma-separated audio events and rather write a linguistically coherent caption in AudioCaps style. Generate one caption. Return a JSON in the following format: {"Caption": Caption}.

Only return the JSON and nothing else. Here is the input information:

---

Figure 12: Prompt used for generating **Caption QA** for LongAudio from MiraData.

# Prompt 1

I will provide you with 2 different types of information about a 10-second audio clip:

1. A list where each comma-separated element indicates the individual events occurring in the audio at various time segments. For example, '(Speech-0.0-0.64)' would mean human speech between 0.0 second to 0.64 second.
2. A caption of the audio describing in a brief and abstract manner the scene in which the audio takes place.

I want you to act as a Prompt Generator. According to the event information and the caption, design some instructions and corresponding responses. The instruction should be designed in a way such that it can be answered only from the audio without the caption and any other detail provided. The instruction should involve one or more hops of complex knowledge and complex reasoning based on the scene created by the audio and the correspnding caption. Ensure that the knowledge and reasoning chains in the instructions are precise and sufficiently challenging, to the extent that only well-educated people and experts in the respective field can provide adequate responses.

The instructions must meet the following conditions:
1. Do NOT use phrases like 'according to the caption' in both the questions and answers; you should ask and answer as if you were observing the image by yourself.
2. The questions and answers should be as diverse as possible.
3. Please don't ask some simple questions about the intensity of the audio or the gender speaking the utterance; your questions must involve some knowledge.
4. Your instructions should not be answered directly based on the image and your instructions. Instead, it requires the test-taker to carefully observe the image and have a deep knowledge of the content within the image in order to answer correctly.
5. If a question cannot be answered, please do not ask.

Come up with 3 diverse instructions for the knowledge topics above with different language styles and accurate answers. The instructions should contain interrogative sentences and declarative sentences. The answers should be less than 30 words.

Output format, which is a list of jsons:

[{'Instruction': instruction example, 'Answer': answer example, 'Knowledge topic': The specific knowledge topic}, {'Instruction': instruction example, 'Answer': answer example, 'Knowledge topic': The specific knowledge topic}, …]
Here are some examples of inputs and outputs:

Input list of audio events: ['(Speech-0.0-0.64)', '(Mechanisms-0.0-10.0)', '(Dog-0.221-0.547)', '(Dog-0.803-0.966)', '(Generic impact sounds-0.885-1.129)', '(Tick-0.99-1.083)', '(Dog-1.432-1.665)', '(Speech-1.537-4.901)', '(Dog-1.921-2.119)', '(Dog-2.456-3.202)', '(Dog-3.434-3.597)', '(Dog-4.016-4.121)', '(Dog-4.936-5.39)', '(Generic impact sounds-5.204-5.611)', '(Speech-5.984-6.787)', '(Tick-6.508-6.636)', '(Dog-6.717-8.266)', '(Generic impact sounds-7.649-8.277)', '(Laughter-8.347-9.488)', '(Dog-9.767-10.0)']
Caption: A baby cries while a woman laughs, creating a joyful and lively atmosphere in a domestic setting.

Output list of jsons: [{'Instruction': 'Analyze the sounds in the audio and determine the most likely cause of the laughter heard towards the end of the recording. Consider the potential interactions between the different sound sources and their temporal overlaps.' , 'Answer': 'The laughter likely results from the playful interaction between the dogs and the baby, as indicated by the overlapping sounds of dogs and the baby's presence.', 'Knowledge topic': 'Human and Animal Behavior Interpretation'}, {'Instruction': 'From the given audio, infer the type of domestic setting depicted in the scene. Base your inference on the variety and sequence of sounds, particularly focusing on the interaction between the human speaking, the dog barking, and other background noises that may be there.', 'Answer': 'The setting is likely a home with an active family environment, evidenced by the continuous presence of dogs, speech, and everyday household sounds.', 'Knowledge topic': 'Environmental Acoustics and Domestic Soundscapes'}, {'Instruction': 'Considering the duration and placement of speech and laughter in the audio, infer the possible emotional dynamics between the speakers. How do these elements interact to shape the scene's atmosphere?', 'Answer': 'The scene likely transitions from a more chaotic or lively mood and finally to a more joyful and relaxed atmosphere.'}]

Input list of audio events: ['(Insect-0.0-0.724)', '(Mechanisms-0.0-9.777)', '(Female speech, woman speaking-0.737-1.434)', '(Bird vocalization, bird call, bird song-1.243-1.775)', '(Insect-2.376-3.182)', '(Female speech, woman speaking-3.386-3.509)', '(Insect-4.397-5.23)', '(Dog-7.906-8.78)', '(Surface contact-8.603-9.654)']
Caption: 'Birds chirp in the distance as a dog barks, creating a lively atmosphere in a peaceful outdoor setting.'
Output list of jsons: [{'Instruction': 'What time of day this scene is likely set in?.' , 'Answer': 'The concurrent presence of insect and bird sounds suggests a natural, outdoor environment, possibly during early morning or evening when such wildlife is typically active.', 'Knowledge topic': 'Environmental Sound Analysis and Wildlife Behavior'}, {'Instruction': 'Analyze the presence and timing of the dog's barking in the latter part of the audio. Considering the preceding sounds and infer the dog's possible reaction or behavior in this context.', 'Answer': 'The dog's barking following the peaceful nature sounds and speech could indicate a response to a new stimulus, possibly a visitor or an animal in the area.', 'Knowledge topic': 'Animal Behavior Analysis in Diverse Sound Environments'}, {'Instruction': 'Deduce the woman's likely activity or purpose in this setting.', 'Answer': 'The woman might be engaging in an outdoor activity like gardening or bird-watching.','Knowledge topic': 'Human activity recognition through scene analysis' }]

Input list of audio events: ['(Music-0.0-10.0)', '(Male singing-0.0-10.0)', '(Male speech, man speaking-0.354-1.364)', '(Male speech, man speaking-7.674-10.0)', '(Crowd-7.681-10.0)']
Caption: 'A basketball bounces while music plays, and a man speaks in an indoor stage environment.'
Output list of jsons:  [{'Instruction': 'Considering the presence of crowd sounds towards the end of the audio, deduce the nature of the event taking place. How do the elements of music, singing, and speech suggest the type of event and audience involvement?', 'Answer': 'The event seems to be a live performance or concert, with the crowd's reaction indicating an engaged and responsive audience, typical in such settings.', 'Knowledge topic': 'Event Atmosphere Analysis'}, {'Instruction': 'Given the continuous presence of music and male singing throughout the audio, analyze the role of the man's speech in shaping the atmosphere of the scene. How does his speech, interspersed with music and singing, contribute to the overall environment?', 'Answer': 'The man's speech likely serves as commentary or narration, adding a personal or interactive element to the musical performance, enhancing the audience's engagement.', 'Knowledge topic': 'Music and Speech Dynamics'}, {'Instruction': 'Identify the genre of music being played and explain how it complements the atmosphere of the indoor stage environment.', 'Answer': 'The genre is likely upbeat or energetic, enhancing the lively ambiance of a sports or performance event in an indoor setting.','Knowledge topic': 'Music Genre Detection and Scene Analysis'}]

Input list of audio events: {timestamp events}
Caption: {caption}
Output list of jsons:

Figure 13: Prompt used for generating **Contextual Sound Event Reasoning QA** for AudioSkills.

```
# Prompt

You are a helpful AI assistant. You are a helpful AI assistant. You need to now act as an audio caption writer for long audios that describes the scene
from an auditory perspective or the sounds that can be heard. You will be provided with a list of lists, where each inner list will have 3 different
kinds of meta-data for a short 4-second segment of a 5-minute audio clip. All lists combined represent all meta-data for all 4 second segments of the
5-minute audio clip.

An example of an inner list with meta-data of a 4-second scene is: ['#C C drives the car', 'The audio contains the sound of something being fried in
oil, followed by the sound of that same item being placed on a paper towel. There is no speech or music present in this clip.', 'A person is cooking in
a kitchen. They are stirring a pot on the stove and adding ingredients to it.']

The first element is an action caption of the 4 second scene. "#C C" is the person doing the action as the video comes from an ego-centric video. The
caption is generated from a captioning model and might be noisy and prone to errors.

The second element is an audio caption of the 4 second scene. The caption is generated from an audio-captioning model and might be noisy and prone to
errors.

The third element is a video caption of the 4 second scene. The caption is generated from a video-captioning model and might be noisy and prone to
errors.

Beyond this, I will also provide you with one action description that describes the actions that take place in the entire egocentric video.

All inner lists are chronologically arranged in 4-second segments from beginning to end. You might take into account the cooccurrence of events inter
or intra-list to be robust to noisy meta-data.

Generate the caption with the following requirements:

1. You need to reason and generate a single sentence caption about the entire audio file in about 15-20 words. You do not need to describe every small
audio event in the caption, write a caption that describes the broader scene from an auditory perspective.

2. Do not use the phrase 'Sound of a'.

3. Use simple words for the captions and only generate the audio caption from an auditory perspective in AudioCaps style.

4. Do not just return a caption with comma separated audio events and rather write a linguistically coherent caption.**The segment-wise audio captions
may be noisy and not correct. You need to verify using the visual captions before framing the final caption.**

Only return the caption and nothing else.
```

Figure 14: Prompt used for generating **Caption QA** for Video-ReCap.

Table 26: Prompts designed for specific audio processing tasks, used to transform foundational audio understanding datasets into QA pairs.

| Task Name | Prompts |
|---|---|
| Instrument Classification | What instrument is playing in the audio?, Identify the instrument in this audio clip., Classify the instrument heard in the music., What is the primary instrument in this track?, Provide the instrument tag for this audio., Which instrument is most prominent in the clip?, What instrument is featured in this music? |
| Audio Captioning | Caption the input audio., Describe the sounds in the audio., Provide a caption for the audio., What is happening in this audio clip?, Summarize the audio content., Describe the events in the audio., What sounds can you hear in this audio?, Give a detailed description of the audio scene., Caption the sounds in the audio., What is the main action or event in the audio?, How would you describe the sounds in this audio? |
| Music Captioning | Describe the music in the audio., Provide a caption for the music., Summarize the characteristics of the music., Summarize the music content in a sentence., Caption the input music. |
| Speech Emotion Classification | Identify the emotion in the utterance., What is the emotion of the utterance?, Describe the emotional tone in this audio., What emotion is expressed in the audio?, What is the primary emotion in this recording?, Identify the feeling conveyed in the utterance., How would you describe the emotion in the audio?, What emotion stands out in this audio clip?, Describe the mood of the speaker., What sentiment is present in the audio?, What is the dominant emotion in this audio?, Classify the emotion expressed in the clip. |

# Prompt

You are a helpful AI assistant. You need to act as a question-answer generator for long audios that generates sub-scene captioning questions. Sub-scene audio captioning questions refer to types of questions that would ask the model to caption a part of the audio before and after a particular set of events. The objective is to train an audio agent with these question-answer pairs to endow it with long-audio understanding and QA abilities. You will be provided with a list of lists, where each inner list will have 3 different kinds of meta-data for a short 4-second segment of a 5-minute audio clip. All lists combined represent all meta-data for all 4 second segments of the 5-minute audio clip.

An example of an inner list with meta-data of a 4-second scene is: ['#C C drives the car', 'The audio contains the sound of something being fried in oil, followed by the sound of that same item being placed on a paper towel. There is no speech or music present in this clip.', 'A person is cooking in a kitchen. They are stirring a pot on the stove and adding ingredients to it.']

The first element is an action caption of the 4 second scene. "#C C" is the person doing the action as the video comes from an ego-centric video. The caption is generated from a captioning model and might be noisy and prone to errors.

The second element is an audio caption of the 4 second scene. The caption is generated from an audio-captioning model and might be noisy and prone to errors.

The third element is a video caption of the 4 second scene. The caption is generated from a video-captioning model and might be noisy and prone to errors.

Beyond this, I will also provide you with one action description that describes the actions that take place in the entire egocentric video.

All inner lists are chronologically arranged in 4-second segments from beginning to end. You might take into account the cooccurrence of events inter or intra-list to be robust to noisy meta-data.

Generate 2 question-answer pairs with the following requirements:

1. The question and answer pair should be for sub-scene captioning. The question should mention two events, separated by a particular and significant scene in the middle which can be captioned. The scene should not be a random scene and can be of moderate length.

2. Mention the starting and ending enclosing events or actions in an abstract way (how a human would question an AI agent) and not directly. The model should reason which events or actions are being referred to and caption the scene in between accordingly.

3. The questions should have some level of complexity and require reasoning and a detailed understanding of the audio.

4. Neither the prompt nor the response should be more than 15-20 words.

5. Do NOT use phrases like "according to the details" in both the prompts or responses; you should ask and answer as if you were hearing the audio by yourself.

6. Most importantly, only generate the questions and answers from an audio perspective, i.e., what you are hearing. An agent should be able to answer it only by hearing the sounds. Do not ask questions that require visuals.

Return a JSON in the following format: {"Question 1": Question, "Answer 1": Answer, "Question 2": Question, "Answer 2": Answer}. **The segment-wise audio captions may be noisy and not correct. You need to verify using the visual captions before framing the final caption.**

Only return the JSON and nothing else.

Figure 15: Prompt used for generating **Subscene QA** for LongAudio from Video-ReCap.

```
# Prompt

You are a helpful AI assistant. You need to act as a question-answer generator for long audios. I will provide you details about a long audio, and you
need to generate reasoning or information-extraction-based QA pairs from it. The objective is to train an audio agent with these question-answer pairs
to endow it with long-audio QA abilities. You will be provided with a list of lists, where each inner list will have 3 different kinds of meta-data for
a short 4-second segment of a 5-minute audio clip. All lists combined represent all meta-data for all 4 second segments of the 5-minute audio clip.

An example of an inner list with meta-data of a 4-second scene is: ['#C C drives the car', 'The audio contains the sound of something being fried in
oil, followed by the sound of that same item being placed on a paper towel. There is no speech or music present in this clip.', 'A person is cooking in
a kitchen. They are stirring a pot on the stove and adding ingredients to it.']

The first element is an action caption of the 4 second scene. "#C C" is the person doing the action as the video comes from an ego-centric video. The
caption is generated from a captioning model and might be noisy and prone to errors.

The second element is an audio caption of the 4 second scene. The caption is generated from an audio-captioning model and might be noisy and prone to
errors.

The third element is a video caption of the 4 second scene. The caption is generated from a video-captioning model and might be noisy and prone to
errors.

Beyond this, I will also provide you with one action description that describes the actions that take place in the entire egocentric video.

All inner lists are chronologically arranged in 4-second segments from beginning to end. You might take into account the cooccurrence of events inter
or intra-list to be robust to noisy meta-data.

Generate 2 question-answer pairs with the following requirements:

1. The prompt should require the model to understand the audio, analyze the content, and extract some information from the long audio. Optionally, it
should require the model to reason. This information might be a significant event in the entire audio, a change, a memory event, or any interesting
aspect. You may also ask a needle-in-the-haystack question about one particular event in the audio. The questions should have some level of complexity.

2. Neither the prompt nor the response should be more than 15-20 words.

3. Do NOT use phrases like "according to the details" in both the prompts or responses; you should ask and answer as if you were hearing the audio by
yourself.

4. Only generate the questions and answers from an audio perspective, i.e., what you are hearing. An agent should be able to answer it only by hearing
the sounds. Do not ask questions that require visuals.

5. Do not give too many hints about other events in the question. Be as abstract as possible.

6. Use simple words.

7. Write the questions from a 3rd person perspective. Do not refer to "#C" anywhere in the question and rather refer as a "person" if you are.

8. Return a JSON in the following format: {"Question 1": Question, "Answer 1": Answer, "Question 2": Question, "Answer 2": Answer}. **The segment-wise
audio captions may be noisy and not correct. You need to verify using the visual captions before framing the final caption.**

Only return the JSON and nothing else.
```

Figure 16: Prompt used for generating **General QA** for LongAudio from Video-ReCap.

```
# Prompt

You are a helpful AI assistant. You need to act as a question-answer generator for long audio that generates reasoning-based sub-scene captioning Question-
Answer pairs. Sub-scene audio captioning questions refer to types of questions that would ask the model to caption a relevant part of a long audio that is
preceded and succeeded by a particular set of events. The questions should require reasoning to be accurately answered. My final objective is to train an
audio agent with these question-answer pairs to endow it with long-audio understanding and QA abilities. The audio is sourced from a video and you will be
provided with multiple types of information about both the audio and video to generate the QA pairs:

Video Short Caption: A short caption summarizing what the entire video is about from a visual perspective.

Video Dense Caption: A detailed caption detailing all events in the entire video from a visual perspective.

Video Main Object Caption: A caption describing the main focus of the entire video from a visual perspective.

Video Style Caption: A short caption summarizing the style in an abstract manner, in which the entire video is shot from a visual perspective.

Video Camera Caption: A caption describing the camera angles in which the entire video is shot from a visual perspective.

Video and Audio Segment Captions: This is the most important information which you should focus on. This is a list of lists. Every inner list has 2 captions
that describe a 10-second segment of the long video. The first caption describes the video, while the second caption describes the audio of the 10-second
segment. The list of segments are in chronological order, which means that the first list is of the first 10-second segment, the second list of the second 10-
second segment, and so on.

Generate 1 question-answer pair with the following requirements:

1. The question and answer pair should be for sub-scene captioning. The question should mention two events, separated by a particular and significant scene in
the middle which can be captioned. The scene should not be a random scene and can be of moderate length. \n2. Mention the starting and ending enclosing events
or actions in an abstract way (how a human would question an AI agent) and not directly. The model should reason which events or actions are being referred to
and caption the scene in between accordingly. \n3. Neither the prompt nor the response should be more than 15-20 words.

4. Do NOT use phrases like "according to the details" in both the prompts or responses; you should ask and answer as if you were hearing the audio by
yourself. Also, if the captions have timestamps please ignore them as they are noisy. Do not include them in the questions or answers.

5. **The audio captions may be noisy and not correct. You need to verify using the visual captions and the other visual cues before framing your QA pair.
Additionally, use the other visual details to understand the main importance of the broader scene and make QA pairs accordingly.**

6. Most importantly, only generate the questions and answers from an audio perspective, i.e., what you are hearing. An agent should be able to answer it only
by hearing the sounds. Do not ask questions that require visuals.

Here is an example with just the captions (you will be provided with more information as stated earlier): "captions": [["A man in a black suit is standing in
a room. He is holding a spoon in his right hand and a fork in his left hand. He is looking at the spoon and fork. The man then puts the spoon and fork in his
mouth.", "The audio contains a background of ominous instrumental music with a slow tempo featuring a spooky electronic keyboard harmony and menacing bass on
the cello. Additionally, there is a male speaking in Mandarin, who seems to be between 16-25 years old and conveys a fearful mood."], ["A man is sitting in a
room with a spoon in his hand.", "In this audio, there is music playing from the beginning to the end. A female voice is speaking in English with a sad mood,
saying 'Try and bend the spoon'. There's also the sound of breathing in between the spoken words."], ["A man in a black suit is sitting in a room with a white
chair and a wooden cabinet. He is holding a spoon in his hands and looking at it.", "The background contains quiet music playing throughout the duration of
the audio. A male voice speaks in English saying 'What truth? There is no spoon.' The speaker's tone is neutral."], ["A man is sitting at a table with a spoon
in his hand.", "The audio contains a wide synth pad playing at a medium-to-high pitch with reverb effect from start to finish of the loop. A child's voice can
be heard saying 'Ben, it is only yourself.' The speech is delivered in English with a sad mood by a female singer."], ["A bald boy is looking at a mirror,
while a woman with braids is standing behind him.", "The audio contains an ambient track with a synth pad playing at a medium-to-high pitch along with
shimmering bells fading out towards the end. A female voice speaking in Mandarin can also be heard saying
'\u5929\u5bab\u4e00\u53f7\u5c06\u4e0e\u4f60\u540c\u884c. The speech has a Harbin accent in Mandarin, delivered by a 34-year-old female speaker."], ["A man is
looking at himself in a mirror.", "In this audio, there is continuous background noise along with intermittent source-ambiguous sounds resembling surface
contact at intervals (0.58 to 1.37, 2.49 to 2.64, 2.88 to 3.07, 3.47 to 3.64, 3.87 to 4.05, 4.46 to 4.63, 4.86 to 5.06, 5.43 to 5.62, 5.86 to 6.04, 6.43 to
6.62, 6.85 to 7.03, 7.44 to 7.62, 7.86 to 8.04, 8.42 to 8.61, 8.85 to 9.03, 9.42 to"], ["A woman in a black hoodie is holding a gun and pointing it at a man
in a black suit. The man is standing in front of a window. The woman is wearing sunglasses and has her hair pulled back. The man is wearing a black suit and
has a black belt. The woman is holding a black gun and has her hair pulled back. The man is wearing a black suit and has a black belt. The woman is holding a
black gun and has her hair pulled back. The man is wearing a black suit and has a black belt. The woman is holding a black gun and has her hair pulled back.
The man is wearing", "The audio contains gunshots, video game background noise, source-ambiguous generic impact sounds, explosions, and gunfire. Additionally,
there is a male voice shouting 'Hey' in English with a fearful mood."], ["A man in a black suit is standing in a room with a gun in his hand. He is pointing
the gun at two men who are standing in front of him. The man in the suit is wearing a black hat and sunglasses. The room is dark and there is a lot of smoke
in the air. The man in the suit is holding a gun in his hand and is pointing it at the two men in front of him. The two men are standing in front of him and
are also holding guns. The man in the suit is wearing a black hat and sunglasses. The room is dark and there is a lot of smoke in the", "The audio contains
sounds of gunshots, artillery fire at intervals (0.00-0.70, 3.26-4.08, and 6.72-10.00), video game background noise throughout (0.00-10.00), automatic gunfire
(0.79-6.73), and generic impact sounds (4.56-4.83). Additionally, there is speech in Mandarin by a male aged 16-25 with a neutral mood saying
'\u65e5\u672c\u653f\u5e9c\u5df2\u7ecf\u8c03\u6574\u4e8e\u4e03\u6708\u56db\u65e5'."]]). Output Question 1: "What can be heard between the child starting to
speak about a spoon and the sudden gunfire?" Output Answer 1: "A man and a child speak about a spoon, amidst orchestral sounds, followed by a woman speak and
the background sound getting tense."

Return a JSON in the following format: {"Question 1": Question, "Answer 1": Answer}. If there is no significant event that can be captioned as such, just
return "None" for both Question and Answer. **Also please disregard any acoustic event in the audio captioning mentioning Mandarin speech. That is a
hallucination from the audio captioning model. Ignoring these events, make the QA pair accordingly.**

Only return the JSON and nothing else. Here is the input information:
```

Figure 17: Prompt used for generating **Subscene QA** for LongAudio from MiraData.

```
# Prompt

You are a helpful AI assistant. You need to act as a question-answer generator for audios of 10 seconds in length. The questions should ask about the
temporal order of events in the audio. To make such a QA pair, I will provide you with 2 types of information about the 10-second audio: 1) An audio
caption that describes the scene from an auditory perspective. It describes all the sound events, some visuals, and the relation between the sounds. 2)
A comma-separated list, where each item in the list has 2 elements: the first is the name of the sound event, the second is the starting time of that
sound in seconds in the 10-second audio, and the third is the ending time of the sound in of that sound in second in the 10-second audio. You need to
generate MCQ-type question-and-answer pairs for temporal understanding. An example is below:

Input Caption: "The wind blows, and water splashes as footsteps echo, suggesting someone is walking near a natural outdoor area."

Input  Timestamp List: "['(Walk, footsteps-0.0-0.278)', '(Wind noise (microphone)-0.0-7.841)', '(Wind-0.0-10.0)', '(Walk, footsteps-0.459-0.798)',
'(Walk, footsteps-1.084-1.317)', '(Walk, footsteps-1.61-1.956)', '(Tick-2.114-2.182)', '(Walk, footsteps-2.227-2.626)', '(Whistling-2.809-3.124)',
'(Walk, footsteps-2.897-3.311)', '(Walk, footsteps-3.687-4.026)', '(Walk, footsteps-4.357-4.68)', '(Tick-4.959-5.049)', '(Walk, footsteps-5.184-
5.531)', '(Walk, footsteps-5.944-6.29)', '(Walk, footsteps-6.787-7.126)', '(Crack-7.423-8.285)', '(Walk, footsteps-7.555-7.976)', '(Walk, footsteps-
8.239-8.533)', '(Walk, footsteps-8.879-9.112)', '(Tick-9.707-9.752)']"

Output Answer: {Question 1 (order): "In what sequence do the sounds first appear in the audio?:

(A) Wind noise

(B) A faint ticking sound appears.

(C) Footsteps

(D) A crack sound", Answer 1 (temporal order): "(A)(C)(B)(D)"

Question 2 (temporal referring): "What sound appears earliest in the audio?

(A) A ticking sound

(B) Crack sound

(C) Wind noise

(D) Whistling", Answer 2 (temporal referring):: "(C) Wind noise"

Question 3 (temporal grounding):: "When does the ticking sound occur in the audio?

(A) The beginning

(B) The middle

(C) The end", Answer 3 (temporal grounding): "(B) The middle"

Question 4 (temporal attribute): "What happens to the wind noise as the audio progresses?

(A) It becomes louder and more intense

(B) It remains unchanged

(C) It becomes softer and calm  Answer 4 (temporal attribute): (B) It remains unchanged}

1. Output a JSON of a similar style in the example.

2. Many sounds may be overlapping, and formulate sequence questions accordingly.

3. You may also ask different questions like "What sound appears last in the audio?", followed by the options and the answer.

4. If a question type cannot be made just return "None" for both Question and Answer.

5. Don't return a QA pair without any options. All questions should be strictly MCQ and have options.

Return only the JSON and nothing else.

Here are the input details:
```

Figure 18: Prompt used for generating **Temporal Reasoning QA** for AudioSkills.

```
# Prompt

Act like an a audio caption generator. I will provide you with an audio caption that describes the scene of an audio. You need to generate 3
linguistically rephrased captions. The rephrased captions should sound similar to the original, from an audio and acoustic scene perspective. You can
change one or more words to generate these captions. The changed words might correspond to acoustic elements, attributes, sounds or describe the actual
sounds behind the acoustic elements mentioned and more vividly.

Here are some examples:

Original Caption: Birds chirp and rustle in the background while a man speaks, possibly in an office setting, followed by the sound of a helicopter.

Rephrased Captions: [Birds chirp and rustle in the background while a man speaks, possibly in a closed room setting, followed by the sound of a
chopper., High-pitched birdsong echoes softly over faint rustling leaves, as a man's voice breaks the calm, followed by a distant helicopter's rhythmic
thrum., Birds chirp and leaves rustle softly in the background as a man speaks, likely indoors, followed by a helicopter's hum.]

Original Caption: A woman plays a guitar and sings to a group of people in a room, while a man's voice is heard in the background, and a baby cries
occasionally.

Rephrased Captions: [A woman strums a guitar and sings warmly to a gathered crowd, as a man's voice murmurs softly and a baby's occasional cries add
brief, poignant interruptions., A woman strums a guitar and sings in a room with a group of people, as a man's voice murmurs in the background and a
baby occasionally cries., A woman sings and plays guitar to an audience in a room, with a man's voice in the background and a baby's intermittent
cries.]. Return only a list of negative captions in a similar fashion.

Here is the new caption:
```

Figure 19: Prompt used for generating **linguistically variant positives** for short-audio captions for CLAP training.

```
# Prompt

Act like an audio caption rewriter. I will provide you with an audio caption that describes the scene of the audio. Rewrite the caption by just
removing the visual elements which do not contribute to describing the acoustic details. Keep the rest of the caption constant, and you may only change
a few words to keep the rewritten words linguistically coherent. Do not change any other words. If no such element is present, just return the original
caption.

Here are some examples:

Original Caption: Birds chirp and rustle in the background while a man speaks, possibly in an office or police setting, with a hazy atmosphere.

Rewritten Caption: Birds chirp and rustle in the background while a man speaks.

Original Caption: A motorboat is speeding by, followed by the sound of wind and wind noise, while remote-controlled boats create ripples on the lake
under a partly cloudy sky.

Rewritten Caption: A motorboat is speeding by, followed by the sound of wind and wind noise, while other boats create ripples on the lake.

Original Caption: A truck performs a burnout on a snowy road, emitting a large plume of black smoke, while a man's voice can be heard as the sun rises
or sets in the clear sky.

Rewritten Caption: A truck performs a burnout on a road while a man's voice can be heard.

Only return the rewritten caption and nothing else.\n\nHere is the new caption:
```

Figure 20: Prompt used for cleaning and removing noise from synthetic short-audio captions.

```
# Prompt

Act like a audio caption generator. I will provide you with an audio caption that describes the scene of the audio. You need to generate 3
compositionally different captions. The compositionally different captions should sound different from the original from an audio and acoustic scene
perspective. Flip one or more words in the caption to generate the captions. You may change one or more words related to order, attributes, sounds or
other words that can change the audio compositionally. You may also exchange words with other words in the caption itself. Keep the rest of the caption
constant.

Here are some examples:

Original Caption: Birds chirp and rustle in the background while a man speaks, possibly in an office or police setting, with a hazy atmosphere.

Negative Captions: [Birds chirp and rustle after a woman speaks, possibly in an office or police setting, with a hazy atmosphere., Birds chirp and
rustle in the background while a man speaks, possibly in an outdoor or garden setting with a hazy atmosphere., Dog barks before a man speaks, possibly
in an office or police setting, with a clear atmosphere.]

Original Caption: A woman plays a guitar and sings to a group of people in a room, while a man's voice is heard in the background, and a baby cries
occasionally.

Negative Captions: [A man plays a piano and sings to a group of people in a room, while a woman's voice is heard in the background, and a baby cries
occasionally., A woman plays a guitar and sings to a group of people outdoors, while a baby's voice is heard in the background, and a man cries
occasionally., A woman plays a guitar and hums to a group of people in a room, while a man's voice is heard in the background, and a baby laughs
occasionally.]. Return only a list of negative captions in a similar fashion.

Here is the new caption:
```

Figure 21: Prompt used for generating **compositionally different negatives** for short-audio captions for CLAP training.

```
# Prompt

You are given three captions:

1. Audio Caption: Generated solely from audio input by an audio language model. It may contain inaccuracies due to similar sounds produced by different
objects.

2. Visual Caption: Describes the scene from a vision-only perspective and can be used as ground truth to understand the scene.

3. Action Caption: Describes the actions from an egocentric (first-person) perspective.

Task:

Understand the scene using the visual caption.

Correct any errors in the audio caption based on the visual information.

Combine insights from all three captions to generate a coherent and accurate audio-visual caption that includes all the given information.

Interweave the audio and visual information together instead of mentioning visual information first and then audio

Instructions:

Omit any mandarin related information

Use the visual caption as ground truth to verify events in the audio caption; discard any misinterpreted events.

Align and integrate information from all captions to create a cohesive narrative.

Intermix the sounds and actions in the audio-visual caption; avoid mentioning sounds and actions separately.

Ensure the final caption accurately reflects the scene and is short, crisp, and to the point.

Do not include any additional text or explanations.

Output Format: Return only an audio-visual caption and nothing else.

Example 1 -
Audio Caption: The audio contains mechanisms such as ratchet - pawl, clock ticking, background noise, and a brief tone resembling a beep or bleep.
There is also a male voice speaking in Mandarin with an angry mood. The audio contains the sound of something being fried in oil, followed by the sound
of that same item being placed on a paper towel. There is no speech or music present in this clip. The audio contains crackling onomatopoeia that
continues from the beginning to the end of the recording (0.00 to 9.27 seconds). There is no speech or music present in this clip.

Visual Caption: A person is stirring a bowl of food with chopsticks. A person is cooking in a kitchen. They are stirring a pot on the stove and adding
ingredients to it. A person is stirring a pot of food on a stove.

Action Caption: #C C opens the bottle. #C C drops the spoon in the plastic container. #C C moves around the kitchen.\nAudio-visual-caption: A person
cooks, stirring a pot with chopsticks and adding ingredients, as frying crackles, brief beeps, and an agitated voice sound in the background.

Example 2 -
Audio Caption: The audio contains sounds of things and mechanisms throughout its entire duration from start to finish (0.00 to 10.00 seconds). The
audio contains domestic sounds, including frying food on a sizzle and the sound of dishes being clanked together. There is also a male voice speaking
in Mandarin saying '你一个人不是很无聊吗', which translates to 'Don't you get bored alone?' The speaker has an age between 16-25 years old and speaks with
a neutral mood. The audio contains mechanisms such as ratchet - pawl, clock ticking, background noise, and a brief tone resembling a beep or bleep.
There is also a segment where someone speaks in Mandarin saying '卖五金市场部的人就能休'.

Visual Caption: A person is cooking in a kitchen. They are stirring a pan of food on the stove. A person is cooking in a kitchen. They are stirring
food in a pan on the stove. A person is cooking in a  kitchen.

Action Caption: #C C stirs the food. #C C stirs food. #C C stirs food in the frying pan with the spoon.

Audio-visual-caption: A person cooks in the kitchen, stirring food on the stove as sizzling and clanking dishes mix with ticking occasionally
interrupted by brief beeps.Generate the result for: Audio Caption: {}, Visual Caption: {}, Action Caption: {}
```

Figure 22: Prompt used for generating **synthetic short-audio captions** from Video ReCAP.

```
#Prompt

Act like an audio caption generator. I will provide you with 2 different captions for a 10-second video clip: 1) An audio caption that describes the audio in
the 10-second clip and 2) A video caption that describes the visual elements in the 10-second video clip. You need to generate a linguistically correct
caption with the following rules: 1) Omit any acoustic element that says Mandarin speech or anything related, it is usually a hallucination 2) Use the visual
caption as ground truth to verify events in the audio caption; discard any misinterpreted events. Correct any errors in the audio caption based on the visual
information. 3) Combine insights from both captions to generate the final caption in AudioCaps style. 4) Do not use the terms 'in the audio', 'is heard',
etc. Describe the scene directly.

Here are some examples:
Video Caption: A man in a black suit is standing in a room. He is holding a spoon in his right hand and a fork in his left hand. He is looking at the spoon
and fork. The man then puts the spoon and fork in his mouth.
Audio Caption: The audio contains a background of ominous instrumental music with a slow tempo featuring a spooky electronic keyboard harmony and menacing
bass on the cello. Additionally, there is a male speaking in Mandarin, who seems to be between 16-25 years old and conveys a fearful mood.
Output Caption: Ominous instrumental music plays with a slow, eerie electronic harmony and deep bass.

Video Caption: A man is driving a car and talking to another man who is sitting in the passenger seat.
Audio Caption: The audio contains various sounds including things and mechanisms throughout its duration. There is also continuous background noise present
during the entire audio. A man speaks at several intervals, initially speaking from 0.00 to 0.53 seconds, then between 2.79 to 4.08 seconds, followed by 6.83
to 7.88 seconds, and finally from 9.11 to 10.00 seconds. In addition to the male speech, there is also a sound of laughter occurring between 4.03 to 4.96
seconds.
Output Caption: Car engine hums softly as two men talk, with brief laughter in between.

Only return the audio caption and nothing else.

Here are the input captions from which you need to generate an audio caption.
Video Caption: {}
Audio Caption: {}
Output Caption:
```

Figure 23: Prompt used for generating **synthetic short-audio captions** from MiraData.

```
#Prompt

Act like an audio caption rewriter. I will provide you with a ground truth audio caption that describes the scene of the audio. I will also provide another
audio and visual caption (for the video corresponding to the audio), using which the ground truth audio caption was generated. You need to rewrite the audio
caption and remove any acoustic elements (attributes, events, etc.) that are not plausible from the audio alone and may have been borrowed from the video
caption while generating the ground truth audio caption.

Here are some examples:
Ground Truth Caption: A man walking on grass.
Additional audio caption: Sound of footsteps on grass
Additional video caption: A man strolling in an empty garden.
Re-written audio caption:  A person walking on grass.

Only return the rewritten caption and nothing else.

Here are the new captions:
```

Figure 24: Prompt used for rewriting audio caption and removing implausible acoustic elements.

```
#Prompt

I want you to act as a Response Evaluator. I will provide you with a response to a question generated by an AI agent that understands audio. The
agent was asked to generate a response to a question about a long audio that ranges between 30 seconds and 5 minutes. To evaluate the response, I
will also provide you with the question and the ground truth answer that is human evaluated. Using the question and the ground truth answer, assign
a score of 1-10 to the response, where 1 is the lowest score and 10 is the highest score. Your evaluation should be based on the detailedness,
correctness, and bluntness of the response. Return a JSON with a single key 'score', where the value of the key is the score. Here is the response:
```

Figure 25: Prompt used for evaluating responses for questions in LongAudioBench.

```
#Prompt

You are tasked with generating question-answer (QA) pairs to help understand the attributes of events in an audio file. The purpose is to create insightful
and specific QA pairs that allow users to query and learn about the loudness, pitch, reverb, and duration of timestamped events in the audio. Below are the
details of the inputs provided and the expected output format:

Input Information
  1.  Audio File Information: The audio contains multiple timestamped events, with each event describing an activity or occurrence in the audio.
  2.  Attributes of Interest:
     - Loudness: The amplitude or perceived volume of the event in LKFS.
     - Pitch: The frequency of the event, often measured in Hz.
     - Reverb: The persistence of sound or echo characteristics in .
     - Duration: The length of time (in seconds) for which the event occurs.
  3.  Event Information:
     - A list of timestamped events with their corresponding captions.
     - Example of an event:

{
"audio_id": audio identifier
  "timestamp": [(dog barking, 0.0-2.4s),(man laughing, 2.4-6.0s)],
  "caption": "A dog barking in the distance, followed by a man laughing",
  "attributes": {
    "loudness": [15, 24],
    "pitch": [450 Hz, 100 Hz]",
    "reverb": [0.5, 1.2],
    "duration": [2.4, 3.6]
  }
}

  4.  Caption: A detailed text description of the entire audio or specific events in the audio.

Generate QA pairs in JSON format that meet the following requirements:
  1.  Each question should focus on the attributes (loudness, pitch, reverb, and duration) of specific events in the audio.
  2.  The questions should explore:
  •  Comparative attributes (e.g., "Which event has the highest pitch?")
  •  Temporal relationships (e.g., "What is the loudness of the event at 00:12:05?")
  •  Descriptive attributes (e.g., "How long does the event 'dog barking' last?")
  3.  Ensure the QA pairs are insightful and specific to the provided audio details.

Expected Output

The QA pairs should be returned in JSON format. Here is an example of the expected output structure:

{"audio_id": <audio_id>
  "qa_pairs": [
    {
      "question": "What is the loudness of the event at timestamp 00:2:05?",
      "answer": "15 LKFS"
    },
    {
      "question": "Which event has the longest duration?",
      "answer": "The event 'a man laughing' with a duration of 3.6 seconds."
    },
    {
      "question": "What is the pitch of the dog barking?",
      "answer": "450 Hz"
    },
    {
      "question": "How much reverb is associated with the event at timestamp 00:2:05?",
      "answer": "0.5"
    },
    {
      "question": "Which event has the highest loudness in the audio?",
      "answer": "Man laughing is the loudest sound in the audio."
    }
  ]
}

Key Notes
  1.  Ensure that all generated questions are tied to the provided events, attributes, and captions.
  2.  Create diverse questions, including:
     - Attribute-specific queries (e.g., loudness, pitch).
     - Temporal queries (e.g., events at specific timestamps).
     - Caption-driven queries (e.g., mapping descriptions to attributes).
     - Comparative or analytical queries (e.g., "Which event has the lowest reverb?")
  3.  If multiple events share similar attributes, mention them explicitly in the answers.

Return the generated QA pairs as a JSON file with the structure shown above. Make sure the questions are well-structured, and the answers are precise and
tied to the input data.

Here is the input:
```

Figure 26: Prompt used for generating **Attribute QA** for AudioSkills.

```
#Prompt

I want you to act as a complex question/answer generator. I will provide you with an audio caption that describes the order in which multiple sounds occur
(using words like 'with', 'after', 'followed by', etc.). Additionally, I will provide you a list of sound events with duration information (e.g., Rain – 0.000–
10.000; the start time is 0.000 seconds, and the end time is 10.000 seconds, so the duration is end time – start time = 10.000 seconds). Your task is to
generate complex multiple-choice questions (MSQs) with four options.

Event-Based Sound Reasoning
(Inferring cause-and-effect relationships between different sounds and events. For example, audio: "A crash followed by a child crying")
Example Question:
What could have caused the emotional response observed in the child's crying from the audio?
Reasoning for Question Generation:
This question focuses on understanding causality between events. The exact sound made by the child after the crash is intentionally omitted to highlight the
cause-and-effect reasoning.
Options:
1. A sudden impact or collision followed by the child getting hurt
2. A soothing lullaby playing softly nearby
3. A friendly conversation between adults in the background
4. A gentle splash of water from a nearby fountain
Correct Option:
A sudden impact or collision followed by the child getting hurt

While designing the options, ensure they contrast clearly with the correct choice, making it easier for a model to use reasoning skills to identify the true
cause of the child's emotional response.
Guidelines:
1. Always ask about events with a significant duration. Use the start and end times provided to determine the duration after each event. Ignore insignificant
events, such as background or static noise, which could confuse listeners due to their ambiguous sources.
2. Focus on events that can be generated by identifiable sources, like human activities (crying, clapping) or animal sounds (barking, meowing).
3. If the audio description and list of audio events doesn't include such identifiable activities, do not generate questions or options for it.
Output Format:
The output format must be json. Here is an example:
{'Question': <generate questions>, 'Options': [<choice1>, <choice2>, <choice3>, <choice4>], 'Answers': <correct option>, Reason: <reason for selecting a
option>}. Make sure question, options and answers must not exceed 20, 10, 10 words respectively.
```

Figure 27: Prompt 1 used for generating **Contextual Sound Event Reasoning QA** for AudioSkills.

```
#Prompt

I want you to act as a complex question/answer generator. I will provide you with a list of audio captions that describe individual sounds that can be heard
in the audio. Your task is to generate information-seeking multiple-choice questions (MSQs) with four options. Each question should focus on extracting
clear, factual information related to the sounds in the audio, not abstract reasoning about natural scenes or events.

Guidelines for Generating Questions:
Focus on information-seeking tasks:

Questions should aim at extracting specific information like weather conditions, time of day, type of sound, or season based on the audio.
Avoid scene understanding or reasoning questions like "What kind of environment is likely represented?" or "What scenario could these sounds represent?".
Ensure clarity in question formulation:

The question must directly ask for information that can be deduced from the audio description, without requiring complex interpretation.
For example:
Correct: "Based on the audio, what season is likely represented?"
Incorrect: "Based on the audio, what scenario could these sounds represent?"
Question formats to prefer:

"What natural phenomenon is indicated by the given sequence?"
"What weather condition is suggested by the sounds?"
"What season is likely represented by the audio?"
"What time of day is suggested based on the sounds?"
Avoid abstract reasoning:

Do not generate questions that require understanding of complex relationships between different sounds (like "wildfire extinguished by heavy rain").
Instead, focus on direct correlations between sounds and known natural events or facts.
Audio description focus:

Focus on the presence of certain types of sounds (e.g., thunder, rain, bird calls) to generate questions about natural facts.
If the audio description lacks identifiable activities related to weather, season, or natural phenomenon, do not generate a question.

Example Question Structure:

{
  "Question": <generate questions>,
  "Options": [<choice1>, <choice2>, <choice3>, <choice4>],
  "Answer": <correct option>,
  "Reason": <reason for selecting the correct option>
}

Examples for Diverse Information-Seeking Questions:
Audio Caption: "Alternating presence of thunder and rain sounds"

Question: "Based on the audio, what weather condition is likely occurring?"
Options: ["Thunderstorm", "Clear skies", "Light drizzle", "Heavy snow"]
Answer: "Thunderstorm"
Reason: "The presence of thunder and rain suggests a thunderstorm."
Audio Caption: "Sound of birds chirping and rustling leaves"

Output Format:
The output format must be json. Here is an example:
{'Question': <generate questions>, 'Options': [<choice1>, <choice2>, <choice3>, <choice4>], 'Answers': <correct option>, Reason: <reason for selecting a
option>}. Make sure question, options and answers must not exceed 20, 10, 10 words.
Here is the audio caption: <ap>
```

Figure 28: Prompt 2 used for generating **Contextual Sound Event Reasoning QA** for AudioSkills.

```
#Prompt

I want you to act as a question/answer generator. I will provide you with list of audio caption that states about individual sounds that can be heard in the
audio. Your task is to generate complex multiple-choice questions (MSQs) with four options. I have provided the question type and example question, answers
below

Event-Based Sound Reasoning: (Inferring cause-and-effect relationships between different sounds and events. For example, audio: "A crash followed by a child
crying")
audio caption: "A crash followed by a child crying"
Example Question: Based on the given audio, What could have caused the emotional response observed in the child's crying from the audio?
Reasoning for Question Generation:This question focuses on understanding causality between events. The exact sound made by the child after the crash is
intentionally omitted to highlight the cause-and-effect reasoning.
Options:
1. A sudden impact or collision followed by the child getting hurt
2. A soothing lullaby playing softly nearby
3. A friendly conversation between adults in the background
4. A gentle splash of water from a nearby fountain
Correct Option:
A sudden impact or collision followed by the child getting hurt

While designing the options, ensure they contrast clearly with the correct choice, making it easier for a model to use reasoning skills to identify the true
cause of the child's emotional response.
Guidelines:
While generating the question make sure that you hide the sound. For example:
Audio caption: A person is pulling silverware out of the dishwasher
Incorrect generation: From the given audio identify why is the person pulling a silverware out of the dishwasher
Correct generation: From the given audio identify why is the person doing the activity that can be heard from the audio
Reason: As the person is pulling the silverware out of the dishwasher, the most possible activity is he/she might be doing is cooking
Correct answer: cooking

Here are some more guidelines:
1. Ignore insignificant events, such as background or static noise, which could confuse listeners due to their ambiguous sources.
2. Focus on events that can be generated by identifiable sources, like human activities (crying, clapping) or animal sounds (barking, meowing).
3. If the audio description and list of audio events doesn't include such identifiable activities, do not generate questions or options for it.
4. Each questions must be prepended by phrases like 'Based on the given audio', 'For the given audio sample', 'Given the audio sample', etc.
Output Format:
The output format must be json. Here is an example:
{'Question': <generate questions>, 'Options': [<choice1>, <choice2>, <choice3>, <choice4>], 'Answers': <correct option>, Reason: <reason for selecting a
option>}. Make sure question, options and answers must not exceed 20, 10, 10 words.
Here is the list of audio caption: <ap>
```

Figure 29: Prompt 3 used for generating **Contextual Sound Event Reasoning QA** for AudioSkills.

```
#Prompt

I will provide you with 2 kinds of information about a 10 second audio:
1) A caption for the audio describing the sounds and the audio events taking place in the audio. This caption is also vision-inspired and constructed with a
combination of visual and audio cues.
2) The transcript of the spoken utterance (if any) in the audio. The transcript is noisy and might not be accurate.

You need to generate 2 question answer-pairs from this information for the audio. Here are the rules you should abide by to generate this question answer-
pair:
1.The questions should require complex reasoning from an AI agent to provide a response. Please do not provide simple question-answer pairs. Your questions
should not be answered directly based on the audio and the question. Instead, it should require the AI agent to carefully perceive the audio and have a deep
knowledge of the content to answer correctly.
2. The question should include reasoning about both the transcript and the sounds, i.e., the question should make the AI agent reason about both the
transcripts and the sound events.
3. Do NOT use phrases like "according to the caption" in both the questions and answers; you should ask and answer as if you were perceiving the audio by
yourself.
4. The questions and answers should be as diverse as possible.
5. The question should not refer to the contents of the transcript or the caption directly as the AI agent will just be exposed to the audio as a whole and
it is the AI agents duty to understand and perceive the speech and sounds from the audio.

Come up with 2 diverse question-answer pairs with different language styles and accurate answers. The questions should contain interrogative sentences and
declarative sentences. The answers should be less than 30 words.

Output format should be a JSON of JSONs of the following format:
{"Question": "The generated question", "Answer": "The generated answer for the question", "Knowledge topic": "The specific knowledge topic"}, {"Question":
...., "Answer": ...., "Knowledge topic": .....}, ...}
Here are 2 examples:

Input Caption: An adult male is speaking, followed by the sound of a computer mouse clicking and scrolling, while he navigates through a file directory,
searching for a specific file, amidst the background sounds of a computer's file explorer.
Input Transcription: installation directory is on my eDrive. I double click Steam and I go to oh excuse me, my mistake. You go to Steam apps then you go to
common.
Output Question: Given the sound of the computer mouse clicking and scrolling, how does the speaker's correction in the middle of its speech affect the
interpretation of the task he is performing?
Output Answer: The speaker's correction suggests an error in file navigation, highlighted by the mouse interactions, indicating an adjustment in his search
strategy.

Input Caption: A man speaks about severe thunderstorms producing damaging winds in various counties in Georgia, while a weather alert warns people to take
precautions on a certain date.
Input Transcription: Blackshear, Odom, Patterson, Scravin, Hoboken, Jessup, Dr. Town and Drace Pond. Doppler radar has indicated some weak rotation with the
Output Question: What could be inferred about the potential severity of the weather conditions given the locations mentioned by the person and the sounds in
the audio?
Output Answer: The audio suggests the severity is high, as the locations listed are part of an urgent weather alert, emphasizing the need for precautions.

If a question cannot be answered, please do not ask. If a question is not possible, please return "None". Please do not return any questions for non-English
or non-sensible transcripts. For all these cases, just return "None" for both questions and answers.

Input Caption:A woman passionately delivers a speech, her voice clear and confident, in an environment marked by female speech and narration.
Input Transcription:  at the end of the year. And this year the snake brings forth much more clarity and resolve and decision-making

Below is the input:
```

Figure 30: Prompt 4 used for generating **Contextual Speech Event Reasoning QA** for AudioSkills.

```
#Prompt

Your task is to generate a diverse set of music knowledge questions and their corresponding answers for each row in the dataset. Each question and answer
pair should be formatted as a JSON object.

These questions will be used to test Audio Language Models, so ensure they are directly relevant to the audio content described in the captions. You should
assign each question an appropriate difficulty level: 'easy', 'medium', or 'hard'. If an answer may vary because different models might generate similar but
not identical responses, frame the question as a multiple-choice question (MCQ).

You will be provided with:
1. Two captions: a music caption and an alternate caption. Both contain details about the music in the audio and their characteristics.
2. prompt_ch: A control sentence describing the chord sequence.
3. prompt_key: A control sentence related to the extracted musical key.
4. chords: The chord sequence contained in the track. This is used as an input for training Mustango.
5. chords_time: Timestamps of the detected chords. This is used as an input for training Mustango.

Use all of these information from these captions as the ground truth to create multiple questions for each audio, focusing questions about the **Harmony and
Chord Progressions** in the audio.

Instructions:

- Each question should refer to the audio file as 'the audio'.
- Ensure the questions are diverse and focus on testing knowledge of harmony and chords in the audio.
- Avoid questions like "What is the chord progression in the audio?", this question is very direct and lame. Rather make questions which can make you think
about some knowledge in terms of harmony and chords.
- Make use of captions provided to find the answer.
- Do not ask question about mood and emotions.
- Do not include too many questions about identifying instruments.
- Format your output precisely as specified.
- Only return the JSON object, nothing else

The expected JSON structure is:

    {{
        'audio_id': {path},
        'question': <question>,
        'options': [<option1>, <option2>, <option3>, <option4>],
        'answer': <correct_option>,
        'difficulty': <'easy'/'medium'/'hard'>,
        'category': <'rhythm and tempo'>
    }}

Below is the input:
```

Figure 31: Prompt 1 used for generating **Information Extraction QA** for AudioSkills.

```
#Prompt

Your task is to generate a diverse set of music knowledge questions and their corresponding answers for each row in the dataset. Each question and answer
pair should be formatted as a JSON object.

These questions will be used to test Audio Language Models, so ensure they are directly relevant to the audio content described in the captions. You should
assign each question an appropriate difficulty level: 'easy', 'medium', or 'hard'. If an answer may vary because different models might generate similar but
not identical responses, frame the question as a multiple-choice question (MCQ).

You will be provided with two captions: a music caption and an alternate caption. Both contain details about the music in the audio and their
characteristics. Use the information from these captions as the ground truth to create multiple questions for each audio, focusing questions about the
**Melody** in the audio.

Instructions:

- Each question should refer to the audio file as 'the audio'.
- Ensure the questions are diverse and focus on testing knowledge of music quality and texture in the audio.
- Make questions which can make you think about some knowledge in terms of texture of the music.
- Music Texture questions can be related to knowledge of quality of the audio, texture, characterics of a voice in the audio,
and other features related to texture and quality. Ask music texture questions in an indirect way.
- Make use of captions provided to find the answer.
- Do  not ask question about mood, emotions and atmosphere of the audio.
- Format your output precisely as specified.
- Only return the JSON object, nothing else

The expected JSON structure is:

    {{
        'audio_id': {path},
        'question': <question>,
        'options': [<option1>, <option2>, <option3>, <option4>],
        'answer': <correct_option>,
        'difficulty': <'easy'/'medium'/'hard'>,
        'category': <'Music Texture'>
    }}

Below is the input:
```

Figure 32: Prompt 3 used for generating **Information Extraction QA** for AudioSkills.

```
#Prompt

Your task is to generate a diverse set of music knowledge questions and their corresponding answers for each row in the dataset. Each question and answer
pair should be formatted as a JSON object.

These questions will be used to test Audio Language Models, so ensure they are directly relevant to the audio content described in the captions. You should
assign each question an appropriate difficulty level: 'easy', 'medium', or 'hard'. If an answer may vary because different models might generate similar but
not identical responses, frame the question as a multiple-choice question (MCQ).

You will be provided with two captions: a music caption and an alternate caption. Both contain details about the music in the audio and their
characteristics. Use the information from these captions as the ground truth to create multiple questions for each audio, focusing questions about the
**Melody** in the audio.

Instructions:

- Each question should refer to the audio file as 'the audio'.
- Ensure the questions are diverse and focus on testing knowledge of rhythm and tempo in the audio.
- Avoid questions like "What is the tempo in the audio?", this question is very direct and lame. Rather make questions which can make you think about some
knowledge in terms of rhythm and tempo.
- Rhythm and tempo questions can be related to knowledge of tempo, beats, rhythm, time signature and other features related to rhythm. Ask rhythm and tempo
questions in an indirect way.
- Make use of captions provided to find the answer.
- Do not ask question about mood and emotions.
- Format your output precisely as specified.
- Only return the JSON object, nothing else

The expected JSON structure is:

    {{
        'audio_id': {path},
        'question': <question>,
        'options': [<option1>, <option2>, <option3>, <option4>],
        'answer': <correct_option>,
        'difficulty': <'easy'/'medium'/'hard'>,
        'category': <'rhythm and tempo'>
    }}

Generate the result for:<temp_string>
audio_id path: <path>
```

Figure 33: Prompt 4 used for generating **Information Extraction QA** for AudioSkills.

```
#Prompt

I will provide you with an audio caption that describes the order in which multiple sounds occur (using words like 'with', 'after', 'followed by', etc.).
Additionally, I will provide a list of sound events with duration information (e.g., Rain – 0.000-10.000; the start time is 0.000 seconds, and the end time
is 10.000 seconds, so the duration is end time – start time = 10.000 seconds). Your task is to generate complex multiple-choice questions (MSQs) with four
options. I have provided the question type and example question, answers below

Source Information Extraction
(Identifying a source of a sound in a complex acoustic scene)
audio caption: "A fire truck's siren can be heard over traffic and human voices."
List of audio events: "['(Fire engine, fire truck (siren)-0.000-10.000)', '(Traffic noise, roadway noise-0.000-10.000)', '(Human voice-8.546-9.227)']"

Type 1. Source Identification in Complex acoustic scene
Example Question: For the given audio, identify the source of the siren.
Options:
1. fire truck
2. Ambulance
3. Police Car
4. Boat
Correct Option:
fire truck
Rules for generating option:
1. Skip the questions for which the sound is background sound or noise
2. When generating the options, identify the multiple sources that can produce the same sound as incorrect options. For example, a siren can be generated by
an ambulance or a police car, but based on the overall acoustic scene described by the audio caption and the list of sound events, it can be inferred that
the most likely vehicle is a fire truck.

Reasoning:
Identify whether other sounds occurring in the audio can describe the acoustic scene, and then determine what type of source could have generated the sound.

While designing the options, ensure they contrast clearly with the correct choice, making it easier for a model to use reasoning skills to identify the true
cause of the child's emotional response.
Guidelines:
1. Always ask about events with a significant duration. Use the start and end times provided to determine the duration after each event. Ignore insignificant
events, such as background or static noise, which could confuse listeners due to their ambiguous sources.
2. Each questions must be prepended by phrases like 'Based on the given audio', 'For the given audio sample', 'Given the audio sample', etc.
Output Format:
The output format must be json. I want you to generate 1 question for each categories. Here is an example:
{'Question type': <question type>, 'Question': <generate questions>, 'Options': [<choice1>, <choice2>, <choice3>, <choice4>], 'Answers': <correct option>,
Reason: <reason for selecting a option>}. Make sure question, options and answers must not exceed 20, 10, 10 words.
Here is the audio caption: <ap>
time aligned audio event: <ae>
```

Figure 34: Prompt 5 used for generating **Information Extraction QA** for AudioSkills.

```
#Prompt

You are an expert AI assistant that is knowledgeable about music production, musical structure, music history, and music styles, and you are hearing audio of
a short clip of music. What you hear is described in the JSON-formatted caption below, describing the same audio clip you are listening to. Answer all
questions as if you are hearing the audio clip. This caption is provided in a JSON list of the form: [{"some_key": "some_value", "other_key":
"other_value"}], where the keys and values represent metadata about the music clip.

The JSON may contain the following fields:

'album.information': optional user-provided information about the album.
'album.tags': optional user-provided tags associated with the track album.
'artist.tags': optional user-provided tags associated with the track artist.
'track.genre_top': the top genre for the track (most frequent as determined by user votes).
'track.genres_all': all genre labels for the track.
'track.information': optional user-provided information about the track.
'track.language_code': the language of the track.
tempo_in_beats_per_minute_madmom: the tempo of the track in beats per minute (BPM).
downbeats_madmom: a list of the downbeats in the song, containing their timing ("time") and their associated beat ("beat_number"). For example, beat_number 1
indicates the first beat of every measure of the song. The maximum beat_number indicates the time signature (for instance, a song with beat_number 4 will be
in 4/4 time).
chords: a list of the chords of the song, containing their start time, end time, and the chord being
played.
key: the key of the song.

Design a conversation between you and a person asking about this music. The answers should be in a tone that an AI assistant is hearing the music and
answering the question. Ask diverse questions and give corresponding answers.
Ask factual questions about the musical characteristics and content of the song, including the style and emotions, audio characteristics, harmonic structure,
presence of various instruments and vocals, tempo, genre, relative ordering of events in the clip, etc.

Only include questions that have definite answers based on the provided metadata or your background knowledge of this specific music as an intelligent AI
assistant. Write as many question as you can using the provided inputs. Try to include a mixture of simple questions ("Is there a saxophone in the song?"
"Are there vocals in the clip?" "What is the approximate tempo of the clip in beats per minute (BPM)?") and more complex questions (""How would you describe
the overall mood and emotions conveyed by the song?"). Make the questions as diverse as possible, and ask about as many different aspects of the song as
possible. Do not mention the name of the artist in the response.

Again, do not ask about uncertain details. Provide detailed answers when answering complex questions. For example, give detailed examples or reasoning steps
to make the content more convincing and well-organized. Explain any musical concepts that would be unfamiliar to a non-musician. You can include multiple
paragraphs if necessary. Make sure that the generated questions contain questions asking about the musical characteristics and content of the song. If there
are multiple plausible answers to a question, make sure to mention all of the plausible choices. Do not specifically reference the provided metadata in the
response; instead, respond as if you are hearing the song and reporting facts about what you hear.

IMPORTANT: Do not use the word "metadata" anywhere in the answers to the questions. DO NOT disclose that metadata about the song is provided to you. Always
answer as if you are an expert who is listening to the audio.

Return a single JSON list object containing the question-answer pairs. Each element in the JSON list should be a JSON object that has the following
structure: {"question": "<QUESTION TEXT GOES HERE>", "answer": "<ANSWER TEXT GOES HERE>"}
```

Figure 35: Prompt used for generating **QA** for AudioSkills using FMA.

```
#Prompt

You are an expert AI assistant that is knowledgeable about music production, musical structure, music history, and music styles, and you are hearing audio of
a short clip of music. What you hear is described in the JSON-formatted outputs below, describing the same audio clip you are listening to. Answer all
questions as if you are hearing the audio clip. This description is provided in a JSON dictionary, where the keys and values represent events in the music
clip.

The JSON dictionary contains the following keys: "composer", "composition", "movement", "ensemble", "notes".

The main component of the JSON is the "notes" field, which is a nested JSON dictionary. The keys in "notes" represent individual instruments, and the values
is a JSON list representing all of the notes played by that instrument in the music clip. Each element in the value JSON list represents one note played in
the music, and includes the following keys:
- start: the start time of the note, in seconds
- end: the end time of the note, in seconds
- pitch: the pitch and octave of the note

In addition to these fields, the JSON also contains the following special annotations:
        - tempo_in_beats_per_minute_madmom: the tempo of the track in beats per minute (BPM).
        - downbeats_madmom: a list of the downbeats in the song, containing their timing ("time") and
their associated beat ("beat_number"). For example, beat_number 1 indicates the first beat of every measure of the song. The maximum beat_number indicates
the time signature (for instance, a song with beat_number 4 will be in 4/4 time).
        - chords: a list of the chords of the song, containing their start time, end time, and the chord being played.
        - key: the key of the song.
Provide a detailed musical description of the clip, from the perspective of a musical expert describing the clip as they hear it being played. Make sure to
describe the ordering of the different instruments (which plays first, which plays at the end), themes or rhythms, arpeggios, chords, repeating patterns,
etc.

The answers should be in a tone that an AI assistant is hearing the music and describing it to a
listener.

Only provide details that are based on the provided metadata or your background knowledge of music as an intelligent AI assistant. Assume that there are no
notes or instruments in the clip besides those in the "notes" data. Explain any musical concepts that would be unfamiliar to a non-musician. You can include
multiple paragraphs if necessary. Do not specifically reference the provided metadata in the response; instead, respond as if you are hearing the song and
reporting a rich description of what you hear. The descriptions should keep in mind that this may only be an
excerpt or part of a song, and not the complete song.

IMPORTANT: Do not use the word "metadata" anywhere in the answers to the questions. DO NOT disclose that metadata about the song is provided to you. Do not
specifically reference the instruments by number (do not say "Violin 1" or "Violin 2"; instead just say "a violin"). Focus more on a high-level description
of the audio, and do not simply list the notes being played; specific notes (i.e. G5 or F#0) should only be mentioned if they are particularly important to
the description of the song. Always answer as if you are an expert who is listening to the audio. Do not mention or ask about the track title, artist, or
album.
```

Figure 36: Prompt used for generating **QA** for AudioSkills using MusicNet, borrowed from Gardner et al. (2023).

```
#Prompt

You are an expert AI assistant that is knowledgeable about music production, musical structure, music history, and music styles, and you are hearing audio of
a short clip of music. What you hear is described in the JSON-formatted caption below, describing the same audio clip you are listening to. Answer all
questions as if you are hearing the audio clip. This caption is provided in a JSON list of the form: [{"some_key": "some_value",
"other_key": "other_value"}], where the keys and values represent metadata about the music clip.

The JSON may contain the following fields:
        genre: a list of genres associated with the song.
        instrument: a list of instruments known to be in the song. Other instruments not listed here may also be present. If the song contains vocals, they
will not be mentioned here.
        mood/theme: a list of moods or themes associated with the song.
        tempo_in_beats_per_minute_madmom: the tempo of the track in beats per minute (BPM).
        downbeats_madmom: a list of the downbeats in the song, containing their timing ("time") and their associated
beat ("beat_number"). For example, beat_number 1 indicates the first beat of every measure of the song. The
maximum beat_number indicates the time signature (for instance, a song with beat_number 4 will be in 4/4 time).
        chords: a list of the chords of the song, containing their start time, end time, and the chord being played.
        key: the key of the song.

Design a conversation between you and a person asking about this music. The answers should be in a tone that an AI assistant is hearing the music and
answering the question. Ask diverse questions and give corresponding answers.

Only ask questions that require complex reasoning about the content in the music, possibly combined with other background knowledge. Here are some examples
of complex questions that you could ask:
- Ask about background knowledge about the music.
- Ask for songs or artists with a similar style.
- Ask about the order of events in the audio, for example, "What comes first, the drum break or the vocals?" Do the piano and the guitar play at the same
time? (For this question, only ask about instruments that are present in the track.)
- Ask about how to learn to play this type of music.
- Ask how a music producer would create the sounds heard in this track.
- Ask about how to change the music in a specific way, for example, to make it better, happier, more danceable, or to sound like another genre.
- Ask how a music professor would describe the track.
- Ask about any cultural, historical or popular references related to this track, in terms that the general public would use.
- Ask to describe the scenarios in which people would listen to this track, again in terms that the general public would use.
- List instructions that could be provided to an AI in order to generate music that is similar to this song, without using the word similar or a reference to
this particular song.

Do NOT ask basic questions that can be answered with a single attribute of the JSON such as:
- What key is the song in?
- What is the genre of this song?
etc.

Only include questions that have definite answers based on the provided metadata or your background knowledge ofthis specific music as an intelligent AI
assistant. Write as many question as you can using the provided inputs. Make the questions as diverse as possible, and ask about as many different aspects of
the song as possible. Again, do not ask about uncertain details. Provide detailed answers to all questions. For example, give detailed examples or reasoning
steps to make the content more convincing and well-organized. Explain any musical concepts
that would be unfamiliar to a non-musician. You can include multiple paragraphs if necessary. If there are multiple plausible answers to a question, make
sure to mention all of the plausible choices. Do not specifically reference the provided metadata in the response; instead, respond as if you are hearing the
song and reporting facts about what you hear. IMPORTANT: Make sure the provided answers do not contain the phrases "the metadata" "based on the provided
metadata". DO NOT disclose that metadata about the song is provided; always answer as if you are an expert who is listening to the audio.

Make sure that the questions are complex, and that the detailed answers reflect your expertise as an expert AI assistant that is knowledgeable about music
production, musical structure, music history, and music styles listening to the clip.

Please return a single JSON list object containing the question-answer pairs. Each element in the JSON list should be a JSON object that has the following
structure: {"question": "<QUESTION TEXT GOES HERE>", "answer": "<ANSWER TEXT GOES HERE>"}
```

Figure 37: Prompt used for generating **QA** for AudioSkills using MTG-Jamendo, borrowed from Gardner et al. (2023).

```
#Prompt

You are an expert AI assistant. I will provide you with:
A JSON object containing:
    -   The path to the audio file.
    -   A question about the audio.
    -   An answer to that question.
    -   A series (list) of captions, where each entry contains: a vision caption and an audio caption respectively
    -   Each pair of captions corresponds to a 10-second segment of the audio clip. You should consider only the audio caption which is the second elemnet in
every list.

Your task is to validate this JSON based on the following criteria:
    1. Correctness of Answer: The answer must correctly address the given question in the context of the audio.
    2. Relevance of Question: The question must be related to the audio content.
    3. Audio-Focus: Both the question and the answer should be about the audio, not about the vision component.
    4. No question or answers should be related to what any person is speaking in specific.
    5.  Remove visual cues from the question and answer: If the above 3 criterias meet, but the question and answers contain visual cues, please rephrase the
question and answer to remove visual cues.

If all these conditions are met, you must return a JSON object with "check": "pass". Otherwise, return the same JSON object but with "check": "fail".

Use the following output format:

{
    "name": "<path_of_the_audio>",
    "prompt": "<question>",
    "output": "<answer>",
    "check": "<pass_or_fail>"
}

Below is the input:
```

Figure 38: Prompt used for **self-verification** of generated QA pairs for MiraData.

```
#Prompt

You are an expert AI assistant. I will provide you with:
A JSON object containing:
    - The path to the audio file.
    - A question about the audio.
    - An answer to that question.
    - A series (list) of captions, where each entry contains: an action caption about the action being performed by the user, an audio caption and a vision
caption respectively.
    - Each pair of captions corresponds to a 4-second segment of the audio clip.

Your task is to validate this JSON based on the following criteria:
    1. Correctness of Answer: The answer must correctly address the given question in the context of the audio.
    2. Relevance of Question: The question must be related to the audio content.
    3. Audio-Focus: Both the question and the answer should be about the audio, not about the vision component.
    4. No question or answers should be related to what any person is speaking in specific.
    5. Remove visual cues from the question and answer: If the above 3 criterias meet, but the question and answers contain visual cues, please rephrase the
question and answer to remove visual cues.

If all these conditions are met, you must return a JSON object with "check": "pass". Otherwise, return the same JSON object but with "check": "fail".

Use the following output format:

{
    "name": "<path_of_the_audio>",
    "prompt": "<question>",
    "output": "<answer>",
    "check": "<pass_or_fail>"
}

Below is the input:
```

Figure 39: Prompt used for **self-verification** of generated QA pairs for ReCap dataset.

Table 27: Prompts designed for specific audio processing tasks, used to transform foundational audio understanding datasets into QA pairs.

| Task Name | Prompts |
|---|---|
| Audio Event Classification | What are the unique sounds in the audio?, Provide a comma separated list of all sounds you hear in the input audio., What sounds can you hear in the audio?, List all the sounds present in the audio., Identify the distinct sounds in the audio clip., What specific sounds are detectable in the audio?, Give a list of sounds you can identify in this audio., What types of sounds are included in the audio?, List the audible elements in the audio., What are the main sounds you notice in this audio?, Provide a list of all recognizable sounds in the audio., What sounds are prominent in the audio clip?, List all distinct sounds in the audio., What are the different sounds occurring in this audio? |
| Sentiment Classification | Identify the sentiment of the utterance., What is the sentiment of the utterance?, What is the primary sentiment of the utterance in this recording? |
| Music Understanding and Classification | Generate music tags including genre, instrument, and mood, Identify the genre, instrument, and theme of this music, What are the tags for genre, instrument, and mood for this track, Describe the genre, instruments, and theme of the audio, What is the genre and mood of the music, Provide tags for the genre, mood, and instruments in the track, What music tags best describe this audio, including genre and theme, Generate tags for the genre and atmosphere of the music, What genre and instruments are featured in this track, What are the primary genre and theme of this audio, Describe the mood, genre, and instrumentation in the music, Identify the genre and musical style of the audio |
| Genre Classification | What is the genre of the music, Identify the genre of this audio clip, Classify the genre of the music in the audio, What musical genre does this clip represent, Provide the genre tag for this music, Which genre best describes the audio, What is the genre label for this music clip, Classify the genre based on the audio content, What category of music does this audio belong to, Determine the genre of the audio |

