# OpenReview forum: "Audio Flamingo 2: An Audio-Language Model with Long-Audio Understanding and Expert Reasoning Abilities"
_ICML.cc/2025/Conference — ICML 2025 poster_

### Official Review · Reviewer_bZqQ · 2025-03-08

**Overall Recommendation:** 4

**Summary:**

This paper introduces Audio Flamingo 2 (AF2), an advanced Audio-Language Model (ALM) designed for long-audio understanding and expert-level reasoning. AF2 leverages a custom CLAP model, synthetic Audio Question Answering (AQA) data, and a multi-stage curriculum learning strategy to achieve state-of-the-art performance in audio understanding and reasoning tasks. The model outperforms existing large-scale models despite having only a 3B parameter language model. Key contributions include the proposal of a new dataset, LongAudio, for long-audio understanding, and a benchmark, LongAudioBench, for evaluating ALMs on long audio segments. The paper also details innovations in data generation, architecture design, representation learning, and training strategies. AF2 demonstrates superior performance across 20+ benchmarks, including foundational audio understanding and reasoning tasks.

**Claims And Evidence:**

The claims made in the paper are well-supported by clear and convincing evidence. The authors demonstrate that AF2 outperforms existing state-of-the-art models, such as GAMA, Qwen-Audio, and Audio Flamingo, across multiple benchmarks. The performance improvements are attributed to the use of high-quality training data (AudioSkills), an improved audio encoder (AF-CLAP), and a novel three-stage curriculum training strategy. The paper provides extensive ablation studies to validate the impact of each design choice, including the use of different audio encoders, training data compositions, and training schedules. The results show that the proposed methods significantly enhance the model's reasoning capabilities and long-audio understanding.

**Essential References Not Discussed:**

N/A

**Experimental Designs Or Analyses:**

The experimental designs and analyses presented in the paper are sound and valid. The authors conduct extensive experiments to compare AF2 with existing state-of-the-art models across various benchmarks. The use of a three-stage curriculum training strategy is validated through ablation studies, demonstrating the importance of gradual context-length extension and data quality. The paper also explores the impact of different audio encoders and training data compositions, providing insights into the factors contributing to the model's performance. The results are consistent and highlight the effectiveness of the proposed methods.

**Methods And Evaluation Criteria:**

The methods and evaluation criteria proposed in the paper are well-suited for the problem of long-audio understanding and reasoning. The authors introduce LongAudio, a large-scale dataset with over 260k AQA instances, specifically designed for training and evaluating ALMs on long audio segments. The benchmark LongAudioBench further validates the model's performance on long-audio tasks. The evaluation criteria include both foundational audio understanding tasks (e.g., classification, captioning) and reasoning tasks (e.g., AQA, entailment), providing a comprehensive assessment of the model's capabilities. Popular benchmarks are also utilized.

**Other Comments Or Suggestions:**

See Questions.

**Other Strengths And Weaknesses:**

Strengths:
Originality: The paper introduces several novel contributions, including the LongAudio dataset, the AF-CLAP audio encoder, and the multi-stage curriculum training strategy. These innovations collectively enhance the model's capabilities and set it apart from existing ALMs.
Significance: The focus on long-audio understanding and expert reasoning addresses a gap in the current literature. The proposed methods and datasets are likely to drive further progress in this area.
Clarity: The paper is well-written and provides a clear description of the methods, experiments, and results. The ablation studies and detailed analyses help readers understand the impact of each design choice. The paper contains enough details.

Weakness:
See Questions.

**Questions For Authors:**

1. Will the dataset be released? It seems to be a huge project if the followers of this paper want to reproduce the dataset.

2.  Why the authors filter out segments with audio-visual similarity above a threshold? Why a high relevance on audio and vision is not a good signal indicating a good data? How to filter out the video whose image are not quite relevant to the audio (e.g., a video with background music added)?

3. The audio feature seems to have a length of 64*2048, how to use it to calculate similarity with T5 feature? Mean-pooling or a linear?

4. Currently the AF2 seems to be an expert model on audio understanding. Can it be extended to 1) multiturn, 2) speech scenario (i.e., accept a spoken question and then answer)?

**Relation To Broader Scientific Literature:**

The key contributions of this paper are closely related to the broader scientific literature on audio-language models and multi-modal learning. The work builds upon previous efforts in contrastive learning for audio-language alignment (e.g., CLAP, Wav2CLIP) and extends them by introducing a custom audio encoder (AF-CLAP) and a large-scale dataset for long-audio understanding (LongAudio). The focus on expert-level reasoning aligns with recent trends in developing models capable of complex reasoning tasks (e.g., GAMA, Qwen2-Audio, SALMONN).

**Theoretical Claims:**

This is not a theoretical paper.

---

> ### Author Rebuttal · Authors · 2025-03-26
>
> We thank you for your thorough review and constructive feedback. We have tried to address each of your concerns point by point.
>
> > Will the dataset be released? It seems to be a huge project if the followers of this paper want to reproduce the dataset.
>
> **Ans.**: Yes, absolutely! As stated on Page 1 of our paper, we will open-source all code, data, and model checkpoints. We have all approvals and plan to release the dataset publicly at the time of ICML notifications (if accepted).
>
> For the purpose of this rebuttal, we are providing access to a part of the QA data at this [anonymous link](https://github.com/anon-sub-openreview/af2). We kindly ask that you do not share this dataset for now.
>
> > Why the authors filter out segments with audio-visual similarity above a threshold? Why a high relevance on audio and vision is not a good signal indicating a good data? How to filter out the video whose image are not quite relevant to the audio (e.g., a video with background music added)?
>
> **Ans.**: We would like to clarify a potential misunderstanding. As stated in Lines 158-160 of the paper, audio-visual similarity is **not** used to discard low-quality segments—it is used to discard **redundant** or highly similar consecutive segments. Our goal is to promote diversity in the training data. Here's how the method works:
>
> 1. We segment each video (and its corresponding audio) into 10-second clips.
> 2. For each 10-second segment, we compute a **visual embedding** as follows:
>    - We extract the middle frame from every 1-second interval (10 frames total).
>    - Each frame is passed through CLIP to get an embedding, resulting in $\mathbb{R}^{10 \times 768}$.
>    - These are mean-pooled to obtain a single visual embedding of shape $\mathbb{R}^{1 \times 768}$.
> 3. We compute the **audio embedding** of the 10-second audio using LAION-CLAP, also resulting in $\mathbb{R}^{1 \times 768}$.
> 4. We then average the audio and visual embeddings to obtain a single **audio-visual embedding** per segment: $\mathbb{R}^{1 \times 768}$.
> 5. Finally, we compute the cosine similarity between consecutive audio-visual embeddings and discard those with high similarity. This ensures diversity in both content and modality for CLAP training.
>
> To address the last part of your question: if a video has low audio-visual correspondence (e.g., background music over unrelated visuals), the embeddings tend to be **highly similar** across segments. As a result, such segments are **filtered out**, since they lack useful diversity.
>
> We will rewrite this line to make it clearer in the final version of our paper.
>
> > The audio feature seems to have a length of 64*2048, how to use it to calculate similarity with T5 feature? Mean-pooling or a linear?
>
> **Ans.**:  Thank you for the question. We would like to clarify a potential misunderstanding.
>
> We follow a similar setup to LAION-CLAP for training our model, using **HTSAT-Large** as the audio encoder. The original HTSAT-Large model produces a feature embedding of shape $64 \times 2048$ for any 10-second audio clip.
>
> For CLAP training, these embeddings are **mean-pooled** and passed through a **linear projection layer**. Specifically:
> - Audio representations of shape $n \times 2048$ (after pooling) are obtained for a batch of size $n$.
> - These are then compared to text features from CLAP, also of shape $n \times 2048$, to compute similarity.
>
> This linear layer is trained jointly with the rest of the CLAP model.
>
> To integrate our trained CLAP with the LLM and build Audio Flamingo 2, we **remove the final linear layer** and instead use the raw $64 \times 2048$ embeddings from the last layer of HTSAT as input to the LLM. This provides the model with richer temporal representations for downstream multimodal understanding.
>
> > Currently the AF2 seems to be an expert model on audio understanding. Can it be extended to 1) multiturn, 2) speech scenario (i.e., accept a spoken question and then answer)?
>
> **Ans.** Thank you for the question. Yes, absolutely!
>
> 1) **Multi-turn Extension:** Just like the original Audio Flamingo, extending to multi-turn interaction is straightforward. We simply include the dialogue history in the text context and condition on prior audio inputs. These are separated using the special `<sep>` token, which is already a learned component in AF2.
>
> 2) **Speech Scenario Extension:** To support speech input, we add speech data to the training corpus. Empirically, we found that:
> - Incorporating **ASR data** during Stage 1, and
> - Including **Speech QA data** during Stage 2 significantly improves AF2’s speech understanding capabilities.
>
> Furthermore, we observed additional benefits when replacing CLAP with a **custom Whisper-based encoder** capable of handling both speech and music. This modification specifically enhances performance on ASR and speech-to-text translation tasks. Extending AF2 to better handle speech is an active area of our ongoing and future work.

---

### Official Review · Reviewer_VWKg · 2025-03-13

**Overall Recommendation:** 5

**Summary:**

This paper propose Audio flamingo 2, an audio language model with advanced audio understanding and reasoning abilities, demonstrated by state of the art performance on several benchmarks. The authors develop a custom clap model, a dataset called LongAudio to extend ALMs to 30s-5 minute audios, and another called AudioSkills with a focus on reasoning, and demonstrate the usefulness of curriculum learning. Also introduced is AF-CLAP, a novel method of doing contrastive pretraining, and 3-stage curriculum learning. The paper shows the use of LLMs to label synthetic data and has a demonstration of high-quality reasoning in the audio domain, all at a smaller size than competitor ALMs.

**Claims And Evidence:**

1) The authors claim that improved data quality can trump scaling of compute/model size in terms of performance. They show this by developing two high quality datasets (LongAudio and AudioSkills) and comparing scaling with and without the high quality datasets, which shows considerable performance differences, and the claim is therefore well justified.
2) The authors claim that their AF-CLAP method is superior to other contrastive language-audio pre-training methods. They justify this claim by training other types of CLAP methods on the same data as their proposed method, and show that the resulting performance does not measure up to AF-CLAP.
3) The authors claim that curriculum learning is helpful and justify this by running detailed comparisons against different training schedules and showing results on a variety of benchmarks.

**Essential References Not Discussed:**

I did not find major references missing

**Experimental Designs Or Analyses:**

The experiments are set up thoughtfully. Various training strategies are considered, different architectural decisions are ablated, dataset design and benchmark creation is well-done. Experimental design is excellent overall.

**Methods And Evaluation Criteria:**

The methods are well justified, the evaluation criteria is correct and comprehensive.

**Other Comments Or Suggestions:**

None

**Other Strengths And Weaknesses:**

Strengths:
1. The breadth and comprehensiveness of the experimental work done here is impressive and is a clear indication of the strength of the results.
2. The scaling and data quality experiments are compelling, and the invention of new datasets are a welcome addition to the community.
Weaknesses:
1. The section explaining the training objective could be improved by giving slightly more intuition to how the loss directly contributes to the goals of improving linguistic invariance and compositional reasoning.
2. The novelty of the paper is in the connection of various components and the invention of new datasets, not necessarily in new insights into the modeling method apart from the loss function. This is, however, a minor weakness and somewhat universal in large-scale pretraining focused work.

## update after rebuttal
I am satistied with the authors responses to my question and will keep my score

**Questions For Authors:**

I do not have any questions of the authors.

**Relation To Broader Scientific Literature:**

The paper builds upon Audio Flamingo, it's predecessor model, and is directly inspired by contrastive language-audio pretraining. It uses several methods from prior work, including HTSTAT for generating audio embeddings and T5 for text embeddings. It relies on benchmarks that are well-suited to the tasks of reasoning and audio understanding that it is attempting to do. The authors also extend the ongoing discussion in the community around data quality via detailed data collection and ablations that specifically prove it's usefulness. Broadly the paper engages deeply with the existing audio understanding literature.

**Theoretical Claims:**

The paper is primarily empirical, there are no real proofs to check.

---

> ### Author Rebuttal · Authors · 2025-03-26
>
> Thank you for the encouraging review. We are happy you liked our paper. We'd just like to clarify that, in addition to the dataset contributions, our paper also presents several modeling insights that we believe are novel and impactful:
>
>
> - **Dynamic batching for efficient training:** As described in Appendix H.2, we introduce a dynamic batching strategy based on audio length. This significantly reduces padding, improves training efficiency, and yields faster convergence with better model performance.
>
> - **Efficiency in long audio modeling:** Section 3.2 explains how our AF2 module acts as an effective alternative to prefix-based architectures for long audio inputs. By leveraging cross-attention instead of increasing context length, we avoid the quadratic time complexity associated with standard attention mechanisms.
>
> - **Superiority of cross-attention over prefix tuning:** Section 6.5 demonstrates that cross-attention outperforms prefix tuning on the same dataset, indicating its effectiveness for audio-text alignment and long-context understanding.
>
> - **Curriculum learning strategy:** In Section 6.7, we compare 10 training schedules and show that our proposed curriculum learning strategy consistently achieves the best performance. We further highlight a counterintuitive insight: fine-tuning the language model weights degrades performance, and curriculum design is crucial for reasoning over long-form audio.
>
> - **High-quality data over large models:** Finally, Section 6.6 shows that training smaller LLMs on high-quality audio-text data results in performance that matches or exceeds that of larger LLMs trained on lower-quality data—underscoring the importance of data quality over model size for audio reasoning.

---

### Official Review · Reviewer_s3Xm · 2025-03-13

**Overall Recommendation:** 3

**Summary:**

This paper introduces Audio Flamingo 2 (AF2), a small yet powerful Audio-Language Model (ALM) with advanced audio understanding and reasoning capabilities. AF2 leverages a custom CLAP model, synthetic AQA data, and a multi-stage curriculum learning strategy to achieve state-of-the-art performance across 20+ benchmarks. It extends audio understanding to long audio segments (30 secs - 5 mins) and introduces LongAudio, a novel dataset for long audio captioning and QA tasks.

**Claims And Evidence:**

The claims in the submission are generally supported by clear and convincing evidence. The paper presents Audio Flamingo 2 (AF2) and provides extensive experimental results across 20+ benchmarks, showing state-of-the-art performance.

**Essential References Not Discussed:**

No.

**Experimental Designs Or Analyses:**

The authors conducted extensive ablation studies to confirm the efficacy of their approach, including comparisons of different CLAP models, data compositions, training schedules, and LLM sizes. They also introduced LongAudioBench, an expert-annotated benchmark for evaluating ALMs on long audio understanding. However, the reliance on synthetic data and LLM-as-a-judge framework for evaluation could introduce biases or limitations in the results.

**Methods And Evaluation Criteria:**

The proposed methods and evaluation criteria are well-suited for the problem of audio understanding and reasoning. The introduction of LongAudio and LongAudioBench provides relevant datasets for evaluating long audio understanding, making the evaluation criteria comprehensive and appropriate.

**Other Comments Or Suggestions:**

**Data Accessibility**: While the authors mention open-sourcing code and data, providing a clear timeline or repository link would enhance accessibility.

**Other Strengths And Weaknesses:**

Strengths:
1. Proposes innovative data generation, architecture design, and training strategies.
2. Achieves SOTA performance across 20+ benchmarks with a smaller model.

Weaknesses:
1. Limited discussion on the model's ability to understand speech content.
2. Potential over-reliance on synthetic data for training, which may affect generalization.

**Questions For Authors:**

What are the potential limitations of using GPT-4o for generating QA pairs in the AudioSkills dataset, and how did you mitigate these limitations?

**Relation To Broader Scientific Literature:**

Not applicable.

**Theoretical Claims:**

The paper primarily focuses on empirical results and architectural innovations rather than presenting formal theoretical proofs. It describes the model's design, training strategies, and performance on various benchmarks without delving into theoretical derivations or proofs.

---

> ### Author Rebuttal · Authors · 2025-03-26
>
> We thank you for your thorough review and constructive feedback. We have tried to address each of your concerns point by point.
>
> > Data Accessibility: While the authors mention open-sourcing code and data, providing a clear timeline or repository link would enhance accessibility.
>
> **Ans.** As stated on Page 1 of our paper, we will open-source all code, data, and model checkpoints. We have all approvals and plan to release the dataset publicly at the time of ICML notifications (if accepted). This will include all QAs and audios released on GitHub and HuggingFace.
>
> For the purpose of this rebuttal, we are providing access to a part of the QA data at this [anonymous link](https://github.com/anon-sub-openreview/af2). We kindly ask that you do not share this dataset for now.
>
> > Limited discussion on the model's ability to understand speech content.
>
> As mentioned in the Abstract and Introduction, Audio Flamingo 2, like its predecessor, primarily focuses on sounds and music. Speech content understanding is intentionally out of scope for AF2, aligning with other models in this family such as Pengi [1], GAMA [2], and LTU [3].
>
> That said, extending AF2 to handle speech is part of our ongoing and future work. We would like to share some preliminary insights:
>
> - **Minimal changes needed for speech support:** Incorporating speech understanding simply requires adding speech data to the training corpus.
> - **Stage-wise improvements:** We observe that including ASR data in Stage 1 and Speech QA data in Stage 2 significantly improves AF2’s performance on speech tasks.
> - **Replacing CLAP with Whisper:** Using a customized Whisper model in place of CLAP enhances AF2’s ability to process speech alongside music, improving performance on ASR and speech-to-text translation tasks.
>
> We are actively working on these extensions and will explore them more fully in future work.
>
> > Potential over-reliance on synthetic data for training, which may affect generalization.
>
> **Ans.** Thank you for the thoughtful question! Most of the audio used in the *AudioSkills* and *LongAudio* datasets is sourced from real-world recordings (as mentioned in the paper). The only exception is the *Counting* skill, which accounts for less than 1% of the total audio data (see Table 14 for detailed statistics). While the question-answer QA pairs are synthetically generated, this modality is inherently artificial—even in human annotations. Importantly, our use of real-world audio ensures there is no sim-to-real gap during training or inference.
>
> > However, the reliance on synthetic data and LLM-as-a-judge framework for evaluation could introduce biases or limitations in the results.
>
> **Ans.** Thank you for raising this concern. We adopt the *LLM-as-a-judge* evaluation framework following widely used practices in long-form video and multimodal benchmarks such as Video-MME [5], as well as other recent LLM evaluation benchmarks for audio understanding such as CompA-R [1] and AIR-Bench [5]. Prior work [4] has demonstrated that this framework provides a more robust and semantically grounded measure of generation quality, especially for open-ended tasks. It also mitigates common issues in traditional evaluation approaches, such as overly strict regex matching or limited coverage of reference answers.
>
> To reduce potential bias, we carefully designed our evaluation prompts (see Fig. 25) to be neutral, consistent, and faithful to the context of each QA pair. We acknowledge that synthetic evaluation frameworks have limitations, and we view this as an active area of improvement for future work.
>
>
> ### Citations
>
> [1] GAMA: A Large Audio-Language Model with Advanced Audio Understanding and Complex Reasoning Abilities (https://aclanthology.org/2024.emnlp-main.361/).
> [2] Pengi: An Audio Language Model for Audio Tasks (https://openreview.net/forum?id=gJLAfO4KUq).
> [3] Listen, Think, and Understand (https://openreview.net/forum?id=nBZBPXdJlC).
> [4] AIR-Bench: Benchmarking Large Audio-Language Models via Generative Comprehension (https://aclanthology.org/2024.acl-long.109/).
> [5] Video-MME The First-Ever Comprehensive Evaluation Benchmark of Multi-modal LLMs in Video Analysis (https://arxiv.org/abs/2405.21075).
> [6] Visual Description Grounding Reduces Hallucinations and Boosts Reasoning in LVLMs (https://openreview.net/forum?id=3PRvlT8b1R)

---

### Official Review · Reviewer_Y32n · 2025-03-13

**Overall Recommendation:** 4

**Summary:**

This paper introduces a state-of-the-art audio understanding LLM, with a focus on long and complex acoustic scenes. Audio understanding has so far been limited to superficial captioning of individual sound events often generating artificially inflated captions that try to give an illusion of complexity while having no more substance than a simple classifier. Thus, developing models that show an actual ability to analyze complex audios in depth, and provide precise sound scene analyses is of wide interest to the audio community. This paper significantly improves over the previous sota in that regard, being competitive with powerful baselines such as Gemini. However, my main concern is about the nature of this papers' contribution: while a few interesting training ideas are introduced, the framework and architecture are almost identical to the original Audio Flamingo, while the training data brings most of the improvement in performance. As a reader, the main outcome of this paper clearly appears to be the proposed training datasets and the long context benchmark. Thus, I would like to ask the authors to explicitly explain their plan in releasing the training data and the evaluation benchmark during the rebuttal period. Moreover, a potential concern is the appropriateness of such a dataset paper for publication at ICML. I think the performance gains that this dataset provides, and the very welcome introduction of a benchmark that goes beyond superficial acoustic labeling make it worth for publication if this data is released.

**Claims And Evidence:**

Claims are clearly supported through extensive experiments.

**Essential References Not Discussed:**

N/A

**Experimental Designs Or Analyses:**

Experiments cover a wide range of audio domains and several baselines among the strongest models for audio understanding LLMs. What is a bit unusual with this paper is that it is both presented as a model paper and a dataset paper. I believe that a clearer narrative would have focused on the dataset and the evaluation, while benchmarking several existing models on this data, which would leave even more room for discussing in depth the challenges of creating audio understanding datasets.

**Methods And Evaluation Criteria:**

The paper comes with some interesting modeling ideas for contrastive training: a "weak" audio captioning model and a visual captioning one to produce rich audio captions makes a lot of sense. Similarly, using captions with correct content but incorrect chronology as negatives during contrastive training is also a simple and interesting trick.

Yet, the improvement from this methodology is marginal wrt the gains obtained from the introduced synthetic dataset.

**Other Comments Or Suggestions:**

L168,C2: "to condition audio representations on the LLM" -> "to condition the LLM on audio representations"

**Other Strengths And Weaknesses:**

N/A.

**Questions For Authors:**

Update post-rebuttal: I appreciate the authors commitment to release a dataset that will be a significant contribution to the field of audio understanding by allowing for more complex and realistic scenarios. I thus increase my score.

**Relation To Broader Scientific Literature:**

I believe the main impact will be the dataset and the long audio benchmark. The contrastive training tricks will probably also become standard, however they are likely more marginal in terms of impact.

**Theoretical Claims:**

None.

---

> ### Author Rebuttal · Authors · 2025-03-26
>
> We thank you for your thorough review and constructive feedback. We have tried to address each of your concerns point by point.
>
> > Thus, I would like to ask the authors to explicitly explain their plan in releasing the training data and the evaluation benchmark during the rebuttal period.
>
> **Ans.** As stated on Page 1 of our paper, we will open-source all code, data, and model checkpoints. We have all approvals and plan to release the dataset publicly at the time of ICML notifications (if accepted). This will include all QAs and audios released on GitHub and HuggingFace.
>
> For the purpose of this rebuttal, we are providing access to a part of the QA data at this [anonymous link](https://github.com/anon-sub-openreview/af2). We kindly ask that you do not share this dataset for now.
>
> >  I believe that a clearer narrative would have focused on the dataset and the evaluation, while benchmarking several existing models on this data, which would leave even more room for discussing in depth the challenges of creating audio understanding datasets.
>
> **Ans.** Thank you for the suggestion. We did benchmark the strongest fully open-source LALM with available training code—**GAMA**—on our dataset. As shown in Table 5, simply introducing new data is insufficient to significantly improve audio understanding performance.
>
> Our results highlight that **it is the combination of better audio perception** (via stronger audio encoder), the **cross-attention architecture**, and **curriculum learning** that leads to meaningful improvements. This holistic approach is critical for the unique challenges posed by long audio understanding.  Moreover, our architecture choice of cross-attention also supports our first exploration of **tackling long-form audio reasoning** . Section 3.2 of the paper details how our cross-attention mechanism offers a more efficient and effective alternative to prefix-based architectures, avoiding the quadratic time complexity that arises from increasing context length in standard attention mechanisms.
>
> > On contributions beyond just data (AudioSkills)
>
> - **Efficiency in long audio modeling:** Section 3.2 explains how our AF2 module acts as an effective alternative to prefix-based architectures for long audio inputs. By leveraging cross-attention instead of increasing context length, we avoid the quadratic time complexity associated with standard attention mechanisms
>
> - **Dynamic batching for efficient training:** As described in Appendix H.2, we introduce a dynamic batching strategy based on audio length. This significantly reduces padding, improves training efficiency, and yields faster convergence with better model performance.
>
> - **Superiority of cross-attention over prefix tuning:** Section 6.5 demonstrates that cross-attention outperforms prefix tuning on the same dataset, indicating its effectiveness for audio-text alignment and long-context understanding.
>
> - **Curriculum learning strategy:** In Section 6.7, we compare 10 training schedules and show that our proposed curriculum learning strategy consistently achieves the best performance. We further highlight a counterintuitive insight: fine-tuning the language model weights degrades performance, and curriculum design is crucial for reasoning over long-form audio.
>
> - **High-quality data over large models:** Finally, Section 6.6 shows that training smaller LLMs on high-quality audio-text data results in performance that matches or exceeds that of larger LLMs trained on lower-quality data—underscoring the importance of data quality over model size for audio reasoning.
>
> - **Long audio modeling:** (also acknowledged by the reviewer) We explore long audio reasoning for the first time and make attempts to build training and benchmarking datasets for this purpose.
>
> - **A proof that better and robust audio representations can improve performance:** Table 3 highlights that AF-CLAP improves Audio Flamningo 2 performance across benchmarks. Beyond the chosen benchmarks (primalrity taken from literature), as also mentioned in Section 3.1.1, AF-CLAP may help Audio Flamningo 2 performance in various real-world scenarios by expanding its breadth of audio understanding (e.g., home sounds). Finally, robust representations also help our UnusualQA task (in demo) as the representations capture fine-grained information about the input audio.

---

### Decision · Program_Chairs · 2025-05-01

**Decision:**

Accept (poster)

**Comment:**

Audio Flamingo 2 (AF2) describes training and evaluating an audio LM with improved synthetic language data, as introduced via LongAudio, LongAudioBench, and AudioSkills. The former addresses gaps in long-context evals, the latter a training dataset for reasoning over audio they shared with submission. While the architecture and curriculum are less novel (R1, R3), the datasets and focus are, plus results are SOTA over many experiments (all reviewers). Improvements due to arch choices, data quality, and scaling are analyzed. Speech content is out of scope (R2) but this is declared. Thorough experiments and new datasets (albeit synthetic w/ synthetic filtering; R2) are a benefit to machine learning, so with the reviewers I support acceptance.

(PS: Though not visible to authors, R4 writes the QA sample and author rebuttal addressed their concerns.)